# Iron dysregulation and inflammatory stress erythropoiesis associates with long-term outcome of COVID-19

Aimee L. Hanson [1,2], Matthew P. Mulè[1,2,3], Hélène Ruffieux [4], Federica Mescia [1,2], Laura Bergamaschi[1,2], Victoria S. Pelly[1,2], Lorinda Turner[1,2], Prasanti Kotagiri[1,2], Cambridge Institute of Therapeutic Immunology and Infectious Disease–National Institute for Health Research (CITIID–NIHR) COVID BioResource Collaboration*, Berthold Göttgens [5], Christoph Hess [1,2,6,7], Nicholas Gleadall[8,9], John R. Bradley[2,10,11], James A. Nathan [1,2], Paul A. Lyons [1,2], Hal Drakesmith [12] & Kenneth G. C. Smith [1,2,13,14] ✉

Persistent symptoms following SARS-CoV-2 infection are increasingly reported, although the drivers of post-acute sequelae (PASC) of COVID-19 are unclear. Here we assessed 214 individuals infected with SARS-CoV-2, with varying disease severity, for one year from COVID-19 symptom onset to determine the early correlates of PASC. A multivariate signature detected beyond two weeks of disease, encompassing unresolving inflammation, anemia, low serum iron, altered iron-homeostasis gene expression and emerging stress erythropoiesis; differentiated those who reported PASC months later, irrespective of COVID-19 severity. A whole-blood heme-metabolism signature, enriched in hospitalized patients at month 1–3 post onset, coincided with pronounced iron-deficient reticulocytosis. Lymphopenia and low numbers of dendritic cells persisted in those with PASC, and single-cell analysis reported iron maldistribution, suggesting monocyte iron loading and increased iron demand in proliferating lymphocytes. Thus, defects in iron homeostasis, dysregulated erythropoiesis and immune dysfunction due to COVID-19 possibly contribute to inefficient oxygen transport, inflammatory disequilibrium and persisting symptomatology, and may be therapeutically tractable.

Prolonged ill health following severe acute respiratory syndrome coronavirus 2 (SARS-CoV-2) infection, often termed post-acute sequelae of Coronavirus Disease 2019 (COVID-19; PASC) or 'long COVID', is defined as the unexplained continuation or development of symptoms ≥3 months from COVID-19 onset[1]. PASC is clinically complex, comprising a spectrum of often nonspecific symptomatology and is placing increasing demands on health resources worldwide[2,3]. Although estimates vary, up to 30% of all individuals infected with SARS-CoV-2, and up to 80% of those discharged from hospital, report ongoing symptoms in the 3–6 months following virus exposure, including breathing difficulties, fatigue/malaise, muscle weakness, chest/throat pain, headache, abdominal symptoms, myalgia, cognitive symptoms and anxiety/depression[3–5]. Although most frequent following severe disease, nonhospitalized individuals infected with SARS-CoV-2 also show an increased likelihood of poor health outcomes at 6 months post infection[2].

A full list of affiliations appears at the end of the paper. *A list of authors and their affiliations appears at the end of the paper. ✉e-mail: smith.k@wehi.edu.au

PASC has been associated with features of acute COVID-19 (refs. [6–8])—including the efficacy of the innate antiviral response—implying that poor viral control perpetuating ongoing inflammation, acute respiratory distress and end-organ damage may predispose individuals to ongoing symptomatology. Various predictors of PASC have been suggested, including female sex[7,9–12], increased viral load at presentation[10], lower peak SARS-CoV-2 antibody titers[6,7], increased duration of hospital stay[13] and reactivation of latent Epstein–Barr virus infection[14,15]. Immune changes persisting for months following COVID-19 have also been detected[16–19], although it is unclear whether these drive PASC or are independently reflective of acute disease severity. Immune abnormalities persist for up to 2 months from COVID-19 symptom onset in patients who require intensive care admission[20], yet longitudinal studies assessing biological and clinical features of COVID-19, with dense repeated sampling from the same individuals spanning acute infection to long-term recovery, are lacking. Such datasets are required to investigate prolonged symptoms in the context of the full disease trajectory and identify early correlates of poor outcome.

Here we present an extended longitudinal characterization of 214 SARS-CoV-2-infected individuals, from asymptomatic to requiring ventilation, who were followed for up to one year from the first SARS-CoV-2-positive swab or symptom onset. Combined analysis of longitudinal immunological, hematological, transcriptomic and clinical data indicated inflammation-driven iron dysregulation that persisted beyond 2 weeks in patients who were hospitalized with COVID-19 and which had apparent physiological repercussions for erythropoiesis and iron homeostasis months after infection. With integrated assessment of patient-reported PASC symptoms, we show that this signature of slow-resolving inflammation, iron dysregulation and ineffective compensatory stress erythropoiesis was a strong early correlate of PASC more than 3 months later.

## Results

### Immune-cell abnormalities persist following COVID-19

A total of 214 individuals PCR-positive for SARS-CoV-2 (enrolled before August 2020) were classified into five groups on the basis of peak COVID-19 severity as follows (M, male; F, female; age, median (range)): asymptomatic (group A; $n = 18$ (3 M and 15 F); 28 (20–71) yr), mild symptomatic (group B; $n = 40$ (9 M and 31 F); 31 (19–58) yr), moderate without supplemental oxygen requirement (group C; $n = 48$ (25 M and 23 F); 59.5 (17–87) yr), moderate with supplemental oxygen given as maximal respiratory support (group D; $n = 39$ (25 M and 14 F); 65 (35–87) yr) and severe with assisted ventilation (group E; $n = 69$ (52 M and 17 F); 56 (25–89) yr; Fig. 1a–c and Supplementary Table 1). All individuals in groups C–E were hospitalized and those in groups A and B were not. Matched blood, plasma and serum samples were collected at various time points up to day 352 post symptom onset or post the first positive swab for group A (hereafter post onset) and analyzed in batches that grouped samples collected in six discrete time windows (days 0–14, 15–30, 31–90, 91–180 and 181–360 post onset; Supplementary Fig. 1). Healthy controls (HCs) with negative SARS-CoV-2 serology ($n = 45$ (25 M and 20 F); 40 (19–73) yr) and historical HC samples that had been stored before November 2019 ($n = 28$ (14 M and 14 F); 62 (22–80) yr) were used as a reference in severity group analyses (Fig. 1a–c). The patients in groups C–E were older than those in the A, B and HC groups, and more frequently men.

To characterize immunological recovery from COVID-19, changes in the absolute number of isolated peripheral blood mononuclear cell (PBMC) subpopulations were assessed for groups A–E relative to the HCs for each time window, with age and sex correction (Fig. 1d and Supplementary Fig. 2). The early T and B cell lymphopenia detected in groups C–E at day 0–14 post onset was resolved by day 15–30 in group C but delayed in groups D and E (Fig. 1d). Absolute numbers of mucosal-associated invariant T (MAIT), Vγ9Vδ2$^{hi}$ T and plasmacytoid dendritic (pDCs) cells in groups D and E remained low beyond day 90

(Fig. 1e and Supplementary Fig. 2). High CD27$^{+}$CD38$^{hi}$ plasmablast counts were detected up to and beyond day 90 in all groups (Fig. 1d). Elevated counts of central memory CD45RA$^{lo}$CCR7$^{+}$CD8$^{+}$ T cells (CD8$^{+}$ T$_{CM}$ cells) were most notable in groups A and B, persisting to day 180 and day 90, respectively (Fig. 1d). An increased ratio of activated/naive CD4$^{+}$ T cells and CD8$^{+}$ T cells remained pronounced in group E until day 360 (Fig. 1f). Collectively, longitudinal immune-cell profiling indicated prolonged immunological disruption (most pronounced in groups C–E) following moderate–severe COVID-19.

### Effect of inflammatory anemia on stress erythropoiesis post COVID-19

Inflammation and disrupted iron homeostasis occurs in hospitalized patients with COVID-19 (refs. [21–24]). The levels of C-reactive protein (CRP) as well as cytokines such as interleukin (IL)-6, IL-10, IL-1ß and tumor-necrosis factor (TNF)-α, which were increased at day 0–14 in the serum of group C–E compared with HCs, resolved slowly over a period of months to a year, with elevated serum concentrations of some inflammatory cytokines, most markedly IL-6, still detectable at day 271–360 post onset (Fig. 2a and Supplementary Fig. 3). The iron-regulating hormone hepcidin (induced by IL-6)[25] blocks the release of iron from cells, particularly from erythrophagocytic macrophages, through direct binding and degradation of the cellular iron exporter ferroportin[26]. Hepcidin was elevated in the serum of groups C–E at day 0–14 compared with the HCs (Fig. 2a), and elevated serum concentrations of the iron storage protein ferritin were seen up to days 30, 90 and 180 for groups C, D and E, respectively (Fig. 2a), suggesting ongoing inflammation and increased cellular iron retention. In contrast, the levels of iron, the iron transport protein transferrin and transferrin iron saturation (TSAT; the ratio of serum iron to the blood's total iron binding capacity) were markedly reduced in the serum of groups C–E at day 0–14 compared with HCs, and serum iron and TSAT remained significantly lower in group E at day 181–270 post onset (Fig. 2a,b and Supplementary Fig. 3). There was little evidence of systemic inflammation or associated disruptions to the iron levels of groups A and B (Fig. 2a).

Low iron in combination with increased ferritin and hepcidin in the serum is a characteristic of inflammatory anemia[27,28], which is associated with dysregulated iron trafficking and disrupted erythropoiesis in the context of systemic inflammation. Groups C and D showed declining concentrations of hemoglobin in the blood relative to HCs for the first 30 days post onset, with hemoglobin levels continuing to decline to day 30–90 for group E (Fig. 2a,c). In addition, the serum levels of the erythropoiesis stimulating hormone erythropoietin (EPO), which is induced by low blood oxygen levels but suppressed by inflammation[29], showed a delayed increase from HC levels, with concentrations peaking at day 15–30 in group E and day 31–90 in groups C and D (Fig. 2a and Supplementary Fig. 3). The four patients in group E who died between day 91 and 180 had the lowest hemoglobin concentrations of all 24 patients profiled at this time (Fig. 2c; mean = 81.5 g l$^{-1}$), which suggests an association between unresolved anemia and COVID-19 severity. Collectively, iron starvation of erythroid cells following inflammatory iron sequestration may compromise the homeostatic response to anemia in moderate–severe COVID-19.

To investigate the effect of altered iron availability on long-term erythropoiesis, we assessed several hematological parameters in groups A–E across days 0–14, 15–30, 31–90 and 90–180 post onset. Reticulocyte counts, which reflect the production of new erythrocytes, were low across all groups (significantly lower for groups B, D and E) at day 0–14 compared with HCs (Fig. 2a,d). A subsequent steep increase in the reticulocyte count and immature reticulocyte fraction (IRF; the fraction of immature reticulocytes in total reticulocytes) resulted in a peak in reticulocyte counts above HC at day 31–90 in groups C, D and E (Fig. 2a,d). Reticulocyte counts in groups A and B resolved HC levels by day 15–30. Total red blood cells remained depleted in group E up to day 91–180 (Fig. 2a,d and Supplementary Fig. 4). The reticulocyte

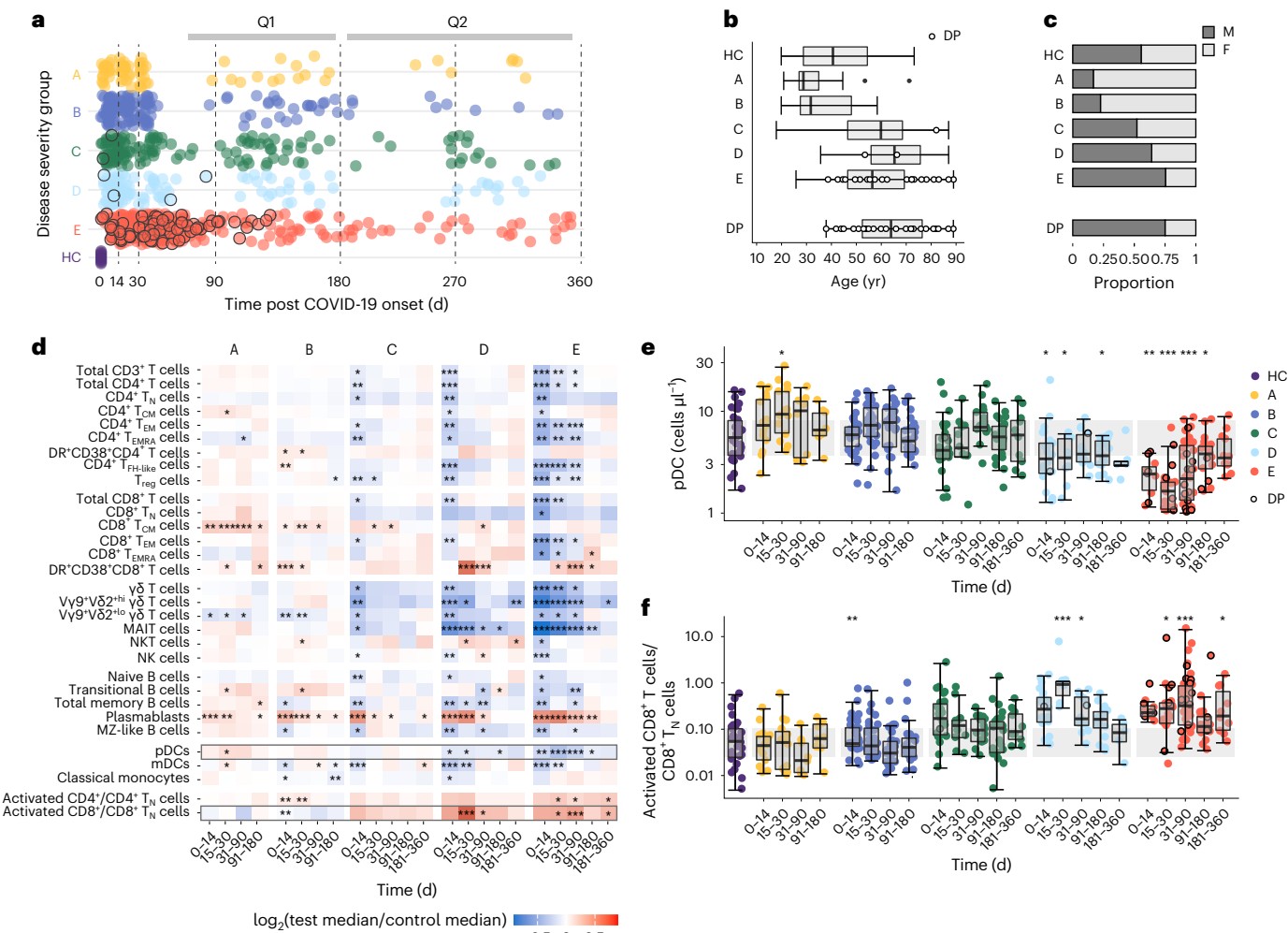

**Fig. 1 | Longitudinal characterization of immunological recovery in COVID-19 severity groups. a**, Distribution of patient sampling across five COVID-19 severity groups over 1 yr post first SARS-CoV-2-positive swab (group A) or symptom onset (groups B–E). The time range of follow-up questionnaire submission (Q1, 3–5 months and Q2, 9–10 months) is indicated (top). **b,c**, Distribution of age (**b**) and sex (**c**) across groups A–E and the HCs defined as in **a**. The demographics for the deceased patients alone are shown (bottom). **d**, Absolute cell count differences (fold change) between the patients in severity groups A–E and HCs during the analyzed time windows. **e,f**, Number of pDCs (**e**) and the ratio of activated/naive CD8+ T cells (**f**) in severity groups A–E and HCs. The gray band represents the interquartile range (IQR) of the HCs; the y axis is shown as a logarithm base ten scale. Box plots show the minimum value, 25th percentile, median, 75th percentile, maximum value and outliers beyond 1.5× the IQR. **d–f**, *P < 0.05, **P < 0.005 and ***P < 0.0005; significance of group effect (per COVID-19 severity group, per time window relative to the HCs) as calculated by linear regression of the log₂-transformed counts (or ratio) with correction for age and sex; no multiple testing correction was applied. DP, deceased patient.

Group A (3 M and 15 F), WHO clinical progression score = 1; group B, mild symptomatic, n = 40 (9 M and 31 F), WHO score = 2–3; group C, moderate without supplemental oxygen requirement, n = 48 (25 M and 23 F), WHO score = 4; group D, moderate with supplemental oxygen given as maximal respiratory support, n = 39 (25 M and 14 F), WHO score = 5); and group E, severe with assisted ventilation, n = 69 (52 M and 17 F), WHO score = 6–10. Repeat samples totaled 73, 148, 132, 114 and 288 across groups A–E, respectively. HCs were sampled at baseline day 0 (n = 60, 34 M and 26 F). Each point represents a time point of blood collection; samples from patients who later died are rimmed in black. Vertical dashed lines, the span of time windows used in all analyses (that is, days 0–14, 15–30, 31–90, 91–180, 181–270 and 271–360 post onset).

---

hemoglobin and mean corpuscular hemoglobin concentration (the average hemoglobin concentration per volume of red blood) remained low in groups C–E at the time of peak reticulocyte production (day 31–90; Fig. 2a,e and Supplementary Fig. 4), suggesting defective stress erythropoiesis that proceeded in the absence of sufficient iron for hemoglobin production.

A gene-set enrichment analysis (GSEA) of genes that were differentially expressed between groups A–E and HCs showed HALLMARK genes linked to heme metabolism as the most strongly upregulated in the whole-blood transcriptome from groups C–E at day 31–90 (GSEA $P < 1 \times 10^{-43}$; Fig. 2f and Extended Data Fig. 1a), consistent with delayed expansion of heme-producing reticulocytes. Reactive oxygen species (ROS), oxidative phosphorylation and hypoxia pathways, among others, were also significantly upregulated at this time (Fig. 2f and

Extended Data Fig. 1a). Genes encoding enzymes involved in heme biosynthesis, such as *ALAS2* and *FECH*, were also significantly overexpressed in groups C–E at day 31–90 compared with HCs (Extended Data Fig. 1b). Genes linked to interferon (IFN), IL-6–JAK–STAT3 and TNF-α signaling were strongly enriched in all groups at day 0–14 relative to HCs (Fig. 2f), indicating that the upregulation of genes linked to heme metabolism occurred later than gene sets capturing the early inflammatory response. To correlate gene-set expression with hematological and immune variables, we used principal component analysis (PCA) of HALLMARK heme metabolism genes to generate a composite 'heme metabolism score' at each RNA-sequencing (RNA-seq) time point (Extended Data Fig. 2a). When assessed over time as a continuous variable, the heme metabolism score increased in early disease, peaked at approximately day 30–50 before declining in groups B–E

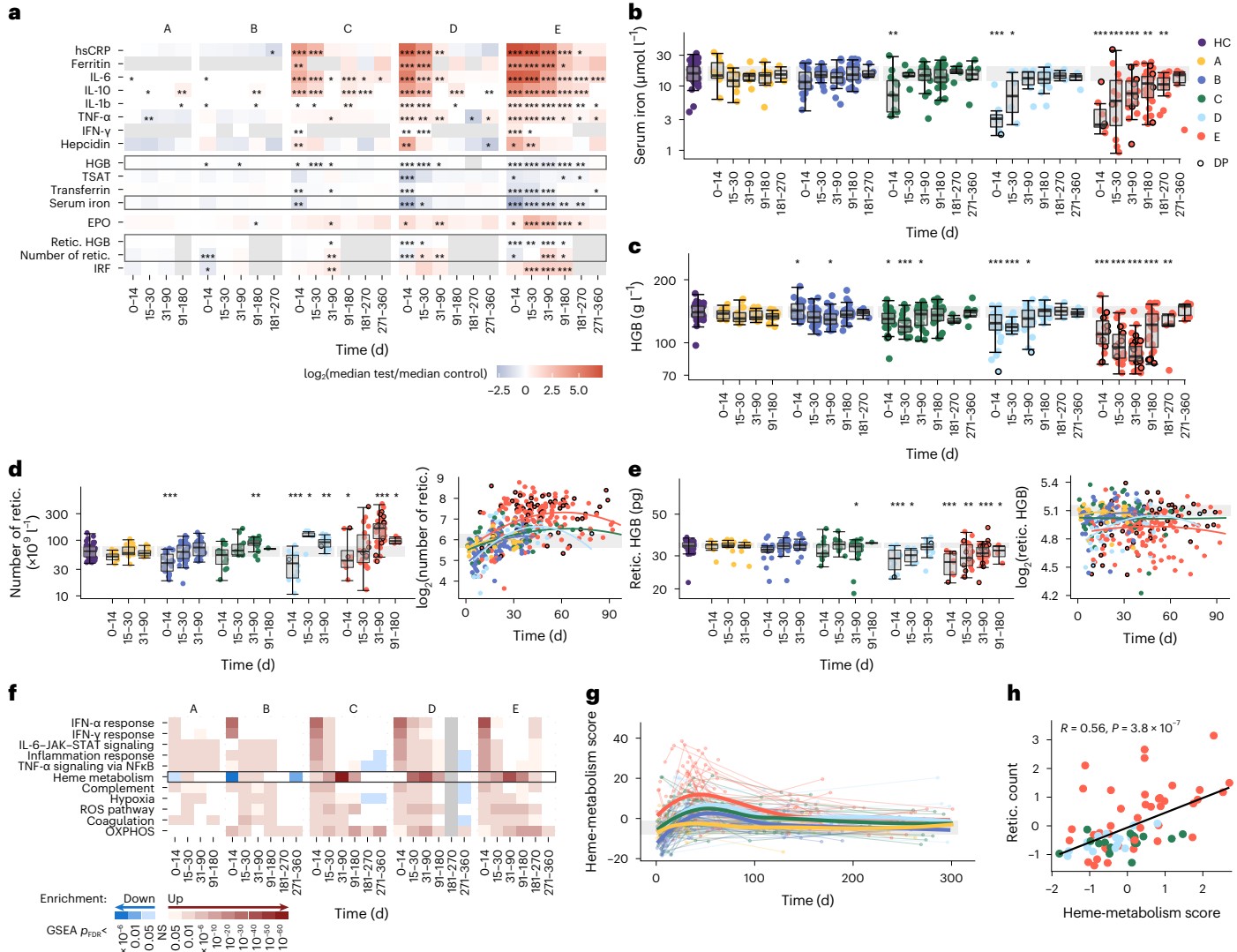

**Fig. 2 | Inflammatory anemia and iron-deprived reticulocyte expansion in patients with moderate–severe COVID-19. a**, Fold change in median serum inflammatory, iron and erythroid cell parameters between patients with COVID-19 in severity groups A–E and HCs or group A and B samples taken at day >90 for ferritin in the absence of HC measures. Fold changes are shown for all time windows. Gray boxes in **a** correspond to the data shown in **b–e**. **b–e**, Serum iron (**b**), hemoglobin (**c**), reticulocyte count (**d**) and reticulocyte hemoglobin (**e**) in patients from groups A–E as in **a**. The gray band represents the IQR of the HCs. Data points from patients who later died are rimmed in black. Box plots show the minimum value, 25th percentile, median, 75th percentile, maximum value and outliers beyond 1.5× the IQR. **a–e**, The significance of group effect (per COVID-19 severity group, per time window relative to HCs) was calculated by linear regression of log$_2$-transformed measures with correction for age and sex; no multiple testing correction was applied. **d**,**e**, Patient-level data plotted against time as a continuous variable (right), with quadratic regression lines fit

for each severity group. **f**, Selection of the top significantly enriched HALLMARK gene sets from GSEA run on the log$_2$-transformed fold change ranked gene lists from comparisons of groups A–E with HCs at each time window. Heat map of false discovery rate (FDR)-adjusted $P$ values ($P_{FDR}$) from the GSEA, with gene sets that were up- or downregulated colored in red and blue, respectively; NS, not significant. **g**, Polynomial splines showing changes in the heme-metabolism score (PC1 from a PCA of heme-metabolism gene-set genes across all sampling time points) over time for groups A–E. The gray band represents the IQR of the HCs. **h**, Correlation between the heme-metabolism score and reticulocyte count of groups C–E (scaled residuals following correction for time post symptom onset) at day 31–90. $R$, Spearman's correlation coefficient. **b–e**,**g**,**h**,The colour key in **b** applies to all panels. hsCRP, high-sensitivity C-reactive protein; OXPHOS, oxidative phosphorylation; HGB, hemoglobin; retic., reticulocyte. *$P$ < 0.05, **$P$ < 0.005 and ***$P$ < 0.0005.

and remained stable in group A (Fig. 2g). After adjustment for the day on which each sample was taken within each time window, the heme metabolism score at day 0–30 and day 31–90 in groups C–E combined was most strongly positively correlated with reticulocyte count (Fig. 2h) and IRF (Extended Data Fig. 2b) measured at the same time point, and negatively correlated with hemoglobin levels (Extended Data Fig. 2b). Together, the pronounced late-stage heme metabolism signature observed at day 31–90 in the whole blood of patients from groups C–E coincided with an increase in iron-deprived reticulocytes following infection.

**Prolonged changes to iron handling in patients with COVID-19**

Iron is essential for cellular respiration and metabolism, yet accumulation of free cytosolic iron catalyzes the production of ROS and contributes to lipid membrane peroxidation and ferroptosis[30,31]. To investigate the consequences of altered cellular iron levels in COVID-19, we investigated the expression of genes from two publicly available gene sets in patient and HC whole-blood transcriptomes over discrete time windows. The first set included 'IRE_HQ' transcripts containing high-quality canonical iron-response elements (IREs) in their 3′ or 5′ untranslated region[32] and the second set included 'iron-homeostasis'

transcripts encoding regulators of iron transport and uptake (such as *TF* and *TFRC*), storage (*FTL*, *FTH1* and *NCOA4*) and antioxidant defense (*GPX4*, *GCLM* and *GCLC*; based on the WikiPathways Ferroptosis gene set[33]; Supplementary Tables 2 and 3). Iron-response proteins IRP1 and IRP2 are post-transcriptional modulators of iron-response genes that regulate cellular iron storage and flux through binding of IREs in target messenger RNA and promoting stabilization or degradation of transcripts[34]. The binding affinity of IRPs for IREs in target genes is dependent on the cellular concentration of iron. IRE_HQ genes showed clear polarization, with genes both significantly up and down regulated in groups C–E at day 0–14 (Fig. 3a and Extended Data Fig. 3a), with most still differentially expressed in group E at day 30–90 (Supplementary Fig. 5), consistent with a response to altered cellular iron concentration[35]. The IRE-containing genes *FTL1* and *FTH1* (encoding ferritin), *SCL4OA1* (encoding the iron exporter ferroportin) and *EPAS1* (encoding the hypoxia-inducible factor HIF-2α) were among those that were upregulated early in hospitalized patients (Fig. 3a and Supplementary Fig. 5), consistent with known responses to high intracellular iron[35]. As IRP1 and IRP2 regulate IRE-containing genes at the post-transcriptional level[35], we validated the observed changes in mRNA expression in PBMCs from 21 day 0–14 COVID-19 samples using mass spectrometry (groups A + B, $n = 7$; group C, $n = 5$; groups D + E, $n = 9$)[36]. Protein mass spectrometry analysis indicated bidirectional regulation of proteins encoded by IRE-containing genes, which was most distinct in groups D and E combined (Extended Data Fig. 3b,c), suggesting that the differential regulation of iron-response genes (probably mediated by IRPs) was detectable at both the transcript and protein level in moderate–severe COVID-19.

Overexpression of iron-homeostasis genes relative to HCs was observed up to day 90–180 in group C, day 30–90 in group D and day 180–270 in group E (Fig. 3b and Extended Data Fig. 3d). The upregulated genes reflected cellular responses consistent with both iron overload and iron deprivation, probably capturing signatures from different blood-cell subsets. Overexpression of genes encoding for constituents of the glutathione peroxidase (GPX4) pathway—including *SLC7A11*, *SLC3A2*, *GCLC*, *GCLM* and *GPX4* (Fig. 3b,c and Extended Data Fig. 3e), which are involved in defenses against ROS-mediated lipid peroxidation—reflected iron overload. In addition, *NFE2L2* (encoding the transcription factor NRF2, a regulator of antioxidant responses) was overexpressed up to day 30–90 in group E (Extended Data Fig. 3f). Changes consistent with cellular iron deprivation were also detectable. Groups C–E showed significant upregulation of *TFRC* (encoding the receptor for transferrin-bound iron) and *NCOA4*, which is involved in ferritin degradation and release of iron stores during instances of increased iron demand (Extended Data Fig. 3e) The fold change in expression of iron-homeostasis genes at day 0–14 in group E relative to HCs was projected onto the relevant KEGG hsa04216 pathway (Fig. 3c). We used a composite 'iron-homeostasis score' to assess changes in iron-homeostasis gene expression across groups A–E compared with HCs over time as a continuous variable (Extended Data Fig. 2a). Prolonged shifts in the transcriptional response to cellular iron levels and demand were observed in groups C–E, with a peak at day 0–14, but remained detectable for months following symptom onset (Fig. 3d). Iron-homeostasis scores correlated strongly with inflammatory parameters, including CRP and IL-6, and inversely correlated with serum iron across corresponding time windows in groups C–E combined (Fig. 3e and Extended Data Fig. 2b). Collectively, whole-blood transcriptional analysis identified gene expression signatures consistent with cellular responses to altered iron status that were slow to resolve following moderate–severe COVID-19.

## Cellular deconvolution of iron signatures with multimodal single-cell data

To elucidate the cell-type origin of the whole-blood transcriptional signatures, we assessed the expression of the iron-homeostasis and IRE gene sets, alongside the HALLMARK heme-metabolism signature reflecting reticulocytosis in moderate-severe COVID-19, in published PBMC cellular indexing of transcriptomes and epitopes by sequencing (CITE-seq) data[37] from a subset of patients infected with SARS-CoV-2 ($n = 36$, groups A–E combined, median 11 days post-onset) as well as from HCs ($n = 11$; Fig. 4a). We used an additional gene set derived from the transcriptome profiling of CaCo-2 cell lines (an in vitro model of intestinal absorptive cells) cultured in iron-free media[38] as a signature for iron starvation (Supplementary Table 4). Three distinct patterns of cell type-specific gene expression were identified (Fig. 4b). First, the heme metabolism signature was nearly exclusively derived from a small cluster of CD71[hi] reticulocytes (Fig. 4b and Extended Data Fig. 4a,b), consistent with the expected tight coupling of this signature to reticulocyte counts. Second, iron homeostasis pathway genes and the IRE-containing genes upregulated in group E were preferentially expressed in myeloid-derived cells (Fig. 4b and Extended Data Fig. 4a,b). These signatures had the highest relative expression in cell clusters annotated as nonclassical CD16+ monocytes, and several classical CD14+ monocyte subsets and dendritic cells (Fig. 4b). Finally, the low-iron signature, which reflected iron starvation in vitro[38], was preferentially expressed in proliferating CD4+ and CD8+ lymphocytes (Fig. 4b), suggesting increased iron demand in proliferating lymphocytes around day 11 post onset, a time point coinciding with limited serum iron availability.

The preferential expression of iron-homeostasis genes in monocytes is consistent with the known erythrophagocytic and iron-acquiring capabilities of these cells[39]. The size of CD16+ and CD14+ monocyte clusters were smaller in COVID-19 than in HC samples (Fig. 4c), consistent with iron scavenging and trafficking to tissues. A repeat analysis of an independent, previously published single-cell COVID-19 dataset[40] indicated similar preferential expression of iron-homeostasis genes in monocyte clusters and reduced frequency of CD14+ classical and CD16+ nonclassical monocytes in COVID-19 samples compared with HCs (Extended Data Fig. 4c). To evaluate the iron demand of various immune-cell subsets during an active viral infection, we analyzed the differential correlation between the expression of cell-surface markers and that of the transferrin receptor CD71 in patients with COVID-19 compared with HCs. CD71 expression was more tightly correlated with markers of innate immune cells (LILRB1, CD64, CD1d and CD1c) and markers of activation (SLC3A2, CD86 and ICAM-1) in the COVID-19 group than in HCs (Fig. 4d), suggesting an increased demand for iron in concert with the activation of innate immune cells during a viral infection. CD71 expression was also elevated on CD16+ and CD14+ monocytes of patients with COVID-19 compared with HCs (Fig. 4e). Thus, a multimodal single-cell analysis identified the cells contributing to signatures of defective iron homeostasis in the blood of patients with COVID-19 and suggested that iron sequestration in monocytes might contribute to the concurrent iron deprivation of proliferating CD4+ and CD8+ lymphocytes.

## Early inflammatory iron dysregulation persists in PASC

To assess the outcome of prolonged iron dysregulation and disrupted erythropoiesis following SARS-CoV-2 infection on PASC, 102 patients in groups B–E (B, $n = 27$; C, $n = 37$; D, $n = 24$; and E, $n = 14$) completed follow-up questionnaires 3–5 months (questionnaire 1, Q1) and 9–10 months (questionnaire 2, Q2) post onset. The severity of seven persisting or new-onset symptoms were scored from zero (worst symptom severity) to five (no symptoms or full recovery; Supplementary Note and Methods). Hierarchical clustering of scores allowed the classification of patients experiencing persisting symptoms (PS) or no PS (NPS) at Q1 and Q2 (Fig. 5a and Extended Data Fig. 5a,b). Persisting symptoms were more frequent in groups C–E than group B, and of those reporting at Q1 and Q2, 65% had PS at both time points (Extended Data Fig. 5c,d). There were no differences in sex or measured early viral titers between the symptom groups (Extended Data Fig. 5e,f).

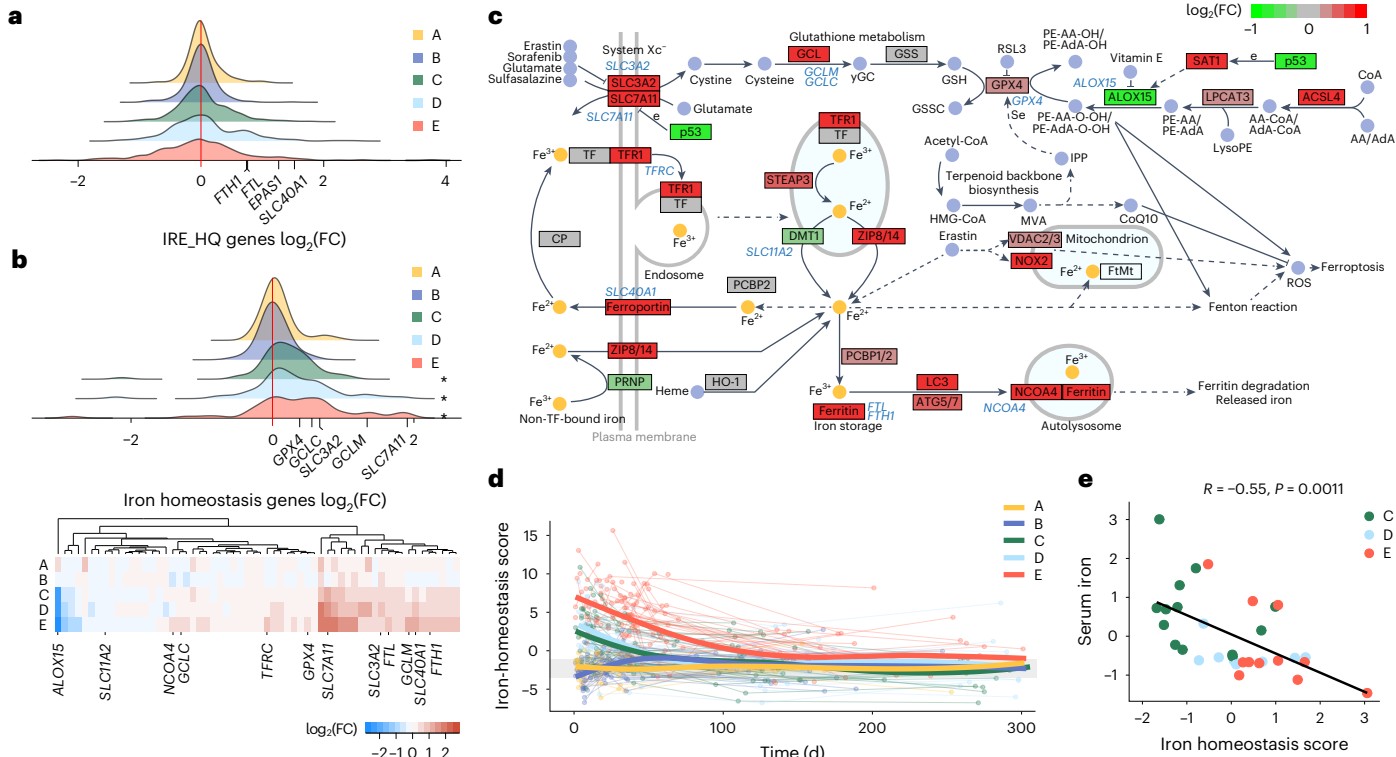

**Fig. 3 | Transcriptional changes to iron-homeostasis pathway genes in hospitalized patients with COVID-19. a**, Distribution of the log₂-transformed fold change (FC) values across 324 measured genes with high-quality conserved IREs in their 3′ or 5′ untranslated region derived from the whole-blood transcriptome comparison of COVID-19 severity groups A–E at day 0–14 and HCs. Four genes of interest are annotated. **b**, Distribution of the log₂(FC) across 60 measured genes in the iron-homeostasis gene set at day 0–14 (top) and heat map of gene-level detail for groups A–E versus HCs at day 0–14 (bottom). *$P < 0.05$,

$P_{FDR}$ values from GSEA. **c**, Schematic of iron-homeostasis pathway (KEGG has04216) with genes colored according to the log₂(FC) in group E at day 0–14. Genes corresponding to those shown in the heat map in **b** are annotated in blue text. **d**, Polynomial splines showing change in iron-homeostasis scores (PC1 from PCA of iron-homeostasis gene-set genes across all sampling time points for groups A–E. The gray band represents the IQR of the HCs. **e**, Spearman correlation between iron-homeostasis score and serum iron in groups C–E (scaled residuals following correction for time) at day 0–14, with points colored by severity group.

The patients with PS were older than those with NPS; however, age did not differ between the PS and NPS groups when the patients were stratified by initial disease severity (Extended Data Fig. 5g and Supplementary Fig. 6), suggesting that age was indirectly associated with PASC only via an association with acute disease severity.

To identify biological variables that could discriminate PS and NPS groups at Q1, we conducted a partial least-squares discriminant analysis (PLS-DA) using previously analyzed immune-cell counts as well as serum and reticulocyte parameters collected within the sequential time windows. The PLS-DA discriminated PS versus NPS better at day 15–30 than during early disease (day 0–14) or at the time of Q1 responses (day 91–180; Fig. 5b and Supplementary Fig. 7). Variable selection identified 15 parameters measured at day 15–30 that predicted PS or NPS designation at Q1 with 72% accuracy (classification error rate, 28%; standard deviation, 2.5%; Fig. 5c). Among these 15 variables, the mean CRP, IL-6, hepcidin and plasmablast counts were higher in PS, and the mean serum iron, transferrin and various immune-cell populations (including CD4⁺ T, CD8⁺ T, NK, regulatory T ($T_{reg}$) and dendritic cells) were lower in the PS group than in NPS at Q1 (Fig. 5c). Unsupervised hierarchical clustering of data from 42 patients (12 PS and 30 NPS) at day 15–30 using only these 15 variables identified a subcluster of 13 individuals, nine of whom (group E, $n = 5$; group D, $n = 2$; group B, $n = 1$; and group C, $n = 1$) were classified as PS at Q1 (Fig. 5d). This analysis suggested that a multivariate signature detectable at day 15–30 could discriminate the patients that experienced PASC at month 3–5 following SARS-CoV-2 infection, independent of hospitalization or oxygen therapy criteria.

Using a multivariate linear regression with age correction to test for an association between PASC symptom groups (NPS or PS) and

biological measures, individuals in the Q1 PS group had significantly lower TSAT and serum iron compared with the NPS group at day 15–30 (Fig. 5e). Reticulocyte counts were elevated in both the PS and NPS groups compared with HCs at day 31–90 (Supplementary Fig. 8), corresponding with the peak of the stress erythropoietic response, although they were significantly higher in PS compared with NPS (Fig. 5e). This suggested that low iron availability, rather than delayed reticulocyte expansion, was a characteristic of stress erythropoietic responses in the PS group. Both CRP and IL-6 were elevated in the PS group compared with NPS at Q1 (day 91–180; Fig. 5e). To further test that the observed differences in biological parameters between PS and NPS were not accounted for by the difference in age between the two groups, we performed pairwise symptom group comparisons in subsets of age-matched patients with COVID-19. This analysis indicated that serum iron, TSAT and hemoglobin levels were significantly lower at day 14–30, reticulocyte counts were significantly higher at day 30–90, and IL-6 and CRP were significantly elevated at day 90–180 in age-matched patients who reported PS at Q1 compared with the NPS group (Extended Data Fig. 6a,b).

Severe COVID-19 and hospitalization has been linked with worse long-term outcome[8,41]. Because PASC was strongly associated with acute disease severity in our cohort, we repeated linear regression analyses including a correction for acute disease severity and following the exclusion of patients from group B to compare hospitalized PS and NPS groups matched for age, sex and severity. Both severity-corrected and severity-matched analyses indicated that serum iron and TSAT were significantly lower at day 14–30 in the PS group compared with NPS (Extended Data Fig. 6c,d). The severity-matched analyses also

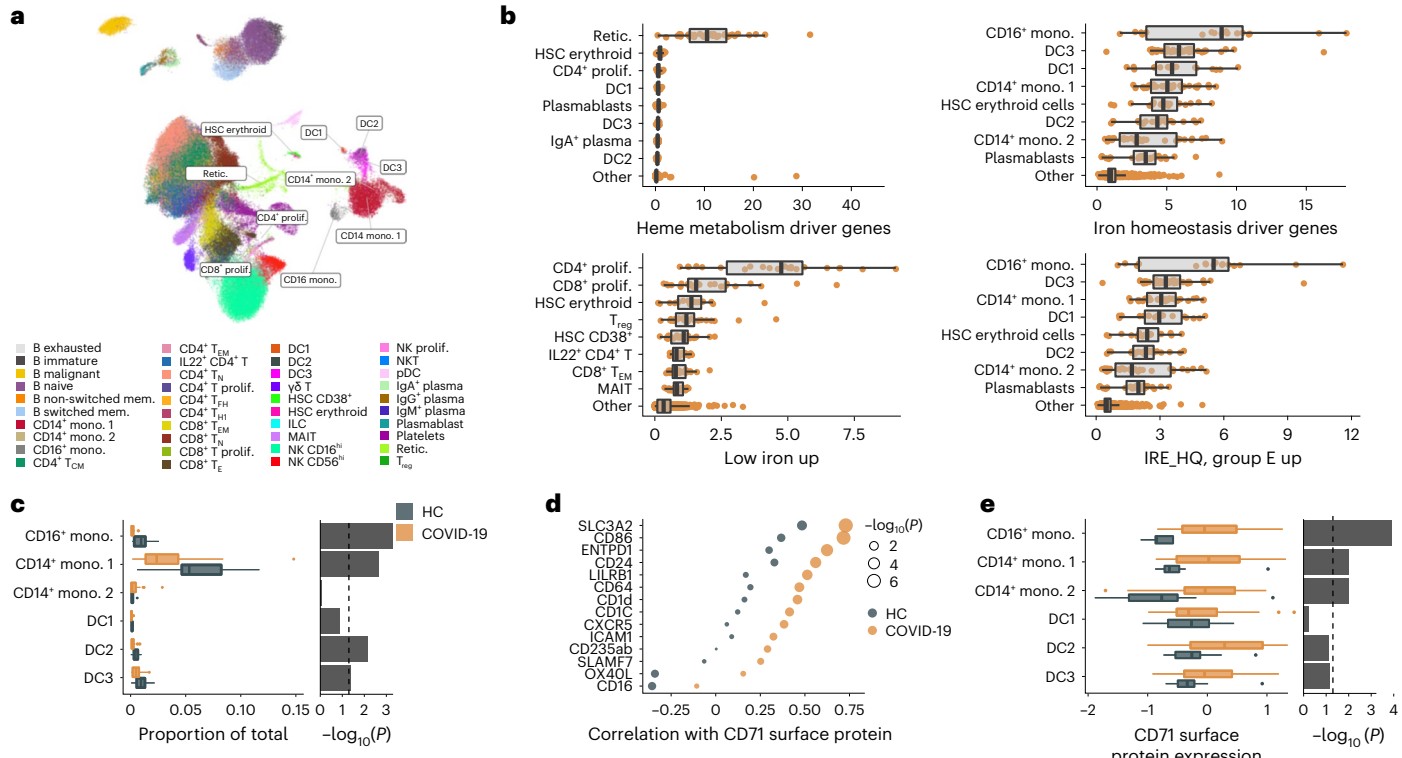

**Fig. 4 | Multimodal single-cell analysis of iron-related signatures. a**, Uniform manifold approximation and projection (UMAP) of CITE-seq data from 36 patients with COVID-19 and 11 HCs, with cells labeled based on previously published cell-type annotations[37]. The UMAP was generated using mRNA expression data and is shown for visualization of cell clusters only. ILC, innate lymphoid cell. **b**, Average expression of heme metabolism and iron-related signature genes aggregated at the sample level within each cell cluster (COVID-19 and HC samples were combined). Cell types with the highest 80th percentile of average signature expression relative to other cell types, across individuals, are shown, with all other clusters merged into the population 'other'. **c**, Comparison of cell frequencies of myeloid populations as a fraction of the total sequenced cells per individual in patients with COVID-19 (groups A–E combined) and HCs

(left). **d**, Differences in the Spearman correlation of normalized CD71 protein expression, across cell clusters, with the surface proteins shown, in patients with COVID-19 (groups A–E combined) and HCs. Top proteins with the greatest difference in correlation (>0.23) are shown. **e**, Differences in normalized CD71 expression within subsets of HCs and patients with COVID-19, with data analyzed at the sample level, aggregated within each cluster per individual (left). **c,d**, Comparison of the COVID-19 and HC samples using −log₁₀-transformed *P* values from a two-sided Wilcoxon rank test (right). **b,c,e**, Box plots show the minimum value, 25th percentile, median, 75th percentile, maximum value and outliers beyond 1.5× the IQR. DC, dendritic cell; mono., monocyte; mem., memory; prolif., proliferating; retic., reticulocyte.

indicated that CRP and IL-6 were significantly elevated in PS compared with NPS at day 90–180 (Extended Data Fig. 6c–f). Although we did not have the statistical power to detect differences in symptom groups within peak disease severity groups (groups B–E), patients in group E with NPS trended toward more rapid recovery of low serum iron and resolution of systemic inflammation than patients in group E with PS (Supplementary Fig. 9). In addition, two of the three individuals with PS in group B had higher CRP at day 15–30 than the 20 individuals in group B with NPS (Supplementary Fig. 9). Together, these findings suggested that disruptions to iron handling that persisted beyond day 0–14, rather than the need for hospitalization or oxygen therapy, was linked to the risk of developing PASC months following acute disease.

Differential expression analysis using whole-blood transcriptomes identified 64 genes that were differentially expressed between the Q1 PS and NPS groups at day 15–30 post onset; these included *EPOR* (encoding the EPO receptor) and *EPAS1* (HIF-2α), which were significantly upregulated in PS (Fig. 5f). These genes are tightly regulated in response to low oxygen carriage, such as in anemia-induced hypoxia[42]. A GSEA analysis indicated that differences in gene expression across biological pathways were greater between the PS and NPS groups at day 15–30 than day 0–14. Individuals in the PS group had pronounced upregulation of heme metabolism and hypoxia pathways as well as ROS, IL-6–JAK–STAT3 signaling and iron homeostasis, among others, at this time (Fig. 5g and Supplementary Fig. 10). Genes linked to IFN

signaling were downregulated in the PS group compared with NPS at day 15–30 (Fig. 5g), suggesting a more transient early IFN response, as previously associated with severe disease[43,44]. Collectively, serum and transcriptional profiles from day 15–30 samples from patients with COVID-19 showed that persisting low iron in serum and delayed resolution of inflammation beyond 2 weeks following SARS-CoV-2 infection differentiated those with PS from those with NPS at 3–5 months, independent of the age and acute disease severity of the patients.

## Discussion

SARS-CoV-2 pathogenesis is well documented but the etiology of PASC remains unclear. Here we show that inflammation and disrupted iron homeostasis persisting beyond 2 weeks post COVID-19 onset best differentiated patients reporting PASC months later. We suggest unresolved inflammation affects long-term pathophysiology through disruptions to cellular iron mobilization and defective, iron-starved stress erythropoiesis that fails to correct the pronounced inflammatory anemia of early disease. Iron loading in monocytes and deprivation in lymphocytes was detected by CITE-seq and reflected in whole-blood transcriptional shifts in iron-response gene sets in patients with moderate–severe COVID-19 and in those later reporting PASC. Low iron availability, for erythropoiesis and cellular metabolism more broadly, potentially results in compromised antiviral immunity and low systemic oxygen carriage throughout and beyond acute infection.

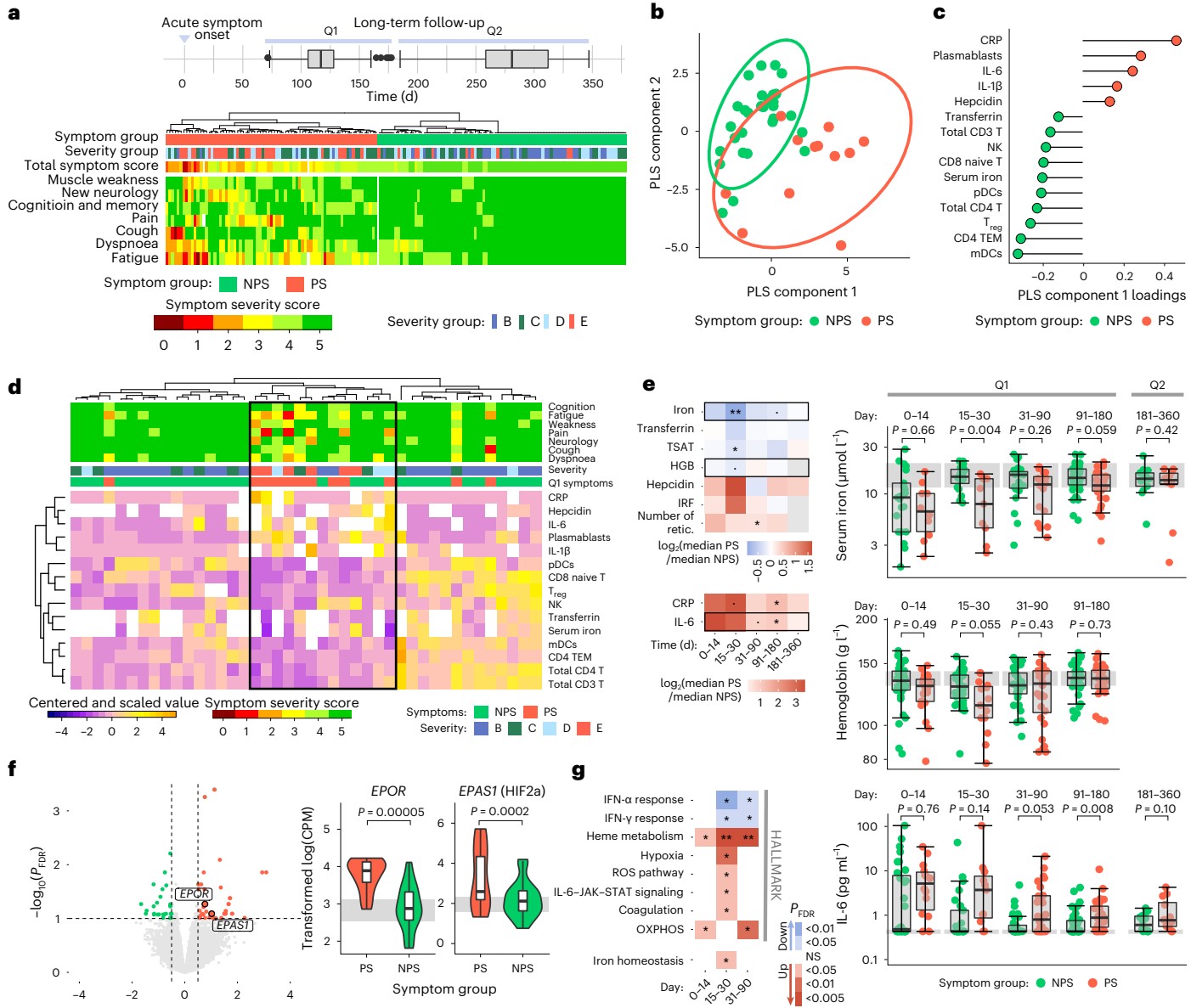

**Fig. 5 | Differences in long-term symptom groups across measured serum, cellular and transcriptional variables. a**, Grouping of patients with PS or NPS from hierarchical clustering of symptom severity scores (0, worst; 5, best) across seven symptom categories. The disease severity group (groups B–E) and total symptom score (summation across symptoms) are indicated above the heat map. The distribution of the responses to the follow-up questionnaires at Q1 and Q2 is shown (top). **b**, PLS-DA analysis of symptom groups from a study conducted on immune-cell counts, serum parameters and reticulocyte data collected between days 15 and 30. **c**, Variables driving differentiation of individuals with NPS and PS on PLS component 1, colored according to the group with highest mean. **d**, Unsupervised hierarchical clustering of patient data from day 15–30, using the 15 leading variables as in **c**. Patient symptom groups, severity groups and symptom severity scores are shown above the heat map. The cluster capturing most PS individuals is outlined by a black box. Missing data are shown in white in the heat map. **e**, Fold change (log₂-transformed) in median serum inflammatory and iron parameters of individuals with PS compared with NPS at different time windows (left). The significance of the symptom group effect was calculated by linear

regression of log₂-transformed measures corrected for age; no multiple testing correction was applied. Patient-level data for the boxed parameters in more detail (right). The gray band represents the IQR of the HCs; the y axis is shown as a logarithm base ten scale. Measures taken at days 0–180 and 181–360 are annotated on the basis of the Q1 and Q2 symptom groups, respectively. **f**, Volcano plot showing genes that are differentially expressed, from differential gene expression analysis with age correction, between the PS (red) and NPS (green) groups at day 15–30 (left). Normalized expression for *EPOR* and *EPAS1* (right); *P* values are from differential gene expression analysis before FDR correction. The gray band indicates the IQR of HC expression. CPM, counts per million reads. **g**, Significantly enriched HALLMARK and iron-homeostasis gene sets from GSEA run on the log₂(FC) ranked gene list from a comparison of NPS and PS groups across time windows. $P_{FDR}$ values from GSEA are shown, with up- and downregulated gene sets in PS colored red and blue, respectively. **a,e,f**, Box plots show the minimum value, 25th percentile, median, 75th percentile, maximum value and outliers beyond 1.5× the IQR. *P < 0.1, *P < 0.05, **P < 0.005 and NS, not significant; mDCs, myeloid DCs.

These abnormalities may help drive PASC and thus inform strategies for prevention or treatment of this complex phenomenon.

Hospitalized patients with COVID-19 often develop inflammatory anemia[21,22,45], a common feature of chronic inflammatory

conditions[28,46,47]. During inflammation, IL-6 stimulates the production of the hormone hepcidin by hepatocytes[25,48], which induces the degradation of ferroprotein, the only known cellular iron exporter[26]. Reduced iron export drives iron accumulation in macrophages, which

would otherwise recirculate iron liberated from phagocytosed senescent erythrocytes[49]. Sequestration of iron during infection helps defend against extracellular pathogens dependent on iron for survival[50] but also starves the erythroid compartment of iron for hemoglobin production, causing anemia[27,28]. We observed reduced serum iron, TSAT and hemoglobin concentrations as well as raised ferritin, hepcidin and IL-6 in COVID-19 severity groups C–E from day 0–14 post symptom onset, indicating inflammatory anemia in moderate–severe disease.

Oxygen transport requires $O_2$ coupling to the iron-containing heme molecules of hemoglobin, so modulation of blood oxygen levels necessitates control of iron availability. During hypoxia, the transcriptional regulator HIF-2α accelerates erythropoiesis via EPO[42,51]. Inflammation and low iron availability antagonize this process by suppressing EPO expression[29,52–54]. Despite experiencing hypoxia that warranted oxygen therapy, patients in groups D and E as well in group C, who did not receive oxygen supplementation, exhibited reduced reticulocyte production and delayed EPO induction in early disease. Following this, and consistent with a stress response to low blood oxygen levels, patients in groups C–E exhibited marked reticulocyte expansion, which peaked at 1–3 months post onset and was reflected in the overexpression of a heme metabolism signature in blood. In our cohort this phenotype was not only seen in severely ill, ventilated patients (group E) but also in hospitalized patients requiring only moderate or no oxygen therapy, and was pronounced in those subsequently reporting PASC. Stress erythropoiesis has been described in anemic mice[55,56] but is less well defined in humans[57]. Iron-starved reticulocytosis probably represents an inadequate physiological response to concurrent hypoxia, inflammatory iron restriction and anemia in moderate–severe COVID-19.

Iron availability is essential for cellular metabolism and regulates the function and proliferative capacity of leukocytes[58–60]. However, iron overload increases susceptibility to ROS-induced ferroptotic cell death[30,61]. Consistent with hepcidin-mediated iron redistribution, we observed transcriptional signatures of iron accumulation in circulating CD16+ classical and CD14+ nonclassical monocytes, potentially predisposing them to cellular dysfunction through ROS-mediated damage and contributing to tissue and organ pathology in patients with COVID-19. Iron-laden macrophages are detectable in post-mortem bone marrow samples of individuals following fatal COVID-19 (ref. [62]), and ferroptosis in the ventricular myocardium or liver may cause end-organ damage and fatal disease[63–65].

In contrast to signatures of high intracellular iron in monocytes, we saw evidence of iron starvation and increased CD71 surface expression in activated and proliferating leukocytes. Low iron availability compromises T cell effector function and humoral immunity[58–60], NK cell activation[66] and neutrophil antimicrobial activity[59], while hypoferremia during vaccination reduces central memory T cell responses and antigen-specific recall in mice[60]. Low serum iron at symptom presentation, coinciding with the induction of adaptive immunity, may impede the generation of SARS-CoV-2 cellular and humoral memory responses in patients with COVID-19. Even in normal iron conditions, hypoxia disrupts humoral immunity in mice by reducing B cell numbers and affinity maturation, defects similar to those observed in severe COVID-19 (ref. [67]). Iron dysregulation and hypoxia may sustain a destructive cycle of impaired immune function, poor viral control and inflammation that contributes to tissue-specific and systemic manifestations of severe acute COVID-19, and potential disruption of long-term immune memory.

Many features of PASC may be driven, at least in part, by the impact of inflammatory iron dysregulation on erythropoiesis and blood oxygen carriage. We found that delayed resolution of inflammation and associated hypoferremia, rather than the magnitude of inflammatory perturbations during acute disease, best discriminated patients reporting persisting symptoms months post infection. Fatigue, pain and mood disorders have been linked to inflammatory anemia in chronic inflammatory conditions[68,69] and are common features of PASC[2,3,5]. Reduced oxygen delivery to muscles during exertion increases reliance on anaerobic glycolysis, elevating lactate production and leading to muscle fatigue and pain[70]. Low iron availability also impairs mitochondrial energy generation in skeletal muscle, decreasing physical endurance[71]. Iron deficiency and cerebral hypoxia have been linked to cognitive impairment and altered mood, and iron deficiency during childhood is a significant risk factor for poor cognitive performance[72–74]. Finally, low oxygen carriage may exacerbate tissue hypoxia and delay repair, and persisting iron dysregulation and anemia have been associated with more severe structural lung abnormalities following COVID-19 (ref. [75]). Speculatively, the generally increased prevalence of iron deficiency in pre-menopausal women may contribute to the higher risk of PASC amongst this demographic[7,9,10] by enhancing the relative magnitude of infection-related iron redistribution against a baseline of lower iron stores.

Worse acute COVID-19 severity is a risk factor for PASC, and severe COVID-19 is predominantly seen in older males. Restricted access to uninfected population controls during the early pandemic resulted in suboptimal age and sex matching of HCs (recruited from healthcare workers) to patients with moderate–severe COVID-19 in this study, leading to differences in the demographic of PS and NPS groups. Although it is probable that acute and long-term symptom severity are to an extent causally linked, careful re-analysis of PASC symptom groups with age, sex and acute disease severity matching indicated that iron dysregulation at day 15–30 and raised inflammatory markers (IL-6 and CRP) at day 90–180 in the PS group were independent of these variables. Several clinical strategies may help mitigate the impact of early iron dysregulation on both acute COVID-19 severity and PASC. Vaccination, or selective antiviral or monoclonal therapy, may prevent sustained disruptions to iron homeostasis driven by severe uncontrolled inflammation. In those with worse disease, treatments directed at correcting abnormal iron distribution might also be considered. Reports that iron overload in the context of β-thalassemia protects from severe disease and mortality in individuals infected with SARS-CoV-2 (ref. [76]) suggest a potential protective effect of increased iron availability, and preliminary reports on the impact of COVID-19 on patients enrolled in the IRONMAN clinical trial of intravenous ferric derisomaltose treatment for heart failure[77] show significantly reduced COVID-19-related severe adverse events in the iron-treated group (2.1%) than the usual care group (5.3%, $P = 0.007$)[78]. This suggests a potential role for iron supplementation in COVID-19. Remobilization of endogenous iron stores can also increase iron availability. This may be achieved either directly through the use of hepcidin inhibitors[79], which have shown efficacy in reversing inflammation-induced hypoferremia[80], or through IL-6 inhibition. The IL-6R blocker tocilizumab, which reduces hepcidin generation, increases hemoglobin levels in patients with rheumatoid arthritis[81], corrects inflammatory anemia in Castleman disease (associated with excessive IL-6 production)[82] and has been trialed as an anti-inflammatory agent in patients with COVID-19 (ref. [83]). Thus, several potential therapies might be trialed to see if they reduce the incidence of PASC in patients with moderate–severe COVID-19.

It is unlikely that these observations are SARS-CoV-2 specific. Disruption of host iron homeostasis is a consequence of many viral infections, both through direct viral mechanisms of interference and as a consequence of the evoked inflammatory response[50,84]. Many infectious diseases—including Ebola[85,86], influenza[87] and SARS[88]—elicit broadly similar post-acute sequelae, suggesting similar iron-redistribution strategies may be considered. This study has implicated disrupted iron homeostasis and iron-deprived stress erythropoiesis that persisted for more than 2 weeks from symptom onset as potential drivers of PASC. If confirmed, this immediately suggests several strategies that could be explored to prevent it.

## Online content

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

[1]Cambridge Institute of Therapeutic Immunology and Infectious Disease, Jeffrey Cheah Biomedical Centre, University of Cambridge, Cambridge, UK. [2]Department of Medicine, University of Cambridge, Addenbrooke's Hospital, Cambridge, UK. [3]NIH–Oxford–Cambridge Scholars Program, Department of Medicine, University of Cambridge, Cambridge, UK. [4]MRC Biostatistics Unit, University of Cambridge, Cambridge Biomedical Campus, Cambridge, UK. [5]British Heart Foundation Centre of Research Excellence, University of Cambridge, Cambridge, UK. [6]NIHR BioResource, Cambridge University Hospitals NHS Foundation, Cambridge Biomedical Campus, Cambridge, UK. [7]Department of Haematology, Wellcome and MRC Cambridge Stem Cell Institute, University of Cambridge, Cambridge, UK. [8]Department of Biomedicine, University and University Hospital Basel, Basel, Switzerland. [9]Botnar Research Centre for Child Health (BRCCH), University of Basel and ETH Zurich, Basel, Switzerland. [10]NHS Blood and Transplant, Cambridge Biomedical Campus, Cambridge, UK. [11]Department of Haematology, University of Cambridge, Cambridge, UK. [12]MRC Translational Immune Discovery Unit, MRC Weatherall Institute of Molecular Medicine, University of Oxford, Oxford, UK. [13]Present address: The Walter and Eliza Hall Institute of Medical Research, Parkville, Victoria, Australia. [14]Present address: University of Melbourne, Melbourne, Victoria, Australia. ✉e-mail: smith.k@wehi.edu.au

## Cambridge Institute of Therapeutic Immunology and Infectious Disease–National Institute for Health Research (CITIID–NIHR) COVID BioResource Collaboration

Hélène Ruffieux[4], Federica Mescia[1,2], Laura Bergamaschi[1,2], Lorinda Turner[1,2], Prasanti Kotagiri[1,2], Christoph Hess[1,2,6,7], Nicholas Gleadall[8,9], John R. Bradley[2,10,11], Paul A. Lyons[1,2] & Kenneth G. C. Smith[1,2,13,14]

A full list of members appears in the Supplementary Information.

## Methods

### Cohort recruitment and biological sample collection

Study ethics approval was obtained from the East of England–Cambridge Central Research Ethics Committee ('NIHR BioResource' REC ref. 17/EE/0025 and 'Genetic variation AND Altered Leucocyte Function in health and disease–GANDALF' REC ref. 08/H0308/176). All participants provided informed consent. We have previously published detailed information on the recruitment, sampling time line, clinical characteristics and demographics of 104 patients admitted to Addenbrooke's Hospital, Royal Papworth Hospital (RPH) NHS Foundation Trust or Cambridge and Peterborough Foundation Trust who tested SARS-CoV-2 positive and 97 asymptomatic or symptomatic healthcare workers attending the Addenbrooke's serology screening program between March and July 2020[20]. An additional 13 patients hospitalized with COVID-19 were recruited following discharge and provided blood samples for cellular and serum assays as well as RNA-seq from 130 days post symptom onset. Individuals who were PCR-positive for SARS-CoV-2 were classified into five groups based on peak disease severity: asymptomatic (group A; $n = 18$, 3 M and 15 F; WHO clinical progression score = 1; age (median (range)), 28 (20–71) yr), mild symptomatic (group B; $n = 40$ (9 M and 31 F); WHO score = 2–3; age, 31 (19–58) yr), moderate without supplemental oxygen requirement (group C; $n = 48$ (25 M and 23 F); WHO score = 4; age, 59.5 (17–87) yr), moderate with supplemental oxygen given as maximal respiratory support using low-flow nasal prongs, face mask, Venturi mask or nonrebreather face mask (group D; $n = 39$ (25 M and 14 F); WHO score = 5; age, 65 (35–87) yr) and severe with requirement for noninvasive ventilation, mechanical ventilation or extracorporeal membrane oxygenation (group E; $n = 69$ (52 M and 17 F); WHO score = 6–10; age, 56 (25–89) yr). The mean time in hospital for patients in groups C, D and E with known date of discharge was 4.7 (IQR, 1–7), 11.2 (IQR, 6–16) and 52.6 d (IQR, 26.5–61.2), respectively. Twenty-four hospitalized patients (group E, $n = 21$; group D, $n = 2$ and group C, $n = 1$) died over the course of the study period. An additional 45 HCs with confirmed negative SARS-CoV-2 serology (25 M and 20 F; age, 40 (19–73) yr) were used as reference in all clinical assays and statistical analyses; 28 additional historical healthy control samples stored previous to November 2019 (14 M and 14 F; age, 62 (22–80) yr) were included in the RNA-seq analyses (total $n = 60$, 26 M and 45 F); age, 50 (19–80) yr). No statistical methods were used to pre-determine sample sizes, and recruitment was based on access to and availability of participants during national lockdown. Demographics of all COVID-19 patients and HC and the baseline clinical features of patients in groups A–E and HCs are included in Supplementary Table 1.

Participant recall and sampling, beyond the study period extensively described previously[20], occurred at approximately 3, 6 and 12 months following recruitment. At each time point, blood samples were drawn in sodium citrate, serum and PAXgene blood RNA tubes (BD Biosciences) and processed by members of the CITIID–NIHR COVID BioResource Collaboration, as previously described[20]. Serum aliquots were taken from approximately 9 ml of blood, spun at 800g for 10 min. Peripheral blood mononuclear cells were isolated from approximately 27 ml blood collected in 10% sodium citrate tubes using Leucosep tubes (Greiner Bio-One) with Histopaque 1077 (Sigma) by centrifugation at 800g for 15 min. The PBMC interface was collected, rinsed twice with autoMACS running buffer (Miltenyi Biotech) and cryopreserved in FBS with 10% dimethylsulfoxide previous to cell staining for immunophenotyping. The PAXgene blood RNA tubes were kept at room temperature for 2 h and then stored at −80 °C before RNA extraction. Patient sampling time lines were aligned by days post COVID-19 symptom onset, or the first positive swab in the case of asymptomatic participants, for all downstream analyses. To remove the possibility of confounding effects due to vaccination on late parameters, final time-point data from 25 participants who had received an mRNA-based COVID-19 vaccine before final sampling (within day 180–360 post onset) and a further

ten for whom vaccine status could not be ascertained following the initiation of UK vaccine regimens in December 2020 were excluded.

### Clinical data collection

Laboratory test results for hospital blood screening assays conducted during the preliminary study period (including serum cytokine, CRP, hemoglobin and ferritin concentrations), and repeated on recall samples and HCs, were extracted from Epic electronic health records (Addenbrooke's Hospital) and MetaVision ICU (RPH). High-sensitivity CRP, hemoglobin and ferritin levels were measured by the NIHR Cambridge Biomedical Research Centre Core Biochemical Assay Laboratory using standard assays. The levels of the serum cytokines IL-6, IL-10, IL-1β, TNF-α and IFN-γ were measured using a High-sensitivity base kit HS cytokine A mag (product code LHSCM000, Bio-Techne R&D Systems) on a Luminex analyzer (Bio-Plex) by the Clinical Immunology Laboratory at Addenbrooke's Hospital.

### Follow-up questionnaire and long-term symptom groups

A follow-up questionnaire for the assessment of long-term outcomes following COVID-19 was based on a published tool developed by Cambridge University Hospitals for assessing rehabilitation need in patients with COVID-19 who had prolonged intensive-care-unit stays[89]. The original tool (the post-ICU presentation screen, PICUPS) was developed by a cross-disciplinary group of experienced clinicians and pilot tested in 26 hospitals across England. The modified questionnaire (Supplementary Note 1) was administered to patients in groups B–E 3–5 months (Q1; $n = 107$; mean 116 d post onset; IQR, 103–127 d post onset) and 9–10 months (Q2; $n = 59$; mean 287 d post onset; IQR, 260–320 d post onset; Supplementary Fig. 1), and assessed a range of long-term self-reported outcomes. The participants were asked to report only on symptoms arising or worsening in severity following SARS-CoV-2 exposure. Only responses to questions requiring symptom severity scoring on a numerical scale from zero (worst) to five (best), as opposed to yes–no or descriptive responses, were included in the symptom group classification. Participant scores across seven symptom categories (fatigue, dyspnea, cough, pain, cognition and memory, new neurology and muscle weakness) were used to classify individuals into PS or NPS symptom groups at each questionnaire time point, as detailed below.

Questionnaire responses across both Q1 and Q2 were clustered using hclust in R. Two distinct clusters of questionnaire responses were defined using cutree, clearly distinguishing participants reporting PS or NPS at long-term follow-up. Biological samples collected before day 180 were analyzed based on the symptom group derived from the Q1 responses, those collected after day 180 were analyzed based on the symptom group derived from the Q2 responses. After exclusion of data from individuals for whom reporting took place after SARS-CoV-2 vaccination (or with unknown vaccination status), $n = 97$ Q1 responses (26, 37, 22 and 12 from groups B, C, D and E, respectively) and $n = 26$ Q2 responses (1, 9, 7 and 9 from groups B, C, D and E, respectively) were linked to previous sampling time points for downstream analysis.

### Serum iron, hepcidin and EPO assays

Quantification of serum iron, total iron binding capacity (TIBC), transferrin, hepcidin and EPO was conducted by the NIHR Cambridge Biomedical Centre Core Biochemistry Assay Laboratory at Addenbrooke's Hospital. Serum iron was measured using the Siemens Healthineers Dimension EXL iron assay (product code DF85) through absorbance-based detection of ferrous iron–Ferene complexes. Transferrin levels were quantified using a Siemens Healthineers Dimension EXL transferrin assay (product code DF103), a turbidimetric assay involving the formation of immune complexes between transferrin and antitransferrin. A Siemens Healthineers Dimension EXL IBCT assay (product code DF84) was used to determine TIBC. This is a colorimetric assay involving the addition of excess iron to saturate transferrin-iron

binding sites, with excess unbound iron incorporated into ferrous iron–Ferene complexes and photometrically quantified as described above. Acidification of the reaction releases transferrin-bound iron for further incorporation into ferrous iron–Ferene complexes, resulting in increased absorbance in proportion to the concentration of transferrin-bound iron and thus TIBC. All assays were automated on a Siemens Dimension EXL analyzer. Transferrin saturation was calculated as: TSAT = (serum iron / TIBC) × 100.

Serum hepcidin levels were measured using a Bio-Techne R&D Systems human hepcidin Quantikine ELISA kit (product code DHP250), a quantitative sandwich ELISA method using an antihuman hepcidin monoclonal capture antibody and detection antibody, the latter conjugated to horseradish peroxidase. The enzyme oxidizes an added chromogen for photometric detection of a colored complex. Serum EPO was measured in a similar fashion using a Bio-Techne R&D Systems human erythropoietin Quantikine IVD ELISA kit (product code DEP00); the analyte was captured using monoclonal mouse anti-EPO and detected using a polyclonal rabbit anti-EPO conjugated to horseradish peroxidase.

### Reticulocyte counts

Reticulocyte counts, IRF fractions, reticulocyte hemoglobin content and mean corpuscular hemoglobin concentrations were measured on blood samples collected between day 0 and 180 using a Sysmex XN-1000 hematology analyzer as per the manufacturer's instructions.

### Flow immunophenotyping

Flow immunophenotyping was performed with five florescent antibody panels, staining approximately $1 \times 10^6$ PBMCs, using a five-laser BD Symphony X-50 flow cytometer. Sample population gating was performed in FlowJo v10.2. The antibody panels and gating schema have been previously described in detail[20]. BD TruCount tubes (product code 340334, BD Biosciences) were used as per the manufacturer's instructions for direct enumeration of T, B and NK cells. Enumerated parent populations were used to calculate the absolute counts (cells ml$^{-1}$) of gated daughter populations.

### Whole-blood RNA-seq

Extraction of whole-blood RNA stored in PAXgene blood RNA tubes (product number 762165, BD Biosciences) was performed using a PAXgene blood RNA kit (product number 762164; PreAnalytiX, Qiagen) as per the manufacturer's protocol. A SMARTer stranded total RNA-seq v2–pico input mammalian kit (product number 634413, Takara) was used as specified by the manufacturer to prepare RNA-seq libraries using 10 ng RNA as the input. The libraries were sequenced using 75-bp paired-end chemistry on a HiSeq4000 instrument (Illumina). The sequencing read quality was assessed using FastQC v0.11.8 (Babraham Bioinformatics), with trimming of SMARTer adapters and poor-quality terminal bases (Phred score threshold < 24) with Trim Galore v0.6.4 (Babraham Bioinformatics). Ribosomal RNA contamination was removed using BBSplit (BBMap v38.67) and clean reads were aligned to the human reference genome GRCh38 using HISAT2 v2.1.0. Alignment.bam files were merged and read count matrix generated using the function featureCounts from the R package Rsubread (v2.0.1). Count data were stored in a DGEList object with accompanying gene annotations and patient metadata for downstream handling (EdgeR v3.28.1). Nineteen samples with fewer than 2,000,000 assigned reads, and one sample with an abnormal read distribution were excluded. Genes with >1 count per million reads in >5% of samples were retained, and genes on the Y chromosome and the X-chromosome inactivation factor *XIST* were excluded, leaving a total of 22,354 genes with expression counts across 610 serial COVID-19 and control whole-blood samples. Normalization for library size was performed using the calcNormFactors function from EdgeR (v3.28.1). The function voom (limma v3.42.2) was applied to the count matrix to estimate the mean-variance relationship,

enabling adjustment for heteroscedasticity. Batch variation identifiable across seven sequential RNA extraction batches was corrected for using the empiricalBayesLM function from the R package WGCNA (v1.69), using transcriptomes from HCs as well as group A and B samples taken beyond day 60 as references. Residual batch variation was corrected for by the inclusion of a batch covariate in the statistical model for differential gene expression analysis as described in the 'Statistical analysis' section. Genes were annotated using the R package AnnotationDbi (v1.48.0).

### Mass spectrometry

Previously published PBMC mass spectrometry data for seven HCs and 21 patients with COVID-19 during early disease (day 0–14), included in a published proteomic analysis of the same disease cohort, were used. All methods are detailed in the associated text[36].

### CITE-seq

The CITE-seq data were downloaded from the public portal https://covid19cellatlas.org/. Cell metadata and raw unique molecular identifier counts for mRNA and antibody-derived tags (surface protein counts) were extracted and analyzed using R as described in the 'Statistical analysis' section. The expression distribution of lineage-defining surface proteins from cell clusters of interest confirmed the validity of subset annotations (Extended Data Fig. 4b).

### Statistical analysis

All statistical analyses were conducted in R v3.6.0 (ref. 90) using custom scripts and publicly available analysis packages. Longitudinal patient data for all biological and clinical measures were analyzed within severity (A–E) or persistent symptom (PS/NPS) groups, within sequential time windows spanning days 0–14, 14–30, 30–90, 90–180, 180–270 and 270–360 post COVID-19 onset. Due to low sample numbers, absolute cell counts derived from samples collected during day 180–360 were analyzed together in one window. Any severity-time window grouping containing two or fewer samples was excluded from analysis. For all severity group analyses, SARS-CoV-2-negative or historically collected HCs (RNA-seq only) were used as the reference comparison group throughout. Measured variables from individuals who went on to report PS were compared with those who went on to report NPS at each sampling time point. Hospital assays for ferritin concentrations were not performed on HC serum, so data collected for group A and B samples taken after day 90 were used as a representative 'healthy' baseline for these measures. For patients with repeated measures collected within a given time window, only the earliest sampling point was retained, except for analyses treating time as a continuous variable. Clinical variables (serum cytokines, inflammatory markers, iron and reticulocyte parameters, and absolute cell counts) were normalized by $\log_2$-transformation and COVID-19 severity group effects were tested using multivariate linear regression with correction for age (treated as a continuous integer value) and sex as covariates. The PASC symptom group effects were tested with age correction only as NPS and PS groups were sex matched at both questionnaire time points.

The validity of using age as a linear covariate for age-bias correction was tested by assessing the nature of age associations with clinical and cellular parameters in HCs and COVID-19 severity group samples taken beyond day 180 (Supplementary Methods). Only 13—including IL-6, IL-10, IRF, CD4 T cells (naive and activated, and naive:activated ratio), CD8 T cells (absolute counts, naive and activated:naive ratio), γδ T cells (total, Vγ9+Vδ2$^{hi}$ and Vγ9+Vδ2$^{lo}$) and MAIT cells—of the 47 measured parameters showed evidence of an association with age. For all these parameters, age effects could be effectively modeled using a linear covariate in HCs and patients with COVID-19. Differences detected between the PS and NPS groups (which also varied in age) were confirmed by a Wilcoxon test in various matched subsamples of the cohort, including age-matched participants, and age and acute disease

severity-matched participants (with exclusion of group B patients, overrepresented in the NPS group, from analyses).

Whole-blood differential gene expression analysis between each severity group–time window comparison and HCs (or symptom group comparison) was performed using the lmFit function from the package limma[91], applied to weighted linear models using voom[92], adjusting for age, sex and RNA extraction batch using the design formula model.matrix(~0 + group + age + sex + extractbatch).

Intra-patient correlations were modeled using duplicateCorrelation to account for repeated sampling, with the patient ID used as a blocking factor. Test statistics for each gene expression comparison were regularized using the empirical Bayes method (eBayes in limma) with $P$ values adjusted for multiple testing using the FDR. Genes with an absolute $\log_2$(fold change) $\geq 0.5$ and $P_{FDR} < 0.1$ were considered significantly differentially expressed. A GSEA analysis was performed using the limma function camera[93] with HALLMARK and WP_FERROPTOSIS (here 'iron homeostasis') gene sets from the MSigDB database[94] or curated gene lists from published transcriptome analyses (IRE_HQ)[32]. Gene-set scores for heme metabolism and iron-homeostasis gene sets were derived using a PCA analysis of gene-set gene expression counts across all sampling time points. PC1 was used to capture variation in gene-set expression across samples as a continuous variable and modeled within each severity group using polynomial splines. Spearman correlations between gene-set scores and biological measures were computed within each analysis time window, using residuals extracted from the regression of each variable with days post onset to first correct for a possible confounding effect of time. The use of raw data or extracted residuals had minimal effect on correlation outcomes.

The CITE-seq UMAP plot shown for visualization purposes was calculated using 50 principal components based on variable mRNA defined using Seurat[95] with the method 'vst' in the FindVariableGenes function. The UMAP R package was used to calculate UMAP embeddings using the hyperparameters $n$ neighbors = 40, spread = 0.5, minimum distance = 0.4 and random state = 42. Raw protein antibody-derived-tag counts were normalized and denoised using the dsb function Model-NegativeADTnorm[96] to remove rescale data to define the background signal of each protein and to remove cell-to-cell technical variations using per cell models and isotype controls. The function arguments were denoise.counts = TRUE, use.isotype.control = TRUE, pseudocount.use = 1 and quantile.clipping = TRUE. The dsb-normalized data were used in downstream analyses. To analyze the relative expression of each signature, raw UMI RNA counts were aggregated using the average expression per individual of each gene, and each gene signature was further aggregated as the mean of the genes in the signature per individual for each cell type. Cell types were defined using the author's published annotations. The 80th percentile of median sample level expression across all cell types are highlighted as defining the main source of the signature, with other cell types merged into the 'other' population. Cell frequencies for each cluster were calculated for each sample as the number of cells divided by the total cells for that individual's sample. Cell frequency comparisons between healthy donors and patients with COVID-19 were tested using a two-sided Wilcoxon rank test. Validation of the relative expression of iron-homeostasis genes across cell types, and monocyte cluster frequencies in the COVID-19 and HC groups, was performed using publicly available CITE-seq data from Schulte-Schrepping and colleagues[40]. The protein correlations across subsets were done on the median marker expression per subset. The subset of proteins with the highest difference in the Spearman's correlation of CD71 with all other proteins in patients with COVID-19 compared with HCs are shown with a cutoff of a difference of 0.23. Analysis of protein correlation with iron homeostasis signatures within cell subsets was done using a linear model. Average dsb-normalized protein expression of each marker in monocytes was associated with the average expression of the iron homeostasis driver gene signature from the earliest time point for the COVID-19 samples.

Protein coefficients were regularized toward the average effect using the eBayes function with limma.

Data collected across cellular, serum and reticulocyte variables, within each time window, were used for supervised PLS-DA analysis of symptom groups (PS versus NPS) using the plsda function from the package mixOmics (v6.10.9)[97]. The function tune.splsda was used to determine the variables that were most informative in symptom group discrimination within the day 14–30 time window, based on 30 permutations of fourfold cross-validation. Fifteen variables selected in ≥85% of permutations were used to cluster Q1 symptom groups based on patient data collected between days 14 and 30.

### Reporting summary

Further information on research design is available in the Nature Portfolio Reporting Summary linked to this article.

### Data availability

All datasets used in the generation of presented figures—including cell counts, serum measures, Sysmex hematology data, PAXGene whole-blood RNA-seq gene expression counts, patient metadata and PASC group assignments—can be downloaded from the Zenodo repository (https://doi.org/10.5281/zenodo.10161238). Whole-blood RNA-seq data are available through the European Genome-Phenome Archive (EGA, ID: EGAS00001005332). CITE-seq processed data are available to download from Array Express using accession number E-MTAB-10026. Published CITE-seq data from Schulte-Schrepping et al.[40] used for replication analysis are available through the EGA (ID: EGAS00001004571). HALLMARK and WP_FERROPTOSIS gene sets are accessible through the MSigDB database (https://www.gsea-msigdb.org/gsea/msigdb/). The IRE gene set IRE_HQ is available as supplement in the article by Hin et al.[32] and the iron-starvation gene set was taken from Table 3 ('genes upregulated in iron-free medium') in the associated publication by Chicault and colleagues[38]. Gene sets are also available in Supplementary Tables 2–4.

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

### Acknowledgements

The members of the Cambridge Institute of Therapeutic Immunology and Infectious Disease–National Institute of Health Research (CITIID–NIHR COVID) BioResource are listed in the Supplementary

Information and include the authors F.M., L.B., J.R.B., P.L. and K.G.C.S. We thank the NIHR BioResource volunteers for their participation, and acknowledge NIHR BioResource centers, NHS Trusts and staff for their contribution. We thank the National Institute for Health and Care Research, NHS Blood and Transplant, and Health Data Research UK as part of the Digital Innovation Hub Programme. The views expressed are those of the author(s) and not necessarily those of the NHS, NIHR or Department of Health and Social Care. We acknowledge the services provided by the University of Cambridge Stratified Medicine Core Laboratory, NIHR Cambridge Biomedical Research Centre (BRC) Core Biochemistry Assay Laboratory, NIHR Cambridge BRC Cell Phenotyping Hub, Cambridge Clinical Immunology Laboratory and University of Cambridge Department of Haematology for sample processing and biological data generation. We thank J. Stevens and L. Stefanucci from the Department of Haematology, University of Cambridge for their involvement in hematology data collection, and M. Potts and M. Weekes from the Department of Medicine, University of Cambridge for providing mass spectrometry data. We thank J. Frost from the Weatherall Institute of Molecular Medicine, University of Oxford MRC for his valuable contribution to the scientific interpretation of findings and manuscript structuring. We thank CVC Capital Partners, the Evelyn Trust (20/75), Addenbrooke's Charitable Trust, Cambridge University Hospitals (12/20A), the NIHR Cambridge Biomedical Research Centre and the UKRI/NIHR through the UK Coronavirus Immunology Consortium (UK-CIC) for their financial support. The BioResource was funded by awards from NIHR to the NIHR BioResource (grant numbers RG94028 and RG85445). K.G.C.S. was funded by a Wellcome Investigator Award (grant number 200871/Z/16/Z) and MRC Programme Grant (grant number MR/W018861/1). A.L.H. was funded by the EU H2020 project SYSCID Grant (grant number 733100).

## Author contributions

Conceptualization: A.L.H., M.P.M., P.A.L., H.D. and K.G.C.S. Methodology: A.L.H., M.P.M., F.M., L.B., V.S.P., L.T., P.K., B.G. and H.D. Formal analysis: A.L.H. and M.P.M. Precursor analysis: L.B., V.S.P., L.T., P.K. and H.R. Data collection and curation: A.L.H., M.P.M., F.M., L.B., V.S.P., L.T., P.K. and N.G. Visualization: A.L.H. and M.P.M. Project administration: F.M. and L.B. Funding acquisition: B.G., J.R.B., P.A.L. and K.G.C.S. Supervision: P.A.L., H.D. and K.G.C.S. Writing (original text): A.L.H., M.P.M., P.A.L., H.D. and K.G.C.S. Writing (editing): A.L.H., M.P.M., H.R., C.H., J.A.N., P.A.L., H.D. and K.G.C.S. All authors contributed to the review of written text.

## Competing interests

The authors declare no conflicts of interest.

## Additional information

**Extended data** is available for this paper at https://doi.org/10.1038/s41590-024-01754-8.

**Correspondence and requests for materials** should be addressed to Kenneth G. C. Smith.

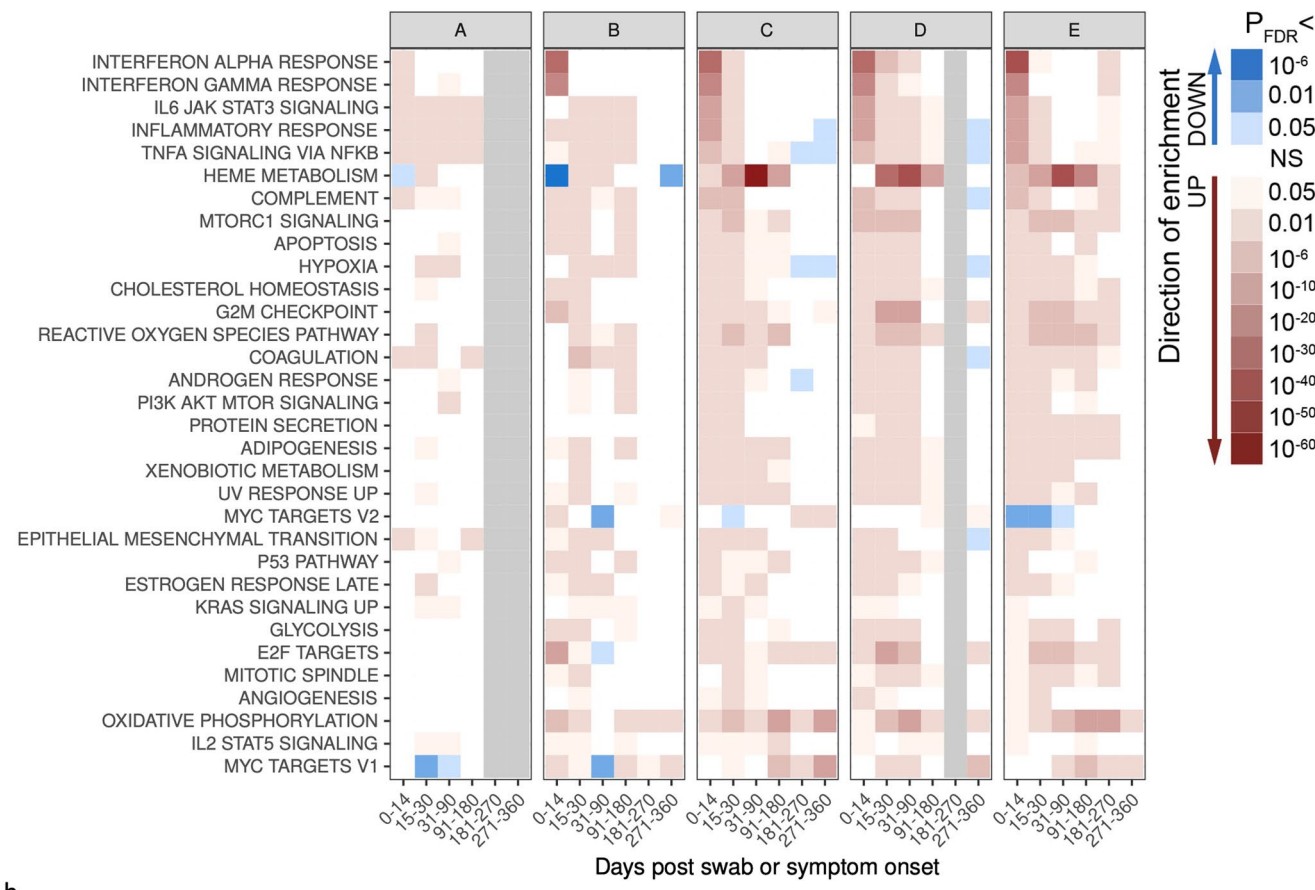

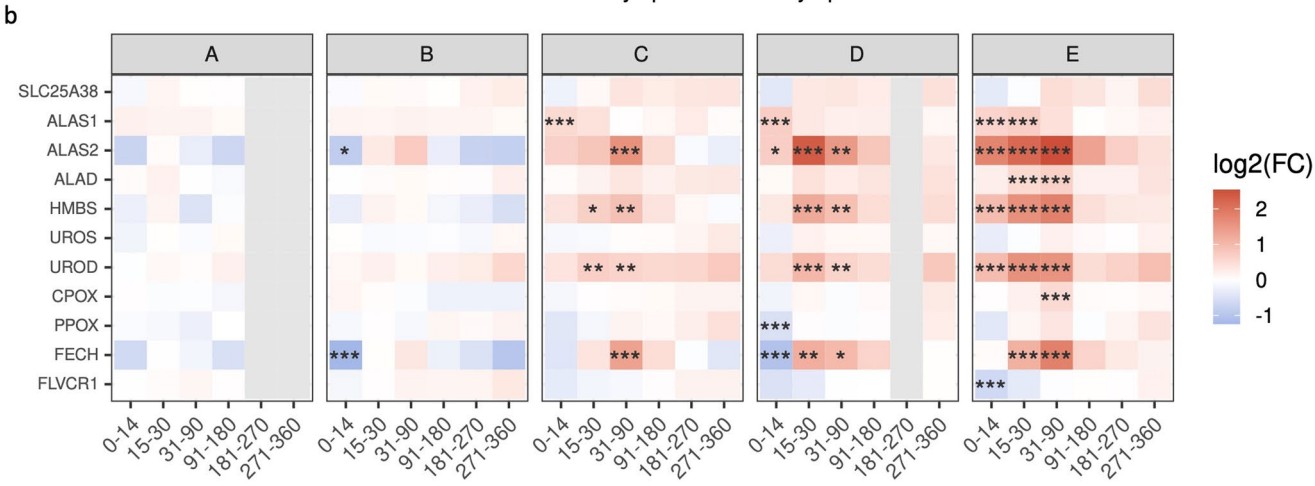

**Extended Data Fig. 1 | Whole blood HALLMARK gene-set enrichment and heme synthesis gene expression changes in COVID-19 severity groups over time. a**, GSEA using MSigDB HALLMARK gene-sets run on the log₂FC ranked gene lists for each COVID-19 severity group (A-E) and time-window comparison with HC. Shade represents FDR adjusted p-value, with gene sets up- or downregulated in COVID-19 colored in red or blue respectively. **b**, Log₂ fold-change of heme synthesis genes as taken from DGE analysis of COVID-19 severity groups and HC with age and sex correction, within time windows from symptom onset. FDR adjusted P-values from linear model fit: *p<0.1, **p<0.05, ***p<0.005.

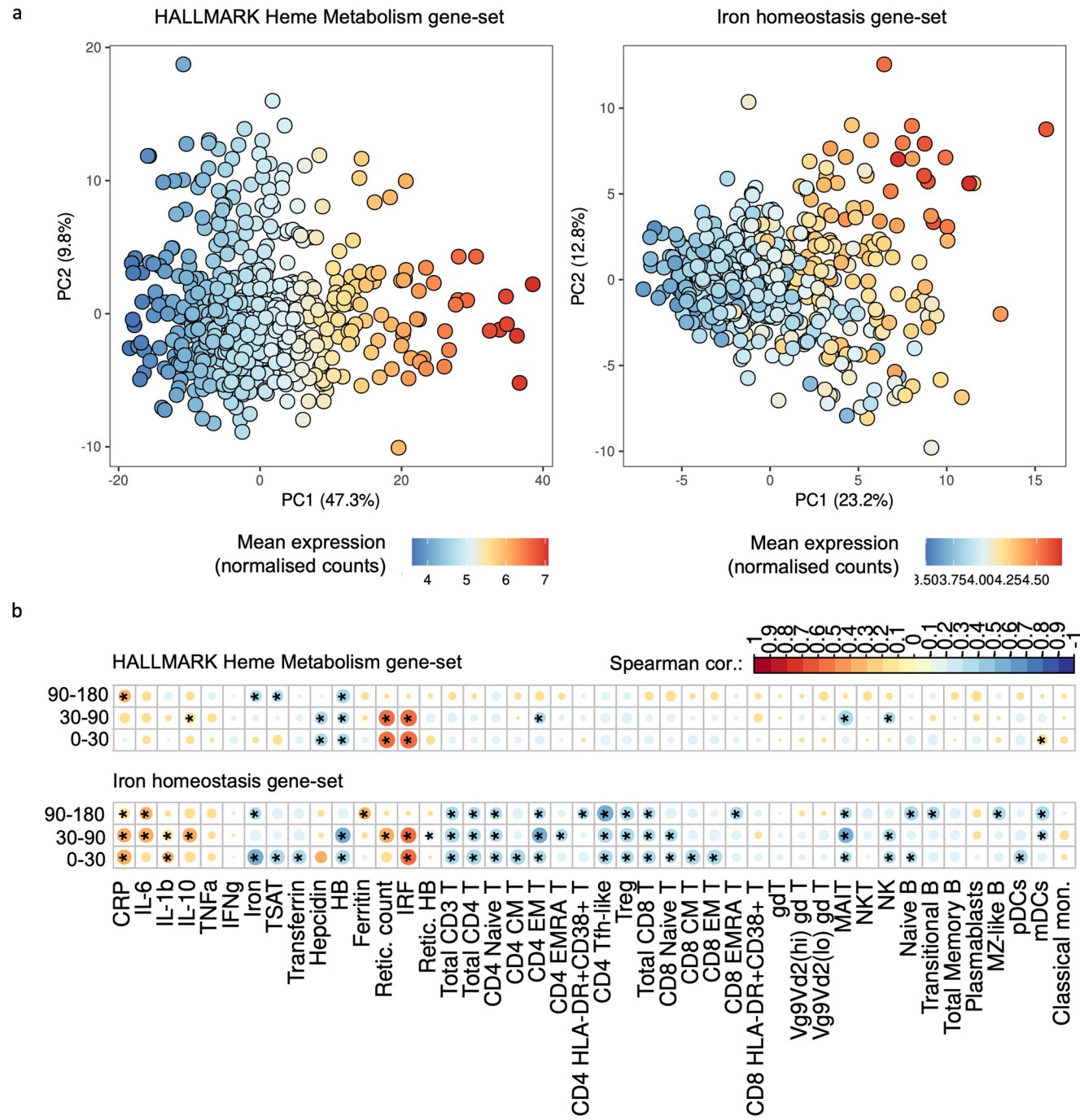

**Extended Data Fig. 2 | Whole blood transcriptional scores correlated with measured serum and cellular parameters in COVID-19 severity groups over time. a**, PCA of COVID-19 patient and HC whole-blood transcriptomes across all sampling timepoints using genes in the HALLMARK heme metabolism gene-set (left), or iron-homeostasis gene-set (right). Points are colored according to mean expression across gene-set genes. PC1 scores are used to capture variation in gene set expression. **b**, Spearman correlation between HALLMARK heme metabolism score (top) or iron homeostasis score (bottom) and other measured biological variables in hospitalized patients (groups C–E combined) within discrete time windows. Variables are corrected for time by extracting residuals from linear regression with days post-onset prior to correlation. Asterisks represent significance at P<0.05 prior to FDR correction, points are colored according to strength of correlation.

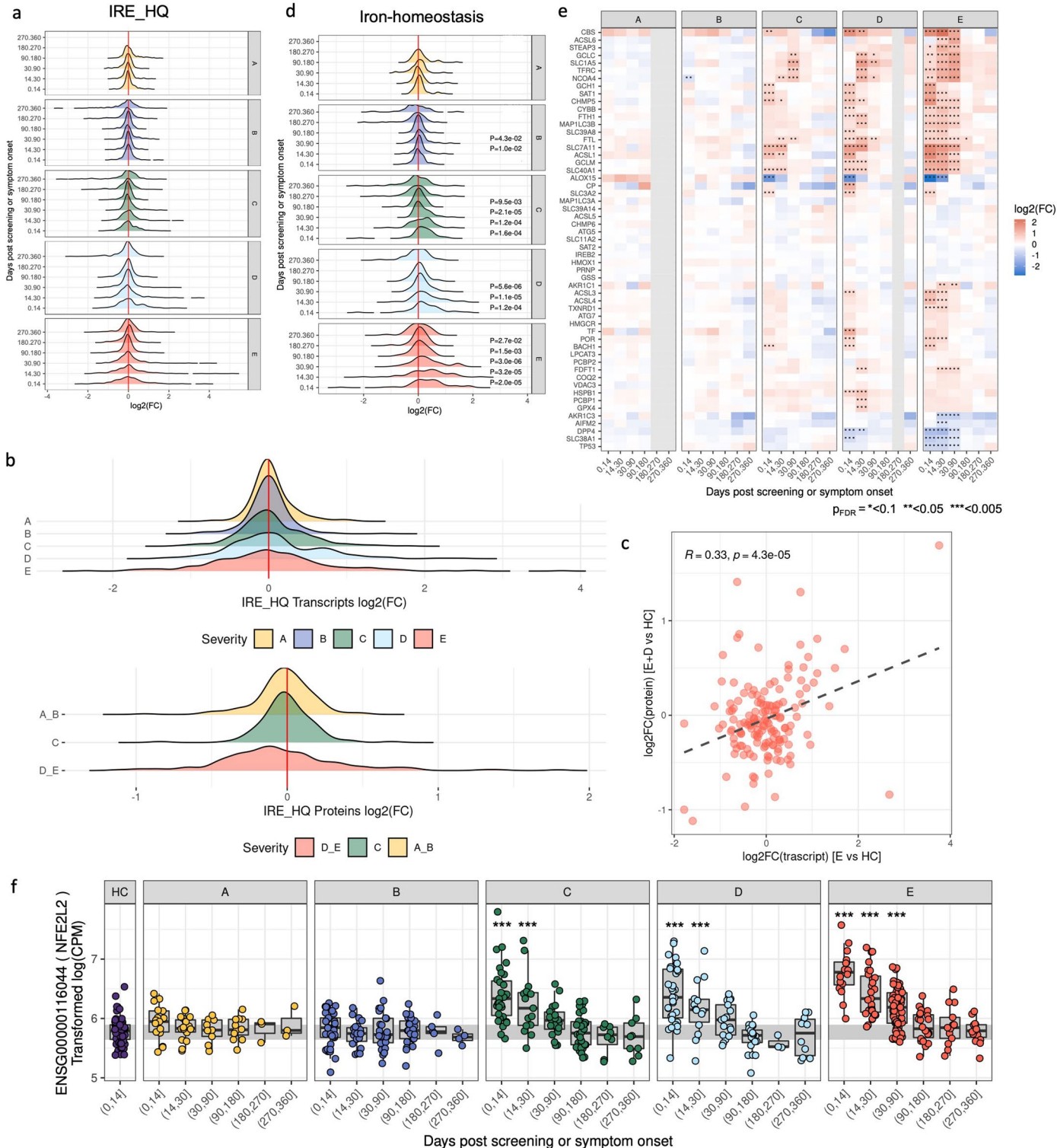

**Extended Data Fig. 3 | Altered expression of iron-response gene-sets in COVID-19 severity groups over time. a**, Distribution of log$_2$FC values across 324 measured genes with high-quality conserved iron-response elements (IRE_HQ) in their 3' or 5' UTR, derived from the whole-blood transcriptome comparison of COVID-19 severity groups (A-E) and HC over successive time windows. **b**, Distribution at day 0-14 from a, compared to corresponding distribution of log$_2$FC values across 150 measured proteins in the IRE_HQ gene set, taken from matched samples within the same time window. Samples from patients in groups D and E, and groups A and B, were analyzed together relative to HC to improve sample sizes (n(HC)=7, n(A_B)=7, n(C)=5, n(D_E)=9). **c**, Correlation of log$_2$FC values from the transcriptional comparison of group E patients and HC, and the protein level comparison of groups D+E and HC, at day 0-14. Pearson's correlation

coefficient and p-value shown. **d**, Distribution of log$_2$FC values across 60 measured iron homeostasis genes, derived from the whole blood transcriptome comparison of COVID-19 severity groups and HC over successive time windows, GSEA p-value from gene-set enrichment analysis shown. **e**, Heat map showing log$_2$FC of each gene in more detail. Significantly differentially expressed genes (P$_{FDR}$<0.1, abs(log$_2$FC) >0.5) are indicated with asterisks: * P$_{FDR}$<0.1, ** P$_{FDR}$<0.05, ***P$_{FDR}$<0.005. **f**, Change in expression of master regulator of the antioxidant response, *NFE2L2* (encoding NRF2), over time and across severity groups. Gray bar indicates IQR of the interquartile range of the HCs. FDR adjusted p-values from linear model fit: *p<0.1, **p<0.05, ***p<0.005. Boxplots show minimum, 25$^{th}$ percentile, median, 75$^{th}$ percentile and maximum, and outliers beyond 1.5 times the interquartile range.

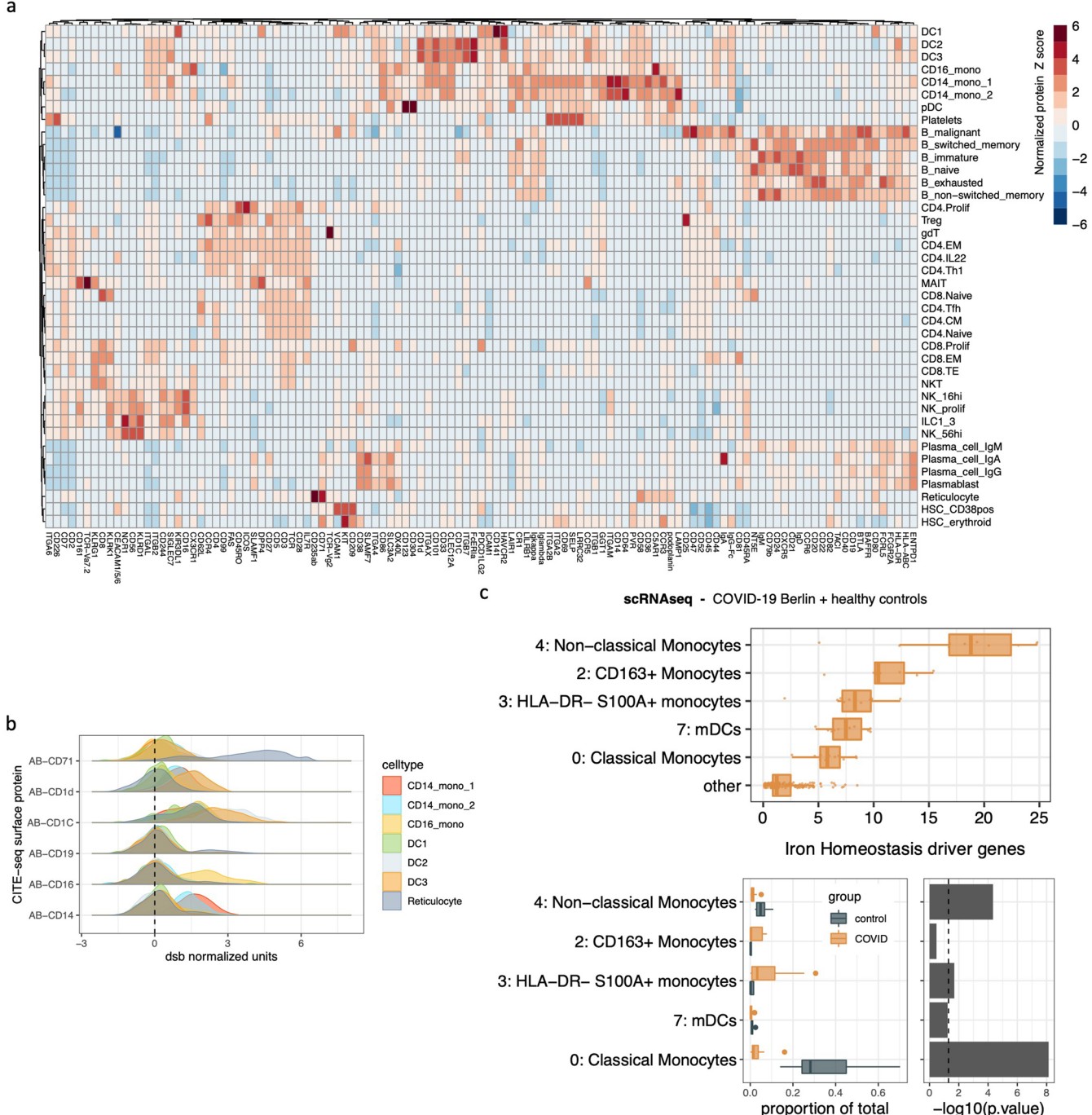

**Extended Data Fig. 4 | Multimodal single cell analysis of iron-related gene signatures. a**, Heatmap of dsb normalized surface protein expression across cell subsets. Values are scaled for each protein across cell clusters. **b**, Distribution of cell surface protein expression in myeloid-derived cell subsets with highest expression of iron-related signatures, and reticulocytes. **c**, Data from Schulte-Schrepping et al. as validation of data shown in Fig. 4b,c. (top) Average expression of iron-homeostasis genes aggregated at the sample level within each cell cluster (COVID-19 patients and HC are combined). Cell types

with the highest 80th percentile of average signature expression relative to other cell types, across individuals, are shown, with all other clusters merged into the population "other". (bottom) Cell frequency of myeloid populations as a fraction of the total sequenced cells per individual, compared between COVID-19 patients (orange) and HC (black). The right margin shows -log10 of the p-value comparing COVID-19 and HC with a two-sided Wilcoxon rank test. Boxplots show minimum, 25th percentile, median, 75th percentile and maximum, and outliers beyond 1.5 times the interquartile range.

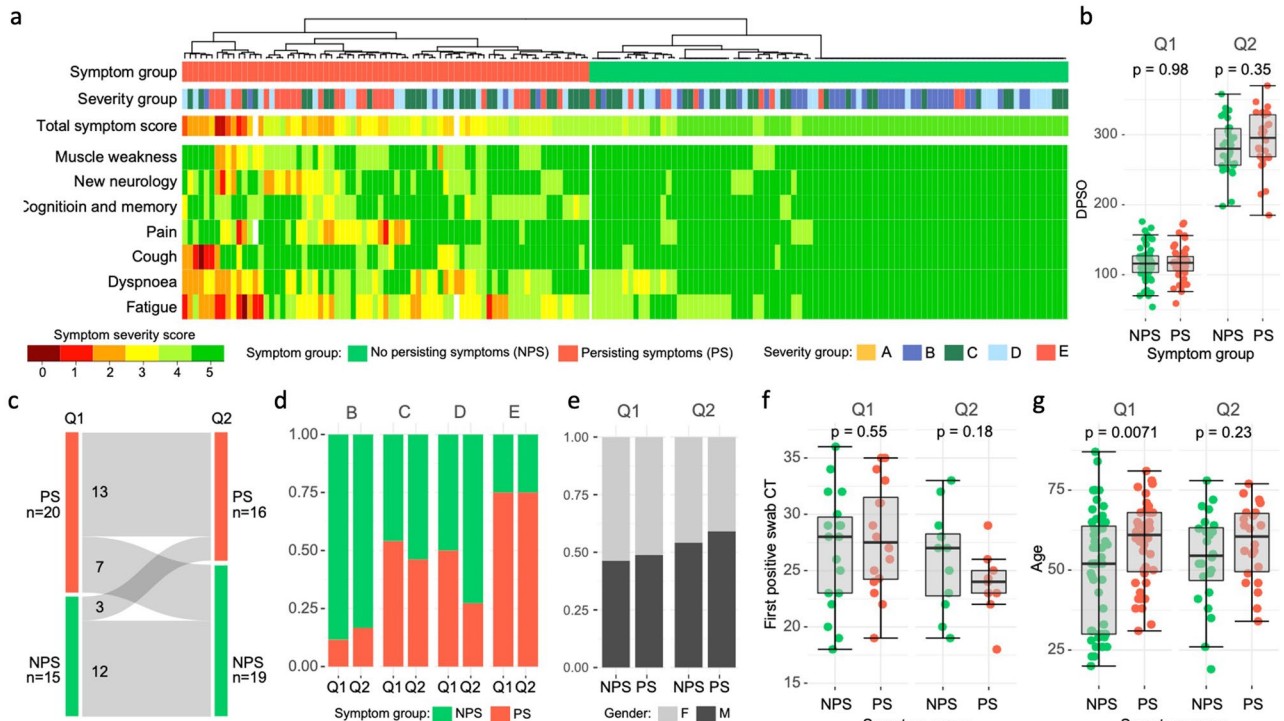

**Extended Data Fig. 5 | Classification and characteristics of PASC persisting symptom (PS) and no persisting symptom (NPS) groups. a**, Grouping of patients with persisting symptoms (PS) or no persisting symptoms (NPS) of COVID-19 from hierarchical clustering of symptom severity scores (0 worst to 5 best) across 7 symptom categories, reported at Q1 (month 3–5 post-onset) and Q2 (month 9–10 post-onset). Disease severity group (B-E) and total symptom score (summation across symptoms) is indicated above heatmap. **b**, Comparison of time from first COVID-19 symptom to follow-up at each questionnaire timepoint for PS and NPS groups. P-value derived from two-sided *t*-test comparison of group means. **c**, Flow of participants between symptom groups at two follow-up timepoints for 35 individuals providing responses for both. **d**, Proportion of NPS and PS individuals within disease severity groups (B-E) at both questionnaire timepoints. **e**, Distribution of sex, **f**, viral titer (as assessed by SARS-CoV-2 positive swab cycle threshold value) and **g**, age between PS and NPS groups at both follow-up time-points. P-values calculated by two-sided *t*-test. Boxplots show minimum, 25th percentile, median, 75th percentile and maximum, and outliers beyond 1.5 times the interquartile range. PS = persisting symptom, NPS = no persisting symptoms, Q1 = questionnaire 1, Q2 = questionnaire 2.

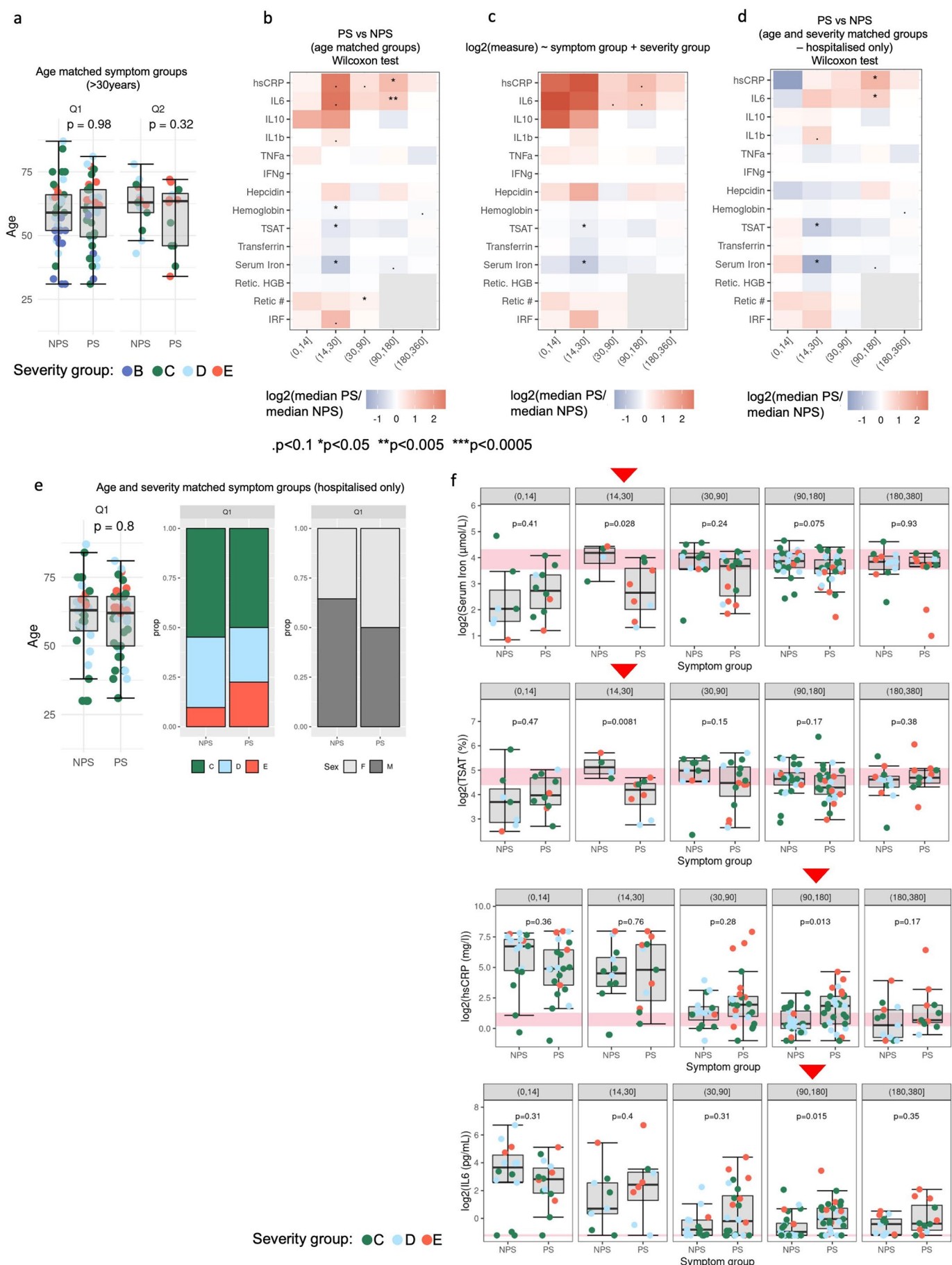

.p<0.1 *p<0.05 **p<0.005 ***p<0.0005

Severity group: ● C ● D ● E

**Extended Data Fig. 6 | See next page for caption.**

**Extended Data Fig. 6 | Sensitivity analysis of PASC symptom group differences. a**, Age matching of symptom groups upon exclusion of individuals <30 years of age. P-value derived from two-sided $t$-test. **b**, Results from re-analysis of clinical parameters using a two-sided Wilcoxon test applied to the age-matched PS and NPS groups shown in **a**. Black asterisks represent significance at p-value.p<0.1, *p<0.05, **p<0.005. **c**, Test of symptom group effect using a linear model applied to $\log_2$ transformed parameters with correction for acute disease severity group (group B-E). P-value thresholds as in **b. d**, Re-analysis of clinical parameters by two-sided Wilcoxon test in age and acute disease severity matched PS and NPS groups. Age and severity matching is shown in **e**, p-value thresholds as in **b. f**, Patient-level findings for iron and inflammatory parameters shown in **d**. Timepoints of interest are indicated with red arrows. Patients are colored by acute disease severity. Boxplots show minimum, 25th percentile, median, 75th percentile and maximum, and outliers beyond 1.5 times the interquartile range. PS = persisting symptom, NPS = no persisting symptoms.

# Reporting Summary

## Statistics

For all statistical analyses, confirm that the following items are present in the figure legend, table legend, main text, or Methods section.

| n/a | Confirmed | |
|---|---|---|
| ☐ | ☒ | The exact sample size (*n*) for each experimental group/condition, given as a discrete number and unit of measurement |
| ☐ | ☒ | A statement on whether measurements were taken from distinct samples or whether the same sample was measured repeatedly |
| ☐ | ☒ | The statistical test(s) used AND whether they are one- or two-sided<br>*Only common tests should be described solely by name; describe more complex techniques in the Methods section.* |
| ☐ | ☒ | A description of all covariates tested |
| ☐ | ☒ | A description of any assumptions or corrections, such as tests of normality and adjustment for multiple comparisons |
| ☐ | ☒ | A full description of the statistical parameters including central tendency (e.g. means) or other basic estimates (e.g. regression coefficient) AND variation (e.g. standard deviation) or associated estimates of uncertainty (e.g. confidence intervals) |
| ☐ | ☒ | For null hypothesis testing, the test statistic (e.g. *F*, *t*, *r*) with confidence intervals, effect sizes, degrees of freedom and *P* value noted<br>*Give P values as exact values whenever suitable.* |
| ☒ | ☐ | For Bayesian analysis, information on the choice of priors and Markov chain Monte Carlo settings |
| ☒ | ☐ | For hierarchical and complex designs, identification of the appropriate level for tests and full reporting of outcomes |
| ☐ | ☒ | Estimates of effect sizes (e.g. Cohen's *d*, Pearson's *r*), indicating how they were calculated |

*Our web collection on statistics for biologists contains articles on many of the points above.*

## Software and code

Policy information about availability of computer code

| | |
|---|---|
| Data collection | Flow cytometry data was gated using FlowJo v10.2. RNA sequencing data preprocessing was performed using a standard pipeline on a compute cluster. Briefly, sequencing read quality was assessed using FastQC v0.11.8 (Babraham Bioinformatics, UK), with trimming of SMARTer adaptors and poor-quality terminal bases (Phred score threshold <24) with Trim Galore v.0.6.4 (Babraham Bioinformatics, UK). Ribosomal rRNA contamination was removed using BBSplit (BBMap v.38.67) and clean reads aligned to the human reference genome GRCh38 using HISAT2 v2.1.0. Alignment .bam files were merged and read count matrix generated using the function featureCounts from the R package Rsubread (v.2.0.1). Count data was stored in a DGEList object with accompanying gene annotations and patient metadata for downstream handling (EdgeR v.3.28.1). |
| Data analysis | All statistical analysis was performed using R v3.6.0 with base and publicly available analysis packages: limma v3.42.2, edgeR v3.28.1, seurat v4.2.1 (in R v4.2.2 for preliminary data processing), dsb v1.0.2, mixOmics v6.10.9. Conversion between gene identifiers for RNASeq data analysis was facilitated by AnnotationDbi v1.48.0. |

For manuscripts utilizing custom algorithms or software that are central to the research but not yet described in published literature, software must be made available to editors and reviewers. We strongly encourage code deposition in a community repository (e.g. GitHub). See the Nature Portfolio guidelines for submitting code & software for further information.

# Data

Policy information about <u>availability of data</u>

All manuscripts must include a <u>data availability statement</u>. This statement should provide the following information, where applicable:

- Accession codes, unique identifiers, or web links for publicly available datasets
- A description of any restrictions on data availability
- For clinical datasets or third party data, please ensure that the statement adheres to our <u>policy</u>

> All datasets used in the generation of presented figures, including cell counts, serum measures, Sysmex hematology data, PAXGene whole-blood RNASeq gene expression counts, patient metadata and PASC group assignments can be downloaded from the Zenodo repository DOI: 10.5281/zenodo.10161238. Whole blood RNA-seq data are available through the European Genome-phenome Archive (EGA, ID: EGAS00001005332). CITE-seq processed data are available to download from Array Express using accession number E-MTAB-10026. Published CITE-seq data from Schulte-Schrepping et al., used for replication analysis, are available through the EGA (ID: EGAS00001004571). HALLMARK and WP_FERROPTOSIS gene sets are accessible through the MSigDB database: https://www.gsea-msigdb.org/gsea/msigdb/. The IRE gene set ("IRE_HQ") is available as supplement in the article by Hin et al., DOI: 10.3233/jad-210200, the iron-starvation gene set was taken from Table 3 ("genes upregulated in iron-free medium") in the associated publication by Chicault et al., DOI: 10.1152/physiolgenomics.00297.2005. Gene sets are also available in Supplementary Tables 2-4.

# Human research participants

Policy information about <u>studies involving human research participants and Sex and Gender in Research.</u>

| | |
|---|---|
| Reporting on sex and gender | Self reported biological sex information was collected on all participants. Males and females were analysed together throughout with correction for sex as a covariate in RNASeq analyses. |
| Population characteristics | Human participants recruited for this study in Cambridge, UK, were of both male and female sex and spanned an age range from ~20-90 years. Ethnicity was not a grounds for exclusion, however the majority of the cohort identified as "white", reflective of the predominance of this group in the UK population. COVID patients were confirmed SARS-CoV-2 positive via PCR testing at recruitment, healthy controls were confirmed SARS-CoV-2 negative or recruited prior to the outbreak of the SARS-CoV-2 virus. Hospitalisation or maximal oxygenation requirements were taken from available clinical records or reported by participants at the time of recruitment, and used to stratify patients based on disease severity as specified in the Methods. |
| Recruitment | Study participants were recruited from patients attending Addenbrooke's and Royal Papworth Hospitals in Cambridge, or Peterborough Foundation Trust, with a PCR test confirmed diagnosis of COVID-19, together with health care works attending the staff COVID-19 screening programme at Addenbrooke's Hospital. Controls were recruited from among screened staff members testing negative for SARS-CoV-2 infection, to supplement sampling from previously collected control recruited as part of the "Genetic variation AND Altered Leucocyte Function in health and disease" (GANDALF) study (below). Recruitment of inpatients at Addenbrooke's Hospital and Health Care Workers was undertaken by the NIHR Cambridge Clinical Research Facility outreach team and the NIHR BioResource research nurse team. Due to the bias towards older males in hospitalised COVID-19 groups, our moderate-severe disease severity groups were skewed towards this demographic. Milder disease patients were more predominantly younger females, the predominant demographic of health care workers sampled. We attempted to recruit controls matched to the full age range and gender split of the patient cohort where possible. Age and sex were corrected for in statistical analyses were they were found to be significant confounding factors. No statistical methods were used to pre-determine sample sizes, and recruitment was based on access to and availability of participants during national lockdown. |
| Ethics oversight | Ethics approval was obtained from the East of England – Cambridge Central Research Ethics Committee (''NIHR BioResource'' REC ref 17/EE/0025, and ''Genetic variation AND Altered Leucocyte Function in health and disease - GANDALF'' REC ref 08/H0308/176). All participants provided informed consent. |

Note that full information on the approval of the study protocol must also be provided in the manuscript.

# Field-specific reporting

Please select the one below that is the best fit for your research. If you are not sure, read the appropriate sections before making your selection.

☒ Life sciences ☐ Behavioural & social sciences ☐ Ecological, evolutionary & environmental sciences

For a reference copy of the document with all sections, see nature.com/documents/nr-reporting-summary-flat.pdf

# Life sciences study design

All studies must disclose on these points even when the disclosure is negative.

| | |
|---|---|
| Sample size | Samples were collected in a time-sensitive manner during the early days of the first COVID-19 outbreak, with new patient recruitment halted when numbers were deemed sufficient for statistical analysis (a n>= 18 for each classified severity group). Repeated sampling was performed |

| | for as long as participants where in hospital, or upon follow-up appointment. To maximise statistical power, samples were analysed in time windows post COVID-19 symptom onset, spanning two weeks to three months in length, retaining the earliest sampling timepoint per participant within these windows. Where available samples fell below n=3 for a given severity group within a given time window, data was not analysed to avoid spurious interpretation. |
|---|---|
| Data exclusions | Following clinician review, 6 originally recruited cases were considered not classifiable, due to complex concomitant pathologies that coexisted with COVID-19 and dominated the clinical picture, confounding the interpretation of clinical outcome. These cases were not included in any analyses. Samples collected from participants at any timepoint following vaccination with a SARS-CoV-2 specific mRNA vaccine, or from individuals where vaccine status could not be ascertained following roll-out of UK COVID-19 vaccination progammes in Cambridge in December 2020, were excluded. This was following preliminary analysis identifying an influence of vaccination on immune cell and cytokine parameters. Nineteen samples with fewer than 2,000,000 assigned reads, and one sample with an abnormal read distribution were excluded from RNASeq analysis following standard quality filtering. |
| Replication | The extensive longitudinal nature of this study made it difficult to find comparable COVID-19 cohorts with which to replicate analysis of late stage disease parameters (beyond one month post symptom onset). We did however seek to replicate the single-cell investigation of altered iron homeostatsis in blood cell populations in early disease. Analysis of publicly available data from Schulte-Schrepping et al. (Cell 2020) confirmed the predominant expression of iron homeostasis genes in monocyte subpopulations, and the decreased number of cells in these populations in the periphery of COVID-19 patients (Supp Fig 13C). |
| Randomization | Samples were randomised by the primary outcome variable (severity group) across all conducted assays. To test for and correct potential drift in instrumentation readings over time in a study spanning one year in duration, technical replicates or standards were included in all biological assays. Compensation controls were used to test for drift in marker fluorescence over time in flow-cytometry data, and gating rigorously checked across early and late timepoint samples by three independent analysts. All hospital run assays included internal controls. Healthy controls and asymptomatic (group A) patients (who showed no evidence of differential expression relative to healthy controls at any timepoint) were used to test for batch effects in temporally processed and sequenced PAXgene extracted RNA. Batch was included as a covariate in the statistical analysis of RNASeq data. |
| Blinding | No blinding was performed as careful clinical curation of severity groups was required for the study design. |

# Reporting for specific materials, systems and methods

We require information from authors about some types of materials, experimental systems and methods used in many studies. Here, indicate whether each material, system or method listed is relevant to your study. If you are not sure if a list item applies to your research, read the appropriate section before selecting a response.

## Materials & experimental systems

| n/a | Involved in the study |
|---|---|
| ☐ | ☒ Antibodies |
| ☒ | ☐ Eukaryotic cell lines |
| ☒ | ☐ Palaeontology and archaeology |
| ☒ | ☐ Animals and other organisms |
| ☒ | ☐ Clinical data |
| ☒ | ☐ Dual use research of concern |

## Methods

| n/a | Involved in the study |
|---|---|
| ☒ | ☐ ChIP-seq |
| ☐ | ☒ Flow cytometry |
| ☒ | ☐ MRI-based neuroimaging |

# Antibodies

| Antibodies used | All antibodies used in this study, supplier names and product codes are provided in the CellPress Star Methods section of the publication that first describes this cohort (Bergamaschi, L et al. 2021, Immunity). |
|---|---|
| Validation | The immunophenotyping flow cytometry panels used in this study were adapted from rigorously validated panels previously developed by members of the NIHR Cambridge BRC Cell Phenotyping Hub for immune phenotyping in complex human immune-mediated diseases. |

# Flow Cytometry

## Plots

Confirm that:

☐ The axis labels state the marker and fluorochrome used (e.g. CD4-FITC).

☐ The axis scales are clearly visible. Include numbers along axes only for bottom left plot of group (a 'group' is an analysis of identical markers).

☐ All plots are contour plots with outliers or pseudocolor plots.

☒ A numerical value for number of cells or percentage (with statistics) is provided.

## Methodology

**Sample preparation**

Each participant provided 27 mL of peripheral venous blood collected into 9 mL sodium citrate tube. Peripheral blood mononuclear cells (PBMCs) were isolated using Leucosep tubes (Greiner Bio-One) with Histopaque 1077 (Sigma) by centrifugation at 800x g for 15 min at room temperature. PBMCs at the interface were collected, rinsed twice with autoMACS running buffer (Miltenyi Biotech) and cryopreserved in FBS with 10% DMSO. All samples were processed within 4 h of collection.

**Instrument**

5-laser BD Symphony X-50 flow cytometer

**Software**

FlowJo v10.2

**Cell population abundance**

For direct enumeration of T, B and NK cells, an aliquot of whole blood (50 ml) was added to BD TruCount tubes with 20ml- BD Mul- titest 6-color TBNK reagent (BD Biosciences) and processed as per the manufacturer's instructions.
Samples were gated in FlowJo v10.2 according to the schema set out in Bergamaschi, L et al. 2021 (Immunity) Data S4. The number of cells falling within each gate was recorded. For analysis, these were expressed as an absolute concentration of cells per ml, calculated using the proportions of daughter populations present within the parent population determined using the BD TruCountsystem.

**Gating strategy**

A detailed gating schema is set out in Bergamaschi, L et al. 2021 (Immunity) Data S4

☒ Tick this box to confirm that a figure exemplifying the gating strategy is provided in the Supplementary Information.

