## [Peer Review File · Nature Immunology]

Peer Review Information

Journal: Nature Immunology

Manuscript Title: Iron Dysregulation and Inflammatory Stress Erythropoiesis Associates with Long-Term Outcome of COVID-19

Corresponding author name(s): Professor Kenneth Smith

Reviewer Comments & Decisions:

Decision Letter, initial version:

5th Apr 2023

Dear Dr. Hanson,

We have now finished reviewing your manuscript entitled "Iron Dysregulation and Inflammatory Stress Erythropoiesis Associates with Long-Term Outcome of COVID-19", reference number NI-A35512-T. Although the editors thought that the manuscript was interesting enough to send out for in-depth review, the reviewers were not in favor of publishing the paper in Nature Immunology because of the conceptual advance your findings represent over earlier work and the appropriateness of the controls in the study cohort. We are therefore returning the reviews to you with the hope that you find them useful when you prepare the paper for another journal.

We realize that this is disappointing. I hope that you continue to consider Nature Immunology for your results most significant for the immunology community and wish you well in your future investigations.

Sincerely,

Ioana Visan, Ph.D.
Senior Editor
Nature Immunology

Tel: 212-726-9207
Fax: 212-696-9752
www.nature.com/ni

Reviewers' comments:

Reviewer #1 (Remarks to the Author):

A. Summary of the key results

The authors analyze data of 214 SARS-CoV-2 patients of varying severity for up to one year from COVID-19 symptom onset. In patients with persistent symptoms, they document unresolved inflammation, anemia and hypoferrremia with eventual compensatory but iron-deficient reticulocytosis. They also report lymphopenia, high plasmablast numbers, low plasmacytoid DC numbers and iron deficiency in lymphocytes. They speculate that iron dyshomeostasis and delayed recovery from anemia cause inefficient oxygen transport, and persistence of symptoms, and may contribute to unresolved inflammation.

B. Originality and significance: if not novel, please include reference

This publication is comprehensive and analyzes a relatively large number -214- patients across the spectrum of severity. However, the number of blood samples 180 days or more after diagnosis is much smaller. There is little here that is new or unexpected. Others have explored similar ideas, albeit with smaller patient numbers:

Anemia and hypoferrremia in persistently symptomatic Covid-19 patients (references 34 and 78):

1. Sonnweber et al. The Impact of Iron Dyshomeostasis and Anaemia on Long-Term Pulmonary Recovery and Persisting Symptom Burden after COVID-19: A Prospective Observational Cohort Study *Metabolites* 2022 Jun 14;12(6):546. doi: 10.3390/metabo12060546.

2. Sonnweber et al. Persisting alterations of iron homeostasis in COVID-19 are associated with non-resolving lung pathologies and poor patients' performance: a prospective observational cohort study. *Respir Res.* 2020 Oct 21;21(1):276. doi: 10.1186/s12931-020-01546-2.

The association of persistent symptoms with chronic inflammation or immune abnormalities:

3. PHOSP-COVID Collaborative Group: Clinical characteristics with inflammation profiling of long COVID and association with 1-year recovery following hospitalisation in the UK: a prospective observational study *Lancet Respir Med.* 2022 Aug;10(8):761-775.

4. Peluso et al. Long-term SARS-CoV-2-specific immune and inflammatory responses in individuals recovering from COVID-19 with and without post-acute symptoms *Cell Rep.* 2021 Aug 10;36(6):109518. doi: 10.1016/j.celrep.2021.109518. Epub 2021 Aug 6.

C. Data & methodology: validity of approach, quality of data, quality of presentation

This is a longitudinal observational study of post-Covid symptoms, and as such cannot attribute causality, the big unanswered question in the long-Covid story. In this regard, the problem is the same as it is for other situations where anemia of chronic inflammation is observed: It is not clear how much anemia and iron sequestration contribute to symptoms. The only way to resolve this conundrum is to treat anemia and the iron disorder and document whether it improves the symptoms or not.

The analytical methodology is up to date, including analysis of PBMC/lymphocyte subsets, key cytokines, and whole blood transcriptomics. However, the data are incomplete and do not provide much new insight as discussed below.

The analysis of IRP1 and IRP2 effects is incomplete because only transcripts are measured, even though 5'IRES exert their effects predominantly by regulating translation. This includes such genes/proteins as FTL1, FTH1, SLC40a1 (ferroportin), and EPAS1 (HIF2alpha) analyzed in line 169-177.

As the authors note, the prominence of heme metabolism transcripts is an expected consequence of

reticulocytosis as these immature erythrocytes have not yet degraded their mRNAs. This is a relatively trivial conclusion.

Whole blood transcriptomics can only detect changes in mRNA-rich cell types that circulate in blood, hence the implication of monocytes in iron transport and lymphocytes as targets of iron deficiency is an expected consequence of their known biology (lines 200-235). The symptoms are not likely to be caused by these cell types directly.

It is not clear that the severity of lymphocyte iron restriction is sufficient to affect their immunologic function, despite statements to the contrary based on much more severe restriction (lines 233-5).

Q1 symptoms correlated with unresolved inflammation (lines 250-258) but this is again not surprising.

The significance of "stress erythropoiesis" may be misinterpreted because of limited sampling. It is likely that iron-restricted reticulocytosis also occurs in patients who recover early from Covid-19, and it may be the delay of erythropoietic recovery marked by delayed occurrence of reticulocytosis that is an adverse prognostic marker for persistent symptoms.

D. Appropriate use of statistics and treatment of uncertainties; no concerns

E. Conclusions: robustness, validity, reliability: no concerns

F. Suggested improvements: experiments, data for possible revision

Line 160 "yet accumulated cytosolic free iron is readily converted to reactive oxygen species (ROS), ..." Iron catalyzes the conversion but it is not "converted".

G. References: appropriate credit to previous work?--Yes

H. Clarity and context: lucidity of abstract/summary, appropriateness of abstract, introduction and conclusions

Figure 1EF could be made easier to see by using better color combinations

Reviewer #2 (Remarks to the Author):

In this study Hanson and colleagues tried to identify early markers of post-acute sequelae of COVID-19 (PASC). Mainly based on investigations on blood immune cell composition, inflammatory soluble factors including parameters of iron metabolism and gene expression studies in the early acute infection period, comparing samples from patients with different severity, they postulate that increased immune cell activation and iron-deficient reticulocytosis are indicative of PACS. PACS represents an important clinical problem and our understanding of pathomechanisms contributing to PACS development is sparse. Thus, well-designed studies revealing such pathomechanisms are urgently needed.

In the present study appropriate state-of-the-art multi-omics-technologies were applied generating high quality data. However, the study design is not appropriate which weakens the obtained findings quit severely.

Main concern:

The authors start off with studying the normalization (in comparison to value of uninfected controls) of blood immune cell markers of their previously published patient groups (HC versus A to E). Although the age of the patients is not provided in the present study, we know from the previous published cohort that controls as well as group A and B patients were much younger in comparison to

hospitalized patient groups C to E. It is well known that certain inflammatory markers such as IL-6 increase and iron, hemoglobin levels decrease with aging. Thus, a direct comparison to the same controls is not correct. Unfortunately, this age bias affects all further analyses, which in my view questions the whole study. This also concerns the most interesting analysis of identifying PACS discriminators in figure 5 and 6. The non-PACS group is heavily dominated by the non-hospitalized group B patients, who of course are much younger compared to the hospitalized group C to E patients. Nearly none of the non-hospitalized patients (group B) developed PACS. Thus, in conclusion such an analysis can only be performed in well age- and sex-matched, much larger cohorts including appropriate controls.

Author Rebuttal to Initial comments

See inserted PDF

Reviewer #1 (Remarks to the Author):

A. Summary of the key results

The authors analyze data of 214 SARS-CoV-2 patients of varying severity for up to one year from COVID-19 symptom onset. In patients with persistent symptoms, they document unresolved inflammation, anemia and hypoferrremia with eventual compensatory but iron-deficient reticulocytosis. They also report lymphopenia, high plasmablast numbers, low plasmacytoid DC numbers and iron deficiency in lymphocytes. They speculate that iron dyshomeostasis and delayed recovery from anemia cause inefficient oxygen transport, and persistence of symptoms, and may contribute to unresolved inflammation.

B. Originality and significance: if not novel, please include reference

This publication is comprehensive and analyzes a relatively large number -214- patients across the spectrum of severity. However, the number of blood samples 180 days or more after diagnosis is much smaller. There is little here that is new or unexpected. Others have explored similar ideas, albeit with smaller patient numbers:

The claims that our study is not novel are based on work that touches on aspects of what we report, but in each case does not replicate or explore the key central observations that we make. We will deal with the specific instances below, but note that our study is unique in that it is to first to:

1. Collect inflammatory and serum iron measures, haematological parameters reflecting the status of erythropoiesis, and whole blood and single cell transcriptomic data reflecting cellular responses to iron dysregulation, repeatedly, from the same patients, for one year following acute COVID-19. This has enabled us to provide a far more comprehensive picture of the physiological consequences of anaemia in these patients than previous studies.
2. Report that delayed erythropoietic responses to early anaemia in moderate to severe COVID-19 are iron deprived, reflecting a profound breakdown in iron recirculation following hepcidin-induced iron restriction.
3. Correlate features of inflammatory anaemia and iron dysregulation across the full trajectory of COVID-19 with the presence, and severity, of late-stage symptoms. This has allowed us to demonstrate that early features of disease, and in particular their haematopoietic consequences, are more strongly correlated with outcome than biological measures collected at the time of late-stage symptom reporting. This observation is both unique, and important, in that it points the way to specific early intervention strategies to correct inflammatory iron restriction, potentially reducing long-COVID incidence and severity.

Thus this work expands substantially on our understanding of iron dysregulation and its consequences in COVID-19, and is not comparable to previous studies that report only on the prevalence of individual aspects of this phenotype.

Anemia and hypoferrremia in persistently symptomatic Covid-19 patients (references 34 and 78):

1. Sonnweber et al. The Impact of Iron Dyshomeostasis and Anaemia on Long-Term Pulmonary Recovery and Persisting Symptom Burden after COVID-19: A Prospective Observational Cohort Study *Metabolites* 2022 Jun 14;12(6):546. doi: 10.3390/metabo12060546.
2. Sonnweber et al. Persisting alterations of iron homeostasis in COVID-19 are associated with non-resolving lung pathologies and poor patients' performance: a prospective observational cohort study. *Respir Res.* 2020 Oct 21;21(1):276. doi: 10.1186/s12931-020-01546-2.

Sonnweber et al. (2020 and 2022) are two of many studies that have reported on the prevalence and persistence of anaemia and iron dysregulation in COVID-19 cohorts (e.g. see also PMID: 32751400, 34677368, 32517773, 33205000), characterising the phenotype using blood-based serum iron measurements and inflammatory parameters. They find an association between persistently high ferritin

levels and initial disease severity. We are not focussed on these well-known associations of iron dysregulation with disease severity, though we confirm them. We have instead characterized the potential cellular manifestations and systemic repercussions of iron maldistribution in the context of ongoing inflammation in COVID-19. Further, we capture the kinetics of iron dysregulation across patients ranging in disease severity, sampled from symptom onset, and show that early dysregulation of iron homeostasis is associated with subsequent persistent symptoms of disease.

The association of persistent symptoms with chronic inflammation or immune abnormalities:

3. PHOSP-COVID Collaborative Group: Clinical characteristics with inflammation profiling of long COVID and association with 1-year recovery following hospitalisation in the UK: a prospective observational study *Lancet Respir Med.* 2022 Aug;10(8):761-775.

The PHOSP-COVID study interrogated patient-reported health outcomes at 5 and 12 months following hospitalisation with COVID-19, and correlated these with clinical baseline variables (such as sex, BMI and need for ventilation). Plasma proteome data, generated from blood taken at the 5-month visit, was compared between patients with mild or severe ongoing clinical features. This study reported extensively on the prevalence of persistent symptoms, however it did not investigate kinetic changes in biological features of acute disease, or correlate these with long-term outcome. It therefore provided a descriptive picture of “long-COVID”, but was not designed to include the dense prospective sampling needed to determine early predictors of PASC in the way our study has done. As a result, and perhaps more importantly, the PHOSP-COVID study did not address the issue of iron homeostasis/erythropoiesis in COVID-19, which is a primary focus of our paper.

4. Peluso et al. Long-term SARS-CoV-2-specific immune and inflammatory responses in individuals recovering from COVID-19 with and without post-acute symptoms *Cell Rep.* 2021 Aug 10;36(6):109518. doi: 10.1016/j.celrep.2021.109518. Epub 2021 Aug 6.

Peluso et al. (2021) analysed samples from 70 COVID-19 patients with diverse disease presentations, with the first samples collected a median of 53 days after symptom onset. They quantified and phenotyped SARS-CoV-2 specific lymphocytes, and demonstrated their stability over time. They found differences in the frequencies of these cell populations between patients grouped by hospitalisation status, ethnicity and prior comorbidities, but no differences in T cell responses between patients who did or did not report persisting symptoms. It is unclear how this study, which focuses on the durability of the adaptive immune response to infection, overlaps with the findings of our work.

The issue of persistent COVID-19 sequelae is a complex, difficult and important one, and has been the subject of a large number of studies. Those papers chosen by the Reviewer only tangentially touch on the major point of this paper (refs 1 and 2), or don't do so at all (refs 3 and 4). The novelty of our work is not impacted by the publications mentioned, though we will cite them and acknowledge the associations between COVID severity and both anaemia and high ferritin that they, and many others, have seen.

Page 14, Line 398:

“A growing body of literature points to a role for altered iron handling in acute COVID-19³⁴⁻³⁸, and persisting iron dysregulation has been associated with initial disease severity in convalescent patients³⁵⁻³⁹” ... “Consistent with previous reports^{34,36,37,40,41}, we observed low serum iron, reduced TSAT score and hemoglobin concentrations, and concurrent raised ferritin, hepcidin and IL-6 in hospitalized COVID-19 patients from symptom onset, indicating AI in moderate-severe disease.”

Page 18, Line 500

“Iron deficiency and cerebral hypoxia have been linked to cognitive impairment and altered mood, and iron deficiency during childhood is a significant risk factor for poor cognitive performance⁸¹⁻⁸³. Inflammation of the type associated with iron dysregulation (with, for example, elevated IL6 in

hospitalized patients 5 months post-discharge) has also been associated with cognitive impairment following COVID-19⁵.”

C. Data & methodology: validity of approach, quality of data, quality of presentation

This is a longitudinal observational study of post-Covid symptoms, and as such cannot attribute causality, the big unanswered question in the long-Covid story. In this regard, the problem is the same as it is for other situations where anemia of chronic inflammation is observed: It is not clear how much anemia and iron sequestration contribute to symptoms. The only way to resolve this conundrum is to treat anemia and the iron disorder and document whether it improves the symptoms or not.

We of course never claim that we have demonstrated causality – as the reviewer points out, this is impossible in an association study. We will revise the manuscript to make that point even more clear. We have however, through dense data generation from repeated blood sampling, provided a compelling hypothesis that warrants highlighting in the literature. Interestingly, recently published data suggests iron overload in the context of transfusion dependent β -thalassemia can protect from severe disease and mortality in SARS-CoV-2 infected individuals, consistent with a potential protective effect of high serum iron levels (PMID: 35355397). We will highlight this in the discussion (as below).

As the reviewer suggests, a clinical trial directly assessing the effect of correcting low serum iron/anaemia on long term patient outcomes is required to prove causality. This is something we advocate in the concluding paragraphs of our paper. The point of this work is to provide the support for exactly such studies.

Page 18, Line 520:

“Recently published data suggests iron overload in the context of β -thalassemia protects from severe disease and mortality in SARS-CoV-2 infected individuals⁸⁶, consistent with a potential protective effect of increased iron availability.”

The analytical methodology is up to date, including analysis of PBMC/lymphocyte subsets, key cytokines, and whole blood transcriptomics. However, the data are incomplete and do not provide much new insight as discussed below.

The reviewer implies this manuscript is compromised as “the data are incomplete”. We agree with the protein issue raised (see below) but, given the enormous amount of data we present, we do not believe that this single relatively minor and easily rectifiable example justifies calling the whole manuscript into question.

The analysis of IRP1 and IRP2 effects is incomplete because only transcripts are measured, even though 5'IREs exert their effects predominantly by regulating translation. This includes such genes/proteins as FTL1, FTH1, SLC40a1 (ferroportin), and EPAS1 (HIF2alpha) analyzed in line 169-177.

The reviewer is correct – we did not present protein data, and this could be informative as the iron response proteins (IRPs) regulate the expression of cellular iron homeostasis genes post-transcriptionally. We can provide some protein data to address the issue of the relationship between mRNA and protein levels in this context. We find, for example, a positive correlation between ferritin (FTL1 and FTH1) transcript and protein levels (as below). We will present this, together with any additional protein data we can source from study collaborators for genes of interest, in a future iteration of the supplementary material.

Pearson correlation between ferritin (*FTH1* and *FTL*) transcript expression and protein expression in matched patient samples from COVID-19 severity groups and HC.

As the authors note, the prominence of heme metabolism transcripts is an expected consequence of reticulocytosis as these immature erythrocytes have not yet degraded their mRNAs. This is a relatively trivial conclusion.

Of course it is trivial – we nowhere suggest otherwise. The enrichment of haem metabolism transcripts in reticulocytes (detected by CITE-Seq; Figure 4B), was presented simply to confirm the origin of the strong late-stage haem-metabolism signature observed in whole blood (Figure 2H), which correlated with the timeframe of reticulocyte expansion. This was done to complete the picture, out of thoroughness, and does not imply we think we have for the first time discovered immature erythrocytes! We could, if necessary, underline that point in a revised manuscript.

Whole blood transcriptomics can only detect changes in mRNA-rich cell types that circulate in blood, hence the implication of monocytes in iron transport and lymphocytes as targets of iron deficiency is an expected consequence of their known biology (lines 200-235). The symptoms are not likely to be caused by these cell types directly.

We are not quite sure precisely what point is being made here, but agree observations made using whole blood transcriptomic data are influenced by the predominant cell types in blood, and interpretation is complicated by the cellular heterogeneity of the sample. It was for this reason we performed the CITESeq analysis described in the paper.

The Reviewer might be making one or both of two points:

1. The circulating monocytes or lymphocytes are unlikely to drive disease because they are circulating. We would agree with this – and would reword the manuscript to emphasise the (perhaps obvious) point that it is likely they reflect similar, or indeed more pronounced, dysregulation of their tissue-localised counterparts.

Page 16 Line 461:

“Although blood monocytes themselves are unlikely to be the cells driving disease, loss of this circulating population during COVID-19, seen here and elsewhere^{17,27,69}, suggest these iron-laden cells may become resident in tissues as macrophages, where they may contribute directly to tissue damage. Iron-laden bone marrow macrophages are detectable in COVID-19 patients post-mortem⁷⁰, and iron accumulation and ferroptosis in the ventricular myocardium or in the liver has been associated with end-organ damage and thence fatal COVID-19 disease⁷¹⁻⁷³.”

2. That dysregulated pro-inflammatory myeloid lineage cells, or dysfunctional lymphocytes, are not related to persistent COVID-19 symptoms irrespective of anatomical localisation. We believe their involvement to be likely, given for example the profound fatigue commonly seen in people with a

range of inflammatory and immune-mediated diseases. In addition, previous work has demonstrated iron dysregulation severely compromises immune cell function and may have pathogenic consequences (PMID: 36197985, 33665641, 32959998, 34064487). However, we agree this remains somewhat speculative, and precise mechanisms have not been defined, and we could emphasise this in a revised manuscript.

It is not clear that the severity of lymphocyte iron restriction is sufficient to affect their immunologic function, despite statements to the contrary based on much more severe restriction (lines 233-5).

This is indeed speculation, but we believe reasonable speculation. We have emphasised this by explaining that the functional consequences of iron restriction on lymphocytes in the context of SARS-CoV-2 infection need to be investigated experimentally.

Page 10, Line 263:

“Together, multimodal single-cell analysis revealed the cells demonstrating signatures of defective iron homeostasis in blood from COVID-19 patients, implicating iron sequestration in monocytes and concurrent iron deprivation of proliferating lymphocytes. To what extent this lymphocyte iron restriction compromises immunity remains to be confirmed, but in vivo studies show profound effects of hepcidin-induced iron restriction on lymphocyte primary and memory responses to vaccination and infection” (PMID: 33665641).

Q1 symptoms correlated with unresolved inflammation (lines 250-258) but this is again not surprising.

The importance of this observation is not that persisting symptoms “correlated with unresolved inflammation”, but rather, COVID-19 patients reporting persisting symptoms beyond 3 months post initial disease can be differentiated by a biological signature reflecting inflammation and anaemia detectable far earlier than this (2-4 weeks) – this is what differentiates it from those reported in previous studies of late timepoints alone as quoted by the Reviewer above. This is a potentially important and novel observation, irrespective of whether it elicits surprise.

The significance of “stress erythropoiesis” may be misinterpreted because of limited sampling. It is likely that iron-restricted reticulocytosis also occurs in patients who recover early from Covid-19, and it may be the delay of erythropoietic recovery marked by delayed occurrence of reticulocytosis that is an adverse prognostic marker for persistent symptoms.

A strength of this study is that its sampling is extensive enough to have provided a detailed answer to this question. When reticulocyte counts are compared between patients with either persisting COVID-19 symptoms (PS) or no persisting symptoms (NPS) and controls, reticulocyte expansion at the 30-90 day time window can be observed in both (albeit to a significantly greater degree in the PS than the NPS group; Figure 5E), possibly reflecting compensatory reticulocytosis in response to more severe initial anaemia. The PS group also have significantly lower haemoglobin, serum iron levels and TSAT score than the NPS group at 14-30 DPSO (parameters which have resolved to normal in the NPS group by this time, Figure 4E). Based on this data, it appears that it is not a delay in reticulocytosis that distinguishes the symptom groups, but the availability of iron for erythropoietic recovery. We have made this clearer to the reader and will show the analyses mentioned in an additional supplementary figure:

Supplementary Figure 20: Reticulocyte counts over time in patients reporting PS or NPS at Q1. P-values calculated by Wilcoxon test relative to HC (* $p < 0.05$, ** $p < 0.005$, *** $p < 0.0005$). Pink bar indicates the HC interquartile range. PS = persisting symptoms, NPS = no persisting symptoms.

Page 11, Line 305:

“Upon direct comparison, Q1 PS individuals had a significant increase in CRP, IL-6 and IL-1 β , and decrease in serum iron, haemoglobin and TSAT score relative those with NPS at 15-30 DPSO (Fig. 5E;

Supplementary Fig. 19). Reticulocyte counts were elevated above HC levels at 31-90 DPSO in both PS and NPS groups (Supplementary Fig. 20), corresponding with the peak of the stress-erythropoietic response reported above, though counts were significantly higher in the PS than the NPS group at this time (Fig. 5E, Supplementary Fig. 19). This suggests that iron availability for erythropoietic recovery, rather than delayed reticulocytosis, is an early differentiator of long-term symptoms groups.”

D. Appropriate use of statistics and treatment of uncertainties; no concerns

E. Conclusions: robustness, validity, reliability: no concerns

F. Suggested improvements: experiments, data for possible revision

Line 160 “yet accumulated cytosolic free iron is readily converted to reactive oxygen species (ROS), ...” Iron catalyzes the conversion but it is not “converted”.

This wording is fixed in the revised version of the manuscript.

Page 7, Line 180:

“Iron is essential for cellular respiration and metabolism, yet accumulated cytosolic free iron catalyses the production of reactive oxygen species (ROS), contributing to lipid membrane peroxidation and cell death (termed ferroptosis).”

G. References: appropriate credit to previous work?—Yes

H. Clarity and context: lucidity of abstract/summary, appropriateness of abstract, introduction and conclusions

Figure 1EF could be made easier to see by using better color combinations

The reviewer is likely to be referring to the pink band showing the IQR here, which we will change to grey in this and other figures to improve contrast. The colour scheme used for severity groups (A-E) is used for continuity with the earlier published characterisation of this cohort (PMID: 34051148).

Reviewer #2 (Remarks to the Author):

In this study Hanson and colleagues tried to identify early markers of post-acute sequelae of COVID-19 (PASC). Mainly based on investigations on blood immune cell composition, inflammatory soluble factors including parameters of iron metabolism and gene expression studies in the early acute infection period, comparing samples from patients with different severity, they postulate that increased immune cell activation and iron-deficient reticulocytosis are indicative of PACS. PACS represents an important clinical problem and our understanding of pathomechanisms contributing to PACS development is sparse. Thus, well-designed studies revealing such pathomechanisms are urgently needed.

In the present study appropriate state-of-the-art multi-omics-technologies were applied generating high quality data. However, the study design is not appropriate which weakens the obtained findings quit severely.

We will address the specific questions raised initially, and then the issue of “study design” in the throes of a real-world pandemic at the end.

Main concern:

The authors start off with studying the normalization (in comparison to value of uninfected controls) of blood immune cell markers of their previously published patient groups (HC versus A to E). Although the age of the patients is not provided in the present study, we know from the previous published cohort that controls as well as group A and B patients were much younger in comparison to hospitalized patient groups C to E. It is well known that certain inflammatory markers such as IL-6 increase and iron, hemoglobin levels decrease with aging. Thus, a direct comparison to the same controls is not correct.

We did present age distribution data in Figure 1B, and already highlight it in the first paragraph of the results section. For ease of reference we will include in the Supplementary a modified version of Supplementary Table S1 from Bergamaschi et al.

Unfortunately, this age bias affects all further analyses, which in my view questions the whole study. This also concerns the most interesting analysis of identifying PACS discriminators in figure 5 and 6. The non-PACS group is heavily dominated by the non-hospitalized group B patients, who of course are much younger compared to the hospitalized group C to E patients. Nearly none of the non-hospitalized patients (group B) developed PACS.

Like many others, we find that those hospitalised with moderate-severe COVID-19 tend to be older on average than the healthy population (Figure 1B), and thus control groups are often not well matched. We have taken great efforts to ensure our results and conclusions are not impacted by this potential bias. We have clearly not made this point strongly enough in the manuscript, and will rectify that with additional text and supplementary figures.

1. We have incorporated an additional supplementary figure (below) which shows all immune cell comparisons between COVID-19 severity groups and healthy controls both with values either corrected for age and sex (red stars) or not (black stars). The former mirror very closely the statistical results derived from non-age-corrected comparisons, and do not change the interpretation of presented findings. Both sets of results are presented in the current Supplementary Figure 2, with Supplementary Figure 3 shown extended cellular data.

Supplementary Figure 2: Heatmap showing log₂ fold-change in absolute cell counts between COVID-19 patients and HC, across severity groups and within time bins. P-values are calculated by T-test comparison of log₂(count + 1) transformed data between test COVID-19 group and HCs (*p<0.05, **p<0.005, *p<0.0005). Asterisks in red indicate p-value from a linear model of group effect on log₂(count + 1) with age and sex correction.**

Page 5, Line 106:

“These differences remained statistically significant when correcting for age and sex to account for biases in patient demographics (Supplementary Fig. 2 and 3) and collectively demonstrate prolonged immunological disruption following moderate-to-severe COVID-19.”

- We have incorporated an additional supplementary figure (below) showing serum inflammatory and iron parameter comparisons between study groups with and without age correction. Again, our findings are robust to age correction, and this does not change the interpretation of the presented data.

Supplementary Figure 5: Heatmap showing log₂ fold-change in median serum inflammatory, iron and erythroid cell parameters between COVID-19 patients and HC (or group A and B samples taken at >90 DPSO in the absence of HC measures for ferritin and hemoglobin), across severity groups and within time bins. P-values are calculated by Wilcoxon test (COVID-19 vs HC), black asterisks, or from a linear model of group effect on log₂(measure) with age and sex correction, red asterisks; *p<0.05, **p<0.005, *p<0.0005.**

Page 6, Line 137:

“Age and sex differences between patient groups and HC did not account for persisting differences in inflammatory and iron-related parameters (Supplementary Fig. 5).”

3. Transcriptomic analysis was originally performed with adjustment for age and sex as covariates, and so we have made no further changes, apart from now highlighting this adjustment in the main text on page 7. The formula used for differential expression in R notation was as below, and is now included in the methods:

```
design <- model.matrix(~0+group+age+sex+extractbatch)
```

Where group specifies a composite variable capturing patient severity group and time window. These estimates thus represent the difference in gene expression that remain after accounting for the differences in expression that were due to these intrinsic factors.

Page 7, Line 187:

“All gene expression comparisons were corrected for age and sex differences between patients and HCs.”

4. The reviewer is correct that older age, moderate-severe COVID patients are over-represented in those who develop PASC, and compared to a younger “non-PASC” group in Figure 5. However, age does not appear to impact PASC risk beyond being a close correlate of acute disease severity. When assessed within each disease severity group (B-E) there is no difference in age between those who do or do not go on to develop PASC, as we now show in Supplementary Figure 17 (below).

Supplementary Figure 17: Age comparison between individuals with PS or NPS at Q1, stratified by initial disease severity group. P-values calculated by T-test. PS = persisting symptoms, NPS = no persisting symptoms.

5. Formal age correction of the data presented in Figure 5E is now included in Supplementary Figure 19 (below). This shows that CRP and IL-6 levels are still statistically significantly higher in the persisting symptom group over the non-persisting symptom group at 90-180 DPSO, iron abnormalities remain significant at 15-30 DPSO and increased reticulocyte counts are still detectable at 31-90 DPSO as reported in the text. As mentioned above, RNASeq analyses already

include correction for age and sex as covariates when comparing “long-COVID” symptom groups and so presented results are robust to these differences.

Supplementary Figure 19: Heatmap showing log₂ fold-change in serum inflammatory and iron measures, reticulocyte parameters and absolute immune cell counts between PS and NPS groups, across severity groups and within time windows. Symptom groupings from 0-180 DPSO are derived from Q1 questionnaire responses, from 180- 360 DPSO from Q2 responses. Black asterisks represent significance as calculated by T-test of log₂ transformed counts (PS versus NPS), as in Figure 5E. **Asterisks in red indicate p-value from a linear model of group effect on cell count with age correction.** All p-value are shown as †p<0.1, *p<0.05, **p<0.005, ***p<0.0005. PS = persisting symptoms, NPS = no persisting symptoms.

Thus the reviewer has raised an important issue. We will emphasise this, and our robust statistical approach to it, in paragraphs in the results and discussion, additions to the Methods and provision of the additional supplementary figures described above.

Thus, in conclusion such an analysis can only be performed in well age- and sex-matched, much larger cohorts including appropriate controls.

Unfortunately studies done at the height of a pandemic are very much “real world” studies, and not clinical trials or mouse experiments. This is one of the largest and most comprehensive COVID-19 studies generated at this time. Samples were collected from sick people who generously agreed to be research subjects, in hospitals filled to bursting with dying patients, by busy staff and volunteers putting themselves at real personal risk. Thus patients were older (because they were) and asymptomatic controls were younger as they were staff picked up at routine screening – we were not able to go out into the community to get age matched asymptomatic controls as there was a lock-down, this would have risked spreading infection and we had neither the staff nor laboratory capacity to do so. As an aside, we were the first to introduce routine staff screening in the UK (PMID: 32392129), performing it in the academic lab as the NHS had no testing capacity.

If we are to learn about the sequelae of acute COVID-19 we have to make the most of these real world studies, and cannot just generate “well age- and sex-matched, much larger cohorts”. Thus we need to apply the best statistical correction methodology we can – which we had done and will now emphasise more – and, as implied by the Reviewer, should describe this issue and the caveats it raises in more detail. But we should not make the perfect the enemy of the good, and in doing so throw out such hard-won and potentially important clinical observations.

Decision Letter, first revision:

Dear Dr. Hanson,

Thank you for your letter asking us to reconsider our decision on your manuscript, "Iron Dysregulation and Inflammatory Stress Erythropoiesis Associates with Long-Term Outcome of COVID-19". We would be willing to reconsider a revised manuscript that addresses the referees' concerns, as outlined in your letter. However, you'll understand that we cannot predict the outcome of the re-review process that may in the end be the same. While I should emphasize that our referees were very well placed to evaluate the manuscript, we may choose to involve a third referee in the event of resubmission, and any additional points this referee may raise would have to be addressed as well.

Once you have made these revisions, please use the URL below to submit the revised manuscript with figures, an updated life science reporting summary and any supplemental checklists, and a point-by-point response addressing the reviewers' criticisms.

The Reporting Summary can be found here:

The Editorial Policy Checklist can be found here: <https://www.nature.com/documents/nr-editorial-policy-checklist.pdf>

[REDACTED]

We hope to receive the revised manuscript within 2 months. Please let us know if you require additional time beyond this deadline.

With kind regards,

Ioana Visan, Ph.D.
Senior Editor
Nature Immunology

Tel: 212-726-9207
Fax: 212-696-9752
www.nature.com/ni

Author Rebuttal, first revision:

See inserted PDF

We are grateful to the Reviewers for their thoughtful comments, which we have addressed with new data or analyses, or with modifications to the text, and in doing so believe we have improved the manuscript.

Reviewer #1 (Remarks to the Author):

A. Summary of the key results

The authors analyze data of 214 SARS-CoV-2 patients of varying severity for up to one year from COVID-19 symptom onset. In patients with persistent symptoms, they document unresolved inflammation, anemia and hypoferrremia with eventual compensatory but iron-deficient reticulocytosis. They also report lymphopenia, high plasmablast numbers, low plasmacytoid DC numbers and iron deficiency in lymphocytes. They speculate that iron dyshomeostasis and delayed recovery from anemia cause inefficient oxygen transport, and persistence of symptoms, and may contribute to unresolved inflammation.

B. Originality and significance: if not novel, please include reference

This publication is comprehensive and analyzes a relatively large number -214- patients across the spectrum of severity. However, the number of blood samples 180 days or more after diagnosis is much smaller. There is little here that is new or unexpected. Others have explored similar ideas, albeit with smaller patient numbers:

The issue of persistent COVID-19 sequelae is a complex, difficult and important one, and has been the subject of a large number of studies. The works referenced below touch on aspects of what we report, but in each case do not replicate or explore the central observations that we make. We address specific instances below, but note that our study is unique in that it is to first to:

1. Collect dense biological and clinical parameters, repeatedly, from the same patients, for one year following acute COVID-19, allowing features of immunopathology and immune recovery across the full trajectory of COVID-19 to be associated with the presence, and severity, of late-stage symptoms. This approach is distinct from studies which have correlated characteristics of acute disease, or late-stage parameters alone, with outcome.
2. Provide a comprehensive picture of the physiological consequences of anaemia in COVID-19 patients, and report that delayed erythropoietic responses to early anaemia in moderate to severe COVID-19 are iron deprived, reflecting a profound breakdown in iron recirculation following hepcidin-induced iron restriction.
3. Demonstrate that early inflammatory anaemia and iron dysregulation, and in particular their haematopoietic consequences, are more strongly correlated with outcome than biological measures collected at the time of late-stage symptom reporting. This observation is both unique, and important, in that it points the way to specific early intervention strategies to correct inflammatory iron restriction, potentially reducing long-COVID incidence and severity.

Thus, this work expands substantially on our understanding of iron dysregulation and its consequences in COVID-19, and is not comparable to previous studies that report only on the prevalence of individual aspects of this phenotype.

Anemia and hypoferrremia in persistently symptomatic Covid-19 patients (references 34 and 78):

1. Sonnweber et al. The Impact of Iron Dyshomeostasis and Anaemia on Long-Term Pulmonary Recovery and Persisting Symptom Burden after COVID-19: A Prospective Observational Cohort Study *Metabolites* 2022 Jun 14;12(6):546. doi: 10.3390/metabo12060546.
2. Sonnweber et al. Persisting alterations of iron homeostasis in COVID-19 are associated with non-resolving lung pathologies and poor patients' performance: a prospective observational cohort study. *Respir Res.* 2020 Oct 21;21(1):276. doi: 10.1186/s12931-020-01546-2.

Sonnweber *et al.* (2020 and 2022) are two of many studies that have reported on the prevalence and persistence of anaemia and iron dysregulation in COVID-19 cohorts (e.g. see also PMID: 32751400, 34677368, 32517773, 33205000). Characterising this phenotype using blood-based serum iron

measurements and inflammatory parameters, they show a shift in the phenotype of anaemia and iron deficiency in recovering COVID-19 patients over time, and an association between persistently high ferritin levels and initial disease severity. We are not focussed on these well-known associations of iron dysregulation with disease severity, though we confirm them. We have instead characterized the potential cellular manifestations and systemic repercussions of iron maldistribution in the context of ongoing inflammation in COVID-19. Further, we capture the kinetics of iron dysregulation across patients ranging in disease severity, sampled from symptom onset, and show that early dysregulation of iron homeostasis is associated with subsequent persistent symptoms of disease. Sonnweber *et al.* instead correlate late-stage features of abnormal iron homeostasis and anaemia with post-COVID pulmonary recovery.

The association of persistent symptoms with chronic inflammation or immune abnormalities:

3. PHOSP-COVID Collaborative Group: Clinical characteristics with inflammation profiling of long COVID and association with 1-year recovery following hospitalisation in the UK: a prospective observational study *Lancet Respir Med.* 2022 Aug;10(8):761-775.

The PHOSP-COVID study interrogated patient-reported health outcomes at 5 and 12 months following hospitalisation with COVID-19, and correlated these with clinical baseline variables (such as sex, BMI and need for ventilation). Plasma proteome data, generated from blood taken at the 5-month visit, was compared between patients with mild or severe ongoing clinical features, with elevated levels of inflammatory mediators shown to be associated with moderate-severe long-term impairment. The study reported extensively on the prevalence of persistent symptoms, however it did not investigate kinetic changes in biological features of acute disease, or correlate these with long-term outcome. It therefore provided a descriptive picture of “long-COVID” and explored late-stage correlates of outcome, but was not designed to include the dense prospective sampling needed to determine early predictors of PASC in the way our study has done. As a result, and perhaps more importantly, the PHOSP-COVID study did not address the issue of iron homeostasis/erythropoiesis in COVID-19, which is a primary focus of our paper.

4. Peluso *et al.* Long-term SARS-CoV-2-specific immune and inflammatory responses in individuals recovering from COVID-19 with and without post-acute symptoms *Cell Rep.* 2021 Aug 10;36(6):109518. doi: 10.1016/j.celrep.2021.109518. Epub 2021 Aug 6.

Peluso *et al.* (2021) analysed samples from 70 COVID-19 patients with diverse disease presentations, first collected a median of 53 days after symptom onset. They quantified and phenotyped SARS-CoV-2 specific lymphocytes, and demonstrated their stability over time. They found differences in the frequencies of these cell populations between patients grouped by hospitalisation status, ethnicity and prior comorbidities, but no differences in T cell responses between patients who did or did not report persisting symptoms. It is unclear how this study, which focuses on the durability of the adaptive immune response to infection, overlaps with the findings of our work.

Collectively, these papers only tangentially touch on the major focus of our work (refs 1-3), or don't do so at all (ref 4). We do not believe the novelty of our work is impacted by the publications mentioned, though we will cite them and acknowledge the associations between COVID severity and both anaemia and high ferritin that they, and many others, have seen.

Page 15, Line 429:

“A growing body of literature points to a role for altered iron handling in acute COVID-19³⁵⁻³⁹, and persisting iron dysregulation has been associated with initial disease severity in convalescent patients^{36,40}” ... “Consistent with previous reports^{35,37,38,41,42}, we observed low serum iron, reduced TSAT score and hemoglobin concentrations, and concurrent raised ferritin, hepcidin and IL-6 in hospitalized COVID-19 patients from symptom onset, indicating AI in moderate-severe disease.”

Page 18, Line 535

“Iron deficiency and cerebral hypoxia have been linked to cognitive impairment and altered mood, and iron deficiency during childhood is a significant risk factor for poor cognitive performance⁸²⁻⁸³. Inflammation of the type associated with iron dysregulation (with, for example, elevated IL6 in hospitalized patients 5 months post-discharge) has also been associated with cognitive impairment following COVID-19⁵.”

C. Data & methodology: validity of approach, quality of data, quality of presentation

This is a longitudinal observational study of post-Covid symptoms, and as such cannot attribute causality, the big unanswered question in the long-Covid story. In this regard, the problem is the same as it is for other situations where anemia of chronic inflammation is observed: It is not clear how much anemia and iron sequestration contribute to symptoms. The only way to resolve this conundrum is to treat anemia and the iron disorder and document whether it improves the symptoms or not.

We did not intend to give the impression that we had demonstrated causality, something impossible in a study such as this. We have looked for sentences that may have inadvertently implied this and altered them:

Page 3: “stark” changed to “potential”

*“Through integration of diverse datasets, we reveal **potential** physiological repercussions of early inflammation-driven iron dysregulation prominent months following infection in those requiring hospitalization for COVID-19.”*

Page 6: changed “consequences of” to “association between”

*“To investigate the **association between** altered iron availability on long-term erythropoiesis, we interrogated hematological parameters.”*

Page 14:

*“This breakdown in iron homeostasis, persisting into and beyond the second fortnight of disease, feasibly compromises homeostatic responses to anemia, **potentially** resulting in low systemic oxygen carriage well beyond recovery from acute infection. The association of these abnormalities with subsequent PASC may **help** drive persistent COVID-19 sequelae and thus **may** inform clinical trial strategies for prevention or treatment of this complex phenomenon.”*

We have, however, provided what we believe to be a compelling hypothesis that warrants highlighting in the literature. Interestingly, recently published data suggests iron overload in the context of transfusion dependent β -thalassemia can protect from severe disease and mortality in SARS-CoV-2 infected individuals, consistent with a potential protective effect of high serum iron levels (PMID: 35355397). Furthermore, preliminary evidence from the IRONMAN study (PMID:36347265), a clinical trial investigating the effect of intravenous (IV) iron on cardiovascular events in heart failure patients, which coincided with the onset of the COVID-19 pandemic, suggests lower rates of COVID-19 related serious adverse events in those treated with IV iron than the usual care group. We have now highlighted these observations in the discussion, which suggest a beneficial effect of iron augmentation in reducing COVID-19 severity, and thus long-COVID symptoms, which are most frequently observed following severe acute disease.

Page 19:

“Indeed, two recent reports support a possible role for iron supplementation during COVID-19. Recently published data suggests iron overload in the context of β -thalassemia protects from severe disease and mortality in SARS-CoV-2 infected individuals⁸⁷, consistent with a potential protective effect of increased iron availability. Furthermore, preliminary reports exploring the unplanned impact of COVID-19 on patients enrolled in the IRONMAN clinical trial of intravenous ferric derisomaltose treatment in heart-failure⁸⁸ show significantly reduced COVID-19-related severe adverse events in the iron treated group (2.1%) than with usual care alone (5.3%, $p=0.007$)⁸⁹.”

As the Reviewer suggests, a clinical trial directly assessing the effect of correcting low serum iron/anaemia on long term patient outcomes might prove causality. This is something we advocate in the concluding paragraphs of our paper. The point of this work is to provide the support for exactly such studies.

The analytical methodology is up to date, including analysis of PBMC/lymphocyte subsets, key cytokines, and whole blood transcriptomics. However, the data are incomplete and do not provide much new insight as discussed below.

The analysis of IRP1 and IRP2 effects is incomplete because only transcripts are measured, even though 5'IREs exert their effects predominantly by regulating translation. This includes such genes/proteins as FTL1, FTH1, SLC40a1 (ferroportin), and EPAS1 (HIF2alpha) analyzed in line 169-177.

We agree that protein data could be informative in interpreting cellular responses to iron status via the iron response proteins (IRPs), which regulate the expression of cellular iron homeostasis genes post-transcriptionally. We have matched PBMC-derived mass spectrometry data collected on a subset of patients and controls from early disease with which we have replicated the transcript level analysis of IRE containing genes presented in Figure 3A.

We calculated protein expression fold-change in patients relative to HC at 0-14 DPSO across 150 genes in the IRE_HQ gene set (included in Supplementary Table 2). The results, now included in Supplementary Figure 12, closely resemble those seen at the transcript level, with IRE-containing genes showing strong bi-directional expression changes in severe disease. Further, calculated \log_2 fold-changes in transcript and protein expression were significantly correlated at this time. Thus, observed protein-level changes in the expression of IRE-containing genes supports our observation of strong cellular responses to altered iron status in severe COVID-19. We have updated the text to reference this additional analysis.

Page 8:

“As IRP-mediated regulation of IRE-containing genes largely occurs post-transcriptionally, we looked to validate observed mRNA expression changes using peripheral-blood mononuclear cell mass spectrometry data from a subset of matched early disease samples²⁴ (Supplementary Fig. 12). Protein level analysis confirmed the distinct bi-directional regulation of IRE-containing genes in moderate-severe COVID-19, and expression fold-changes relative to HCs derived from both datasets were significantly correlated. Thus, differential regulation of iron-homeostasis genes in moderate-severe COVID-19, likely mediated by IRPs, is detectable on both the transcript and protein level.”

Supplementary Figure 12: Validation of transcript-level expression changes in IRE_HQ gene-set genes on the protein level. (A) Distribution of \log_2FC values across 324 measured genes with high-quality conserved iron-response elements (IRE_HQ) in their 3' or 5'UTR, derived from the whole blood transcriptome comparison of COVID-19 severity groups (at 0-14 DPSO) and HC (as in Figure 3A). (B) Corresponding distribution of \log_2FC values across 150 measured proteins in the IRE_HQ gene set, taken from matched samples within the same time window. Samples from patients in groups D and E, and groups A and B, were analyzed together relative to HC to improve sample sizes ($n(HC)=8$, $n(A_B)=7$, $n(C)=5$, $n(D_E)=9$). (C) Correlation of \log_2FC values from the transcriptional comparison of group E patients and HC, and the protein level comparison of groups D+E and HC, at 0-14 DPSO. Pearson's correlation shown coefficient and P-value shown.

As the authors note, the prominence of heme metabolism transcripts is an expected consequence of reticulocytosis as these immature erythrocytes have not yet degraded their mRNAs. This is a relatively trivial conclusion.

Yes we agree this is trivial: evidence for enrichment of haem metabolism transcripts in reticulocytes was included out of thoroughness given we were using CITESeq to "deconvolute" the bulk RNA-Seq data, rather than to suggest a major new discovery. We have added the word "expected" on page 9 to make this clear.

Page 9:

"First, the heme-metabolism signature was nearly exclusively derived from a small cluster of CD71^{hi} reticulocytes (Fig. 4B, Supplementary Fig. 16A,B), consistent with the expected tight coupling of this signature to reticulocyte counts (Fig. 2H)."

Whole blood transcriptomics can only detect changes in mRNA-rich cell types that circulate in blood, hence the implication of monocytes in iron transport and lymphocytes as targets of iron deficiency is an expected consequence of their known biology (lines 200-235). The symptoms are not likely to be caused by these cell types directly.

We are not quite sure precisely what point is being made here, but agree observations made using whole blood transcriptomic data are influenced by the predominant cell types in blood, and interpretation is complicated by the cellular heterogeneity of the sample. It was for this reason we performed the CITESeq analysis described in the paper, which confirmed that signatures of dysregulated iron homeostasis observed in whole blood could be ascribed to specific cell populations.

The Reviewer might be making one or both of two points:

1. The circulating monocytes or lymphocytes are unlikely to drive disease because they are circulating. We would agree with this and have reworded the manuscript to emphasise that it is likely they reflect similar, or indeed more pronounced, dysregulation of their tissue-localised counterparts.

Page 17:

"Although blood monocytes themselves are unlikely to be the cells driving disease at tissue sites, loss of this circulating population during COVID-19, seen here and elsewhere^{17,28,70}, suggest these iron-laden cells may become resident in tissues as macrophages, where they may contribute directly to tissue damage. Iron-laden bone marrow macrophages are detectable in COVID-19 patients post-mortem⁷¹, and iron accumulation and ferroptosis in the ventricular myocardium or liver has been associated with end organ damage and thence fatal disease⁷²⁻⁷⁴."

2. That dysregulated pro-inflammatory myeloid lineage cells, or dysfunctional lymphocytes, are not related to persistent COVID-19 symptoms irrespective of anatomical localisation. We believe the involvement of these cells to be likely, given for example the profound fatigue commonly seen in people with a range of inflammatory and immune-mediated diseases. In addition, previous work

has demonstrated iron dysregulation severely compromises immune cell function, which may have pathogenic consequences stemming from poor viral control (PMID: 36197985, 33665641, 32959998, 34064487). However, we agree this remains somewhat speculative, and precise mechanisms have not been defined. We now emphasise the importance of more targeted tissue-based studies in the additional text below.

It is not clear that the severity of lymphocyte iron restriction is sufficient to affect their immunologic function, despite statements to the contrary based on much more severe restriction (lines 233-5).

We agree that this is speculation, but we believe very reasonable speculation. We have emphasised this by explaining that the functional consequences of iron restriction on lymphocytes in the context of SARS-CoV-2 infection have been explored by others.

Page 10:

“Together, multimodal single-cell analysis revealed the cells contributing to signatures of defective iron homeostasis in blood from COVID-19 patients, implicating iron sequestration in monocytes and concurrent iron deprivation of proliferating lymphocytes. To what extent this lymphocyte iron restriction compromises immunity remains to be confirmed, but in vivo studies show profound effects of hepcidin-induced iron restriction on lymphocyte primary and memory responses to vaccination and infection”²⁹”

Q1 symptoms correlated with unresolved inflammation (lines 250-258) but this is again not surprising.

The importance of this observation is not that persisting symptoms “correlated with unresolved inflammation”, but rather, COVID-19 patients reporting persisting symptoms beyond 3 months post initial disease can be differentiated by a biological signature reflecting inflammation and anaemia detectable far earlier than this (2-4 weeks). This is what differentiates our observation from those reported in previous studies of late timepoints alone, allowing suggestion of strategies to correct inflammatory anaemia early to prevent ongoing symptomatology.

The significance of “stress erythropoiesis” may be misinterpreted because of limited sampling. It is likely that iron-restricted reticulocytosis also occurs in patients who recover early from Covid-19, and it may be the delay of erythropoietic recovery marked by delayed occurrence of reticulocytosis that is an adverse prognostic marker for persistent symptoms.

A strength of this study is that its sampling is extensive enough to have provided a detailed answer to this question. When reticulocyte counts are compared between patients with either persisting COVID-19 symptoms (PS) or no persisting symptoms (NPS) and controls, reticulocyte expansion at the 30-90 day time window can be observed in both (albeit to a significantly greater degree in the PS than the NPS group; Figure 5E), possibly reflecting compensatory reticulocytosis in response to more severe initial anaemia. However, the PS group have significantly lower haemoglobin, serum iron levels and TSAT score than the NPS group at 14-30 DPSO (parameters which have resolved to within the normal HC range in the NPS group by this time, Figure 4E). Based on this data, it appears that it is not a delay in reticulocytosis that distinguishes the symptom groups, but the availability of iron for erythropoietic recovery. We have made this clearer to the reader and will show the analyses mentioned in an additional supplementary figure:

Supplementary Figure 21: Reticulocyte counts over time in patients reporting PS or NPS at Q1. P-values calculated by Wilcoxon test relative to HC (*p<0.05, **p<0.005, ***p<0.0005). Pink bar indicates the HC interquartile range. PS = persisting symptoms, NPS = no persisting symptoms.

Page 12:

“Upon direct comparison, Q1 PS individuals had a significant increase in CRP, IL-6 and IL-1 β , and decrease in serum iron, haemoglobin and TSAT score relative those with NPS at 15-30 DPSO (Fig. 5E; Supplementary Fig. 20). Reticulocyte counts were elevated above HC levels at 31-90 DPSO in both PS and NPS groups (Supplementary Fig. 21), corresponding with the peak of the stress-erythropoietic response reported above, though counts were significantly higher in the PS than the NPS group at this time (Fig. 5E, Supplementary Fig. 20). This suggests that iron availability for erythropoietic recovery, rather than delayed reticulocytosis, is an early differentiator of long-term symptoms groups.”

D. Appropriate use of statistics and treatment of uncertainties; no concerns

E. Conclusions: robustness, validity, reliability: no concerns

F. Suggested improvements: experiments, data for possible revision

Line 160 “yet accumulated cytosolic free iron is readily converted to reactive oxygen species (ROS), ...” Iron catalyzes the conversion but it is not “converted”.

This wording is fixed in the revised version of the manuscript.

Page 7:

“Iron is essential for cellular respiration and metabolism, yet accumulated cytosolic free iron catalyses the production of reactive oxygen species (ROS), contributing to lipid membrane peroxidation and cell death (termed ferroptosis).”

G. References: appropriate credit to previous work?—Yes

H. Clarity and context: lucidity of abstract/summary, appropriateness of abstract, introduction and conclusions

Figure 1EF could be made easier to see by using better color combinations

We have changed the shade of the pink band showing the IQR in this and other figures to improve contrast. The colour scheme used for severity groups (A-E) is used for continuity with the earlier published characterisation of this cohort (PMID: 34051148).

Reviewer #2 (Remarks to the Author):

In this study Hanson and colleagues tried to identify early markers of post-acute sequelae of COVID-19 (PASC). Mainly based on investigations on blood immune cell composition, inflammatory soluble factors including parameters of iron metabolism and gene expression studies in the early acute infection period, comparing samples from patients with different severity, they postulate that increased immune cell activation and iron-deficient reticulocytosis are indicative of PACS. PACS represents an important clinical problem and our understanding of pathomechanisms contributing to PACS development is sparse. Thus, well-designed studies revealing such pathomechanisms are urgently needed.

In the present study appropriate state-of-the-art multi-omics-technologies were applied generating high quality data. However, the study design is not appropriate which weakens the obtained findings quit severely.

We will address the specific questions raised initially, and then the issue of “study design” in the context of a real-world pandemic at the end.

Main concern:

The authors start off with studying the normalization (in comparison to value of uninfected controls) of blood immune cell markers of their previously published patient groups (HC versus A to E). Although the age of the patients is not provided in the present study, we know from the previous published cohort that controls as well as group A and B patients were much younger in comparison to hospitalized patient groups C to E. It is well known that certain inflammatory markers such as IL-6 increase and iron, hemoglobin levels decrease with aging. Thus, a direct comparison to the same controls is not correct.

We present age distribution data in Figure 1B, and highlight it in the first paragraph of the results section. We have now further emphasised the issue of age and, for ease of reference, we will include in the Supplementary a modified version of Supplementary Table S1 from our first published description of this cohort (PMID: 34051148) showing the demographics of all patients groups and HCs in more detail.

Page 4:

“The demographics of this cohort have been described¹⁷ and are presented in Supplementary Table 1.”

*“Sampling density and cohort demographics are shown in **Fig. 1A-C** and **Supplementary Table 1**, and patient timelines in **Supplementary Fig. 1**. Patients in group C, D and E were older than in A and B, and more frequently men (**Fig. 1B,C**).”*

*“Patients in group C, D and E were older than in A and B, and more frequently men, in part as a result of known associations with disease severity in COVID-19, and in part due to recruitment constraints during the peak of the pandemic (**Fig. 1B,C**). Conclusions are therefore drawn only after carefully statistical correction for age and sex.”*

Unfortunately, this age bias affects all further analyses, which in my view questions the whole study. This also concerns the most interesting analysis of identifying PACS discriminators in figure 5 and 6. The non-PACS group is heavily dominated by the non-hospitalized group B patients, who of course are much younger compared to the hospitalized group C to E patients. Nearly none of the non-hospitalized patients (group B) developed PACS.

Like many others, we find that those hospitalised with moderate-severe COVID-19 tend to be older on average than the healthy population (Figure 1B), and thus control groups are often not well matched. This is an important issue, however we have taken great efforts to ensure our results and conclusions are not impacted by this potential bias. It appears this was not made clear enough in the manuscript, and we have now rectified this with additional text and supplementary figures confirming that age is not driving the observed differences between study groups we report.

1. We have incorporated an additional supplementary figure (below) which shows all immune cell comparisons between COVID-19 severity groups and healthy controls both with values either corrected for age and sex (red stars) or not (black stars). The former mirror very closely the statistical results derived from non-age-corrected comparisons, and do not change the interpretation of presented findings. Both sets of results are presented in the current Supplementary Figure 2, with Supplementary Figure 3 showing extended cellular data.

Supplementary Figure 2: Heatmap showing log₂ fold-change in absolute cell counts between COVID-19 patients and HC, across severity groups and within time bins. P-values are calculated by T-test comparison of log₂(count + 1) transformed data between test COVID-19 group and HCs (*p<0.05, **p<0.005, ***p<0.0005). Asterisks in red indicate p-value from a linear model of group effect on log₂(count + 1) with age and sex correction.

Page 5:

“These findings remained statistically significant after correction for age and sex (Supplementary Fig. 2 and 3) and demonstrate prolonged immunological disruption following moderate-to-severe COVID-19.”

- We have incorporated an additional supplementary figure (below) showing serum inflammatory and iron parameter comparisons between study groups with and without age correction. Again,

our finding are robust to age correction, and this does not change the interpretation of the presented data.

Supplementary Figure 5: Heatmap showing log₂ fold-change in median serum inflammatory, iron and erythroid cell parameters between COVID-19 patients and HC (or group A and B samples taken at >90 DPO in the absence of HC measures for ferritin and hemoglobin), across severity groups and within time bins. P-values are calculated by Wilcoxon test (COVID-19 vs HC), black asterisks, or from a linear model of group effect on log₂(measure) with age and sex correction, red asterisks; *p<0.05, **p<0.005, *p<0.0005.**

Page 6:

“Age and sex differences between patient groups and HC did not account for persisting differences in inflammatory and iron-related parameters (Supplementary Fig. 5).”

3. Transcriptomic analysis was originally performed with adjustment for age and sex as covariates, and so we have made no further changes, apart from now highlighting this adjustment in the main text on page 7. The formula used for differential expression in R notation is as below, and is now included in the methods:

```
design <- model.matrix(~0+group+age+sex+extractbatch)
```

Where group specifies a composite variable capturing patient severity group and time window. These estimates thus represent the difference in gene expression that remain after accounting for the differences in expression that were due to these intrinsic factors.

Page 7:

“All expression comparisons across gene-sets of interest were corrected for age and sex differences.”

4. It is correct that older age, moderate-severe COVID patients are over-represented in those who develop PASC, and compared to a younger “non-PASC” group in Figure 5. However, age does not appear to impact PASC risk beyond being a close correlate of acute disease severity. When assessed within each disease severity group (B-E) there is no difference in age between those who do or do not go on to develop PASC, as we now show in Supplementary Figure 18 and discuss in the text (below).

Supplementary Figure 18: Age comparison between individuals with PS or NPS at Q1, stratified by initial disease severity group. P-values calculated by T-test. PS = persisting symptoms, NPS = no persisting symptoms.

Page 11:

“PS was more frequent in the hospitalized than mild disease groups (C-E versus B), and thus the PS patients were older than those with NPS, (statistically significantly so at Q1). However, age did not differ between PS and NPS groups when patients were stratified by initial disease severity (Supplementary Fig. 17 and 18)”

5. Formal age correction of the data presented in Figure 5E is now included in Supplementary Figure 20 (below). This shows that CRP and IL-6 levels are still statistically significantly higher in the persisting symptom group over the non-persisting symptom group at 90-180 DPSO, iron abnormalities remain significant at 15-30 DPSO and increased reticulocyte counts are still detectable at 31-90 DPSO as reported in the text. As mentioned above, RNASeq analyses had already included correction for age and sex as covariates when comparing “long-COVID” symptom groups, and so presented results are robust to these differences.

Supplementary Figure 20: Heatmap showing \log_2 fold-change in serum inflammatory and iron measures, reticulocyte parameters and absolute immune cell counts between PS and NPS groups, across severity groups and within time windows. Symptom groupings from 0-180 DPSO are derived from Q1 questionnaire responses, from 180- 360 DPSO from Q2 responses. Black asterisks represent significance as calculated by T-test of \log_2 transformed measures (PS versus NPS), as in Figure 5E. **Asterisks in red indicate p-value from a linear model of group effect on \log_2 transformed measures with age correction.** All p-value are shown as $\dagger p < 0.1$, $* p < 0.05$, $** p < 0.005$, $*** p < 0.0005$. PS = persisting symptoms, NPS = no persisting symptoms.

Thus, in conclusion such an analysis can only be performed in well age- and sex-matched, much larger cohorts including appropriate controls.

Unfortunately studies done at the height of a pandemic are very much “real world” studies, and not clinical trials. This is one of the largest and most comprehensive COVID-19 studies generated at this time. Samples were collected from sick people who generously agreed to be research subjects, by busy staff and volunteers putting themselves at real personal risk. Thus, perfect age-matching was not possible: hospitalised patients were on average older than asymptomatic and un-infected controls recruited from hospital staff. Lockdowns prevented recruitment of age matched healthy individuals from the wider community. We were the first to introduce routine staff screening in the UK (PMID:32392129), a service which enabled this research to be undertaken from the very early days of the pandemic. If we are to learn about the sequelae of acute COVID-19 we have to make the most of these real-world studies and so we have applied the best statistical correction methodology we can to maximise the use of this valuable and unique dataset. In response to the Reviewers quite reasonable concerns about this, we have therefore emphasised the importance of this issue and provided more comprehensive age-adjusted data in the Supplementary material. We believe this addressed the age issue robustly, and that we should not make the perfect the enemy of the good, and in doing so throw out such hard-won and potentially important clinical observations.

Decision Letter, second revision:

24th Jul 2023

Dear Dr. Hanson,

Your Article, "Iron Dysregulation and Inflammatory Stress Erythropoiesis Associates with Long-Term Outcome of COVID-19" has now been seen by 3 referees. While the work is of potential interest, the reviewers have raised substantial concerns that must be addressed. As such, we cannot accept the current manuscript for publication, but would be interested in considering a revised version that addresses these serious concerns.

Should you find yourself able to thoroughly address the referees' concerns, please let me know. We believe it is essential to perform the analysis suggested by referee 3 (and address the concerns from referee 2) in order to assess whether altered iron homeostasis is due to PASC or age and severity of disease. If your conclusions stand, and the novelty of the paper has not been compromised in the interim, we would invite you to submit a revised paper. At resubmission, please include a "Response to referees" detailing, point-by-point, how you addressed each referee comment. This response will be sent back to the referees along with the revised manuscript.

In addition, please include a revised version of any required reporting checklist. It will be available to referees (and, potentially, statisticians) to aid in their evaluation if the manuscript goes back for peer review. A revised checklist is essential for re-review of the paper. The Reporting Summary can be found here:

When submitting the revised version of your manuscript, please pay close attention to our [href="https://www.nature.com/nature-portfolio/editorial-policies/image-integrity">Digital Image Integrity Guidelines. and to the following points below:](https://www.nature.com/nature-portfolio/editorial-policies/image-integrity)

[REDACTED]

We would hope to receive the revised manuscript within 2 months. If you cannot send it within this time, please let us know. We will be happy to consider your revision so long as nothing similar has

been accepted for publication at Nature Immunology or published elsewhere.

Nature Immunology is committed to improving transparency in authorship. As part of our efforts in this direction, we are now requesting that all authors identified as 'corresponding author' on published papers create and link their Open Researcher and Contributor Identifier (ORCID) with their account on the Manuscript Tracking System (MTS), prior to acceptance. ORCID helps the scientific community achieve unambiguous attribution of all scholarly contributions. You can create and link your ORCID from the home page of the MTS by clicking on 'Modify my Springer Nature account'. For more information please visit www.springernature.com/orcid.

Thank you for the opportunity to review your work.

Sincerely,

Ioana Visan, Ph.D.
Senior Editor
Nature Immunology

Tel: 212-726-9207
Fax: 212-696-9752
www.nature.com/ni

Reviewers' Comments:

Reviewer #1:

Remarks to the Author:

The authors' response to the reviewers' concerns was informative and helpful. The study makes a useful contribution to understanding the pathophysiology of long COVID, to which unresolved inflammation, prolonged anemia and systemic iron dysregulation may indeed contribute. The persistence of inflammation, iron abnormalities and anemia in the period 15-30 days after symptom onset appears to differentiate those patients who develop prolonged symptoms from those who don't. It will take larger studies to confirm the predictive power of this observation but the pandemic is fortunately waning so we will have to query the generalizability of this observation to other infections. Whether early treatment of iron abnormalities can prevent persistent symptoms is also an open question.

Reviewer #2:

Remarks to the Author:

Previous concerns:

The authors start off with studying the normalization (in comparison to value of uninfected controls)

of blood immune cell markers of their previously published patient groups (HC versus A to E). Although the age of the patients is not provided in the present study, we know from the previous published cohort that controls as well as group A and B patients were much younger in comparison to hospitalized patient groups C to E. It is well known that certain inflammatory markers such as IL-6 increase and iron, hemoglobin levels decrease with aging. Thus, a direct comparison to the same controls is not correct.

Unfortunately, this age bias affects all further analyses, which in my view questions the whole study. This also concerns the most interesting analysis of identifying PASC discriminators in figure 5 and 6. The non-PACS group is heavily dominated by the non-hospitalized group B patients, who of course are much younger compared to the hospitalized group C to E patients. Nearly none of the nonhospitalized patients (group B) developed PASC.

Authors reply to comments above:

We present age distribution data in Figure 1B, and highlight it in the first paragraph of the results section. We have now further emphasised the issue of age and, for ease of reference, we will include in the Supplementary a modified version of Supplementary Table S1 from our first published description of this cohort (PMID: 34051148) showing the demographics of all patient groups and HCs in more detail.

Reply by reviewer:

To me the issue of age as a contributor of the PASC signature is insufficiently addressed. It will be important to dissect, whether age and age-related alterations or altered iron-homeostasis independent of age is a driving factor of persisting symptoms and PASC.

First, no information on age of the controls is provided within Supplementary Table S1.

Authors reply to comments above:

Like many others, we find that those hospitalised with moderate-severe COVID-19 tend to be older on average than the healthy population (Figure 1B), and thus control groups are often not well matched. This is an important issue, however we have taken great efforts to ensure our results and conclusions are not impacted by this potential bias. It appears this was not made clear enough in the manuscript, and we have now rectified this with additional text and supplementary figures confirming that age is not driving the observed differences between study groups we report.

Reply by reviewer:

Certainly, hospitalised, and moderate-severe COVID-19 patients tend to be older than non-hospitalised patients. However, I disagree with the statement that this automatically applies also to the controls. Controls can and should be always selected according to the patient population studied.

Authors reply to comments above:

1. We have incorporated an additional supplementary figure (below) which shows all immune cell comparisons between COVID-19 severity groups and healthy controls both with values either corrected for age and sex (red stars) or not (black stars). The former mirror very closely the statistical results derived from non-age-corrected comparisons, and do not change the interpretation of presented findings. Both sets of results are presented in the current Supplementary Figure 2, with Supplementary Figure 3 showing extended cellular data.

Reply by reviewer:

The authors used a linear model for the age correction, which can only be applied if a linear relationship between age and the tested parameter exist. This need to be proven first. In addition, from the modified heatmaps shown in Supplementary Figures 2, 3 and 5 clearly shows that the significance for some important parameters such as pDCs, iron or ferritin are lost, especially at the late timepoints!

Authors reply to comments above:

4. It is correct that older age, moderate-severe COVID patients are over-represented in those who develop PASC, and compared to a younger "non-PASC" group in Figure 5. However, age does not appear to impact PASC risk beyond being a close correlate of acute disease severity. When assessed within each disease severity group (B-E) there is no difference in age between those who do or do not go on to develop PASC, as we now show in Supplementary Figure 18 and discuss in the text (below).

Reply by reviewer:

Indeed, suppl. figure 18 shows that when assessing for the original disease severity group no differences in age could be observed. However, the same figure also nicely highlights the age difference between the severity groups and that the proportion of PS patients within the "young" disease group B is much lower. Why don't the authors simply show that a direct age comparison of PS and NPS patients over all severity groups?

Authors reply to comments above:

5. Formal age correction of the data presented in Figure 5E is now included in Supplementary Figure 20 (below). This shows that CRP and IL-6 levels are still statistically significantly higher in the persisting symptom group over the non-persisting symptom group at 90-180 DPO, iron abnormalities remain significant at 15-30 DPO and increased reticulocyte counts are still detectable at 31-90 DPO as reported in the text. As mentioned above, RNASeq analyses had already included correction for age and sex as covariates when comparing "long-COVID" symptom groups, and so presented results are robust to these differences.

Reply by reviewer:

This is the most important change in the manuscript. This shows that the significance is reduced or even gets lost for nearly all the discriminating parameters. Although, the significance for CRP and IL-6 is only reduced, other differences such as IL-1b, HGB and iron are lost completely throughout or at later timepoints which are relevant for PASC diagnosis. In addition, as stated before the age correction was performed assuming a linear relationship.

Thus, I would urge to run the following comparisons:

- 1) Run the PS versus NPS comparisons for severity groups C and D for which a decent number of both PS and NPS patients exists, and which are nicely age balanced.
- 2) Run the PS versus NPS comparisons for different age groups.
- 3) Correct for severity.

Reviewer #3:

Remarks to the Author:

In my review I only address the point if co-founders (most importantly: age, severity) were adequately addressed statistically. The main issue in this analysis is that a large proportion of the non-PASC patients are much younger, and they are patients that had lower degrees of the disease (A,B vs.

C-E), Moreover, the groups have different sex distributions (see figure 1).

To dissect if e.g. the altered iron homeostasis contributes to PASC or is simply a function of age and severity of the patient cohort, the authors in their revision use linear modelling. The authors don't provide a clear methods description (e.g. how did they encode their variables? Is age a number, or is it a factor?). If (what I assume) the age is encoded as a number, this "correction" would only be appropriate if the effect of age is indeed linear, which I doubt. If it is non-linear, then unevenness in the age distributions will result in significant associations, even when there is none. The (most simple, yet not perfect) way to overcome this would be to group patients into small age bins and then perform linear modelling, where a parameter is estimated for each age group.

A much more intuitive and more appropriate way to analyze this data is to test the associations in sub-groups groups that are homogenous in age, disease severity and sex. So for instance, take each severity group with sufficient number of PASC, make sure that age and sex are comparable, and then test differences between PASC and non-PASC this within each group.

It would also be very beneficial to plot the measured parameter (such as iron) against age as a scatter plot for each disease severity, and mark PASC and non-PASC patients.

To conclude, the current analysis is not suited to dissect if the associations observed are mainly due to cofounders, or if they are associated with PASC.

Further review of the statistical methods was hindered as the methods are not fully disclosed (see e.g. above), and the data was not available for analysis, as the link to zonodo provided required a name and email for excess, preventing anonymous peer review. Data analysis code was not available.

Author rebuttal, second revision:

See inserted PDF

Black = Reviewer comments, Blue = Author Comments, Purple = Changed text, Grey = Existing text

Reviewer #1 (Remarks to the Author):

The authors' response to the reviewers' concerns was informative and helpful. The study makes a useful contribution to understanding the pathophysiology of long COVID, to which unresolved inflammation, prolonged anemia and systemic iron dysregulation may indeed contribute. The persistence of inflammation, iron abnormalities and anemia in the period 15-30 days after symptom onset appears to differentiate those patients who develop prolonged symptoms from those who don't. It will take larger studies to confirm the predictive power of this observation but the pandemic is fortunately waning so we will have to query the generalizability of this observation to other infections. Whether early treatment of iron abnormalities can prevent persistent symptoms is also an open question.

We thank Reviewer 1 for their response and agree that larger and more targeted studies will be required to confirm whether, and the extent to which, early inflammatory iron dysregulation may predict long-term outcomes. We believe it unlikely that these observed disruptions to iron homeostasis are unique to SARS-CoV-2 infection. It is hope this study will also prompt further research into the clinical efficacy of treating inflammatory anaemia and poor iron recirculation in other infectious diseases for which long-term sequelae are a common or suspected complication.

Reviewer #2 (Remarks to the Author):

The authors start off with studying the normalization (in comparison to value of uninfected controls) of blood immune cell markers of their previously published patient groups (HC versus A to E). Although the age of the patients is not provided in the present study, we know from the previous published cohort that controls as well as group A and B patients were much younger in comparison to hospitalized patient groups C to E. It is well known that certain inflammatory markers such as IL-6 increase and iron, hemoglobin levels decrease with aging. Thus, a direct comparison to the same controls is not correct.

To me the issue of age as a contributor of the PASC signature is insufficiently addressed. It will be important to dissect, whether age and age-related alterations or altered iron-homeostasis independent of age is a driving factor of persisting symptoms and PASC.

First, no information on age of the controls is provided within Supplementary Table S1.

We have now rectified this. Supplementary Table 1 now includes the demographics (age and sex) of the 45 healthy controls used in all statistical comparisons of clinical parameters and absolute cell counts, and the 60 healthy controls used in the bulk RNASeq analyses.

Certainly, hospitalised, and moderate-severe COVID-19 patients tend to be older than non-hospitalised patients. However, I disagree with the statement that this automatically applies also to the controls. Controls can and should be always selected according to the patient population studied.

We detailed in our previous response to reviewers the insurmountable practical difficulties which hindered the precise age matching of healthy controls to this real-world COVID-19 cohort at the onset of a global pandemic, but perhaps did not give this sufficient emphasis in the text. Under these circumstances it would have been both unethical and illegal to have approached isolating age- and gender-matched controls during a national lock-down, putting them at risk of infection. We have included additional text in the methods to emphasise why our cohort demographics are biased in this way, and the steps taken to interrogate and address potential consequences of this.

“Supplementary Table 1 details the demographics and baseline clinical features of COVID-19 severity groups and HCs. Owing to the difficulty of accessing and recruiting matched, uninfected population controls during the pandemic, due to the risk of spreading infection, staff shortages and the need to comply with lockdown restrictions, HC samples were predominantly collected from uninfected healthcare workers. Consequently, the demographic of this population could not be well matched to that of hospitalized COVID-19 patients, who were of significantly older age. As described in the statistical methods below, we have thoroughly assessed the potential consequences of this age bias in our cohort, and validated our age-corrected findings with sensitivity analyses to ensure they are robust.”

We detail below the measures used to ensure this age bias has not impacted on the conclusions we have drawn in this study. We have now included a number of these analyses, alongside the additional sensitivity analyses requested by the reviewers, in the supplementary materials. To make it clearer to readers that age (and sex) effects have been given due consideration, we have now presented the age and sex corrected findings for all parameters in the primary figures, rather than the supplementary, and adjust the text accordingly. Details of these changes, and validation of our method of age correction, is provided below.

The authors used a linear model for the age correction, which can only be applied if a linear relationship between age and the tested parameter exist. This need to be proven first.

We have performed additional analyses to address the concerns of Reviewers 2 and 3 relating to the presence of an association between age and the serum and cellular parameters measured in our cohort, and the nature of this relationship where it is detected. We would like to emphasise two important points:

1. Of the 47 clinical and cellular parameters presented in Figures 1D and 2A, 13 show evidence of age-dependent changes in this cohort, which ranges from 17-89 years: IL-6, IL-10, immature reticulocyte fraction, CD4 T cells (naïve and activated and activated:naïve ratio), CD8 T cells (absolute counts, naïve counts and activated:naïve ratio), gd T cells (total, Vg9+Vd2^{hi} and ^{lo} subsets) and MAIT cells. These associations were observed in HC and/or in HCs + late-stage COVID-19 patient samples collected >180 DPSO, with correction for acute disease severity. We include for reviewers an additional Supplementary Methods (SM) document with the results of these tests for age-association, and figures showing age regressed data across COVID-19 severity groups during early (0-14 days post symptom onset; DPSO) and late (final sampling timepoint >180 DPSO) stage disease (Figure SM1-SM2, Table SM1-SM2). Shown below are examples for four parameters of interest (CRP, serum iron, haemoglobin and pDCs). It is evident that age is not significantly associated with these parameters in HCs, nor in COVID-19 patients at the final timepoint of sampling. Age effects seen during acute disease (0-14 DPSO) are very clearly driven *via* the confounding of age with disease severity and are not acting independently. Accordingly, we see that age is not significantly associated with clinical parameters (with the exception of those with age associations listed above) at this early timepoint when disease severity group is corrected for in linear regression (see Table SM3-SM4 for per-severity group tests).

2. Where age associations *do* exist with clinical parameters, we can find no strong evidence that these relationships cannot be appropriately approximated by linear models (see examples below and Figures SM1-SM2) and corrected accordingly. We further tested for non-linearity with inclusion of a quadratic parameter for age in the regression of age against each normalised (log transformed) clinical parameter in HCs.

$$\text{lm}(\log_2(\text{parameter}) \sim \text{age} + \text{age}^2)$$

The quadratic term was insignificant for all parameters. The Supplementary Methods document includes the results of the performed statistical tests for non-linear age effects, with age treated as either a linear or quadratic predictor in HC (Tables SM1-SM2), and within the test for COVID-19 severity group effects (Table SM3-SM4). We acknowledge that for some parameters, within some severity groups at timepoints where observations are sparse, we are underpowered to test the appropriateness of either linear or non-linear fits of increasing complexity for age. However, based on our best available models in HCs and amalgamated severity groups (as shown) a linear fit is suitable to approximate existing age effects, thus we have continued to correct for age in this fashion.

The Reviewers' concern that age discrepancies may account for differences between COVID-19 severity groups and HCs, particularly at late timepoints, and between long-COVID symptom groups (as discussed below), is understandable. However, these analyses show that age does not have a significant independent effect on the majority of tested parameters in this study. Importantly, we find no evidence that inflammatory and iron related parameters towards which we draw the most focus (namely CRP, iron measures, haemoglobin, reticulocyte count and reticulocyte haemoglobin) are strongly influenced by age in this cohort. In the few instances where age associations are detected, inclusion of age as a covariate in the linear model of severity group effect appears appropriate to account for this bias. Based on the analyses above, we are confident that the biological conclusions drawn from our findings are accurately informed by observed perturbations owing to SARS-CoV-2 infection, and not underlying age disparities. This is further confirmed in the sensitivity analyses below.

As the Reviewer states, it is always desirable to match controls precisely to studied populations. In practice, this is difficult in an observational study which spans study groups of hugely variable phenotype and underlying risk demographic (asymptomatic to fatal COVID-19), without drastically pruning valuable observations. Across 5 severity groups and 5-6 time windows we have up to 30 unique group comparisons with HCs. To have 30 independently matched subsets of HCs would be an elaborate response to this problem given we have shown our present statistical approach of age correction is a valid way to utilise all available data. We have now included the following text in the results and methods to justify this choice:

Page 4, Line 87:

"Patients in group C, D and E were older than in A and B, and HC, and more frequently men, in part as a result of known associations with disease severity in COVID-19, and in part due to recruitment constraints during the peak of the pandemic (Fig. 1B,C, see methods for details). Conclusions are therefore drawn only after statistical correction for age and sex."

Page 40, Line 873:

“The validity of using age as a linear covariate for age-bias correction was tested by assessing the nature of age associations with clinical and cellular parameters in HCs, and in COVID-19 severity group samples taken >180 DPSO (**Supplementary Methods**). Only 13 of 47 measured parameters showed evidence of an association with age, including IL-6, IL-10, IRF, CD4 T cells (naïve, activated and naïve:activated ratio), CD8 T cells (absolute counts, naïve and activated:naïve ratio), gd T cells (total, Vg9+Vd2^{hi} and ^{lo}) and MAIT cells. For all these parameters, age effects could be effectively modelled using a linear covariate in HCs and COVID-19 patients. Differences detected between PS and NPS groups (which also varied in age) were confirmed by Wilcoxon test in various matched subsamples of the cohort, including age matched participants, and age and acute disease severity matched participants (with exclusion of group B patients, overrepresented in the NPS group, from analyses).”

As stated above, age corrected data is now presented in the main figures. Replacement panels for Figures 1D-F and 2A-E are shown below. Supplementary figures showing individual-level cellular and clinical data (now Supplementary Figures 2-4) now show age corrected p-values only.

Replacement panels for Figure 1D,E and F showing COVID-19 severity group effects tested using linear modelling with age correction (appropriately formatted Figure 1 can be seen in the revised manuscript):

Replacement panels for Figure 2A-E showing COVID-19 severity group effects tested using linear models with age correction (appropriately formatted Figure 2 can be seen in the revised manuscript):

In addition, from the modified heatmaps shown in Supplementary Figures 2, 3 and 5 clearly shows that the significance for some important parameters such as pDCs, iron or ferritin are lost, especially at the late timepoints!

The age and sex corrected data presented above still support the central observations reported. We still conclude that immunological disruptions following moderate-severe COVID-19 are prolonged and slow resolving and that iron homeostasis is perturbed from early disease, disrupting the resolution of early anaemia.

With age correction our data show:

- Elevated levels of inflammatory markers resolve slowly, some remaining raised for up to one year following severe disease.
- MAIT cells, gd T cells and pDCs are the immune populations that remain depleted in the periphery for the longest period of time; ongoing high-plasmablast counts and elevated activated:native CD4/CD8 T cell ratio in severe disease suggest ongoing immune activation
- Consistent with anaemia of inflammation we see reduced circulating iron availability for erythropoiesis, low haemoglobin levels for up to and beyond 3 months in moderate-severe disease, and iron-deprived reticulocytosis peaking at 30-90 DPSO

The focus of this study was not to interrogate those specific parameters that remain perturbed at very late timepoints following acute COVID-19. It was to characterise the nature and consequences of early and persisting inflammatory iron-dysregulation, and explore whether variation in clinical and immunological parameters measured in early disease associate with long-term outcome.

We have changed these specific points of the text to reference the replacement figures included above:

Page 4, Line 98:

*“Resolution of early T and B cell lymphopenia in hospitalized patients was apparent by 15-30 DPO in group C, but delayed in groups D and E. MAIT cells, $V\gamma 9V\delta 2^{hi}$ T cells and pDCs were slowest to recover, with counts ~~were remaining~~ lower than in HCs beyond 90 DPO, ~~and low plasmacytoid dendritic cells (pDC) failed to recover~~ (Fig. 1E). High plasmablast counts persisted for ~~up to six months~~ at least 90 DPO in all groups, and an increase ratio of activated:naïve CD4 and CD8 T cells remained pronounced in hospitalized patients ~~groups~~, to one year in group E (Fig. 1F). ~~These findings remained statistically significant after correction for age and sex.~~ **Supplementary Fig. 2** shows patient-level data for all measured parameters in greater detail. Collectively, longitudinal immune-cell profiling demonstrated prolonged immunological disruption following moderate-to-severe COVID-19.”*

Page 5, Line 112:

*“Raised levels of C-reactive protein (CRP) and cytokines (IL-6, IL-10, IL-1 β and TNF) resolved slowly in groups C-E, ~~with IL-6 with elevated concentrations of some inflammatory cytokines, most markedly IL-6, remaining elevated above HC levels~~ still detectable at 270-360 DPO (Fig. 2A, **Supplementary Fig. 3**).”*

Indeed, suppl. figure 18 shows that when assessing for the original disease severity group no differences in age could be observed. However, the same figure also nicely highlights the age difference between the severity groups and that the proportion of PS patients within the “young” disease group B is much lower. Why don't the authors simply show that a direct age comparison of PS and NPS patients over all severity groups?

We did show such a direct age comparison – it was presented in Supplementary Figure 17G (now 15G), with the age discrepancy emphasised in the text (as below). Note that this figure also shows that the long-term symptom groups are matched for sex, and so sex has not been corrected for in PASC group analyses. We have now emphasised this:

Page 11, Line 265

*“PS were more frequent in the hospitalized than mild disease groups (C-E versus B), and thus the PS patients were older than those with NPS, (statistically significantly so at Q1). However, age did not differ between PS and NPS groups when patients were stratified by initial disease severity (**Supplementary Fig. 15** and **16**), suggesting that age is indirectly associated with PASC only via an association with acute disease severity. The sex distribution of PS and NPS groups was matched at both questionnaire timepoints.”*

This is the most important change in the manuscript. This shows that the significance is reduced or even gets lost for nearly all the discriminating parameters. Although, the significance for CRP and IL-6 is only reduced, other differences such as IL-1b, HGB and iron are lost completely throughout or at later timepoints which are relevant for PASC diagnosis. In addition, as stated before the age correction was performed assuming a linear relationship.

The reviewer is referring to the age correction performed in the re-analysis of the data presented in Figure 5E, comparing individuals with persisting symptoms (PS) or no persisting symptoms (NPS) of COVID-19. We emphasise some points regarding this figure and the outcomes of age correction:

1. Figure 5 and the associated text is concerned with describing a multivariate signature present in early disease which best discriminates individuals who go on to report PASC symptoms much later. We show that features of persisting inflammatory iron dysregulation form part of this signature, a homeostatic disruption potentially relevant for long-term recovery. We do not attempt to identify any single parameter that is predictive of PASC, and given that PASC likely comprises a highly heterogenous collection of underlying clinical pathologies, do not have the

power to dissect this complexity. We also highlight a number of parameters (CRP, IL6 and serum iron) which we observe to remain statistically different between PS and NPS symptom groups at the first timepoint of PASC symptom reporting (90-180 DPSO).

2. The reviewer is concerned that age may be driving differences in these groups. As we have shown above, age does not significantly influence the clinical parameters shown in Figure 5E with the exception of immature reticulocyte fraction and IL-6. Also shown above, detected age associations can be modelled linearly, and so age correction using a linear predictor is a valid way to account for age effects where they exist. Below are results from three analyses of the same data:

- A. The comparison of PS vs NPS groups by Wilcoxon test (as presented in the original Figure 5E)
- B. The comparison of PS vs NPS groups that have been age matched via the exclusion of individuals <30 years of age.
- C. The symptom group effect as tested using linear models with age correction:
$$\text{lm}(\log_2(\text{measure}) \sim \text{symptom group} + \text{age})$$

As shown in the figures below, p-values for some parameters fluctuate slightly about arbitrary significance thresholds. This is to some extent expected in analysis B given that there is a large degree of variation about the group mean for measured clinical parameters in the early weeks-month of disease. Age, although not independently associated with most clinical parameters shown, is confounded with both initial disease severity, and with PASC group *via* initial disease severity, capturing a portion of the variability explained by the primary predictor in analysis C when included as a covariate. Irrespective of this, lower serum iron and TSAT at 14-30 DPSO remain significantly associated with persisting symptoms at 3-5 months in all analyses. Haemoglobin levels are also lower at this time, though above the threshold for significance in the age corrected linear model ($p = 0.055$). All analyses also show an increase in reticulocyte number at 30-90 DPSO in the PS group, indicative of a higher degree of stress erythropoiesis in response to inflammatory anaemia. The RNA sequencing data, an entirely independent data modality, consolidates that disruptions to iron homeostasis and hypoxic stress are early features of the long-term PS group (Figure 5F and G). Further, both IL-6 and CRP remain significantly higher in the PS relative to the NPS group at 90-180 DPSO in both age matched and age corrected analyses. We find this to be collective evidence that both early and late signatures of iron dysregulation and inflammation associated with PASC are detectable, biological interpretation of our data is valid, and these findings are not being driven by age disparities between study groups.

For continuity with the first portion of the paper, we have presented the results from the linear model with age correction in Figure 5E and adjusted the text accordingly. We have also referenced the age matched sensitivity analysis, which is now included in the supplementary material (Supplementary Figure 19):

*** p<0.0005, **p<0.005, *p<0.05

Page 12, Line 284

“Upon ~~direct comparison~~ testing of PASC symptom group effects by multivariate linear regression with age correction, Q1 PS individuals had a significant increase in CRP, IL-6 and IL-1B, and a decrease in serum iron, hemoglobin and TSAT score significantly lower TSAT and serum iron than the NPS group at 15-30 DPSO relative to those with NPS at 15-30 DPSO (Fig. 5E).”

...

“To further confirm that PS and NPS group differences could not be accounted for by underlying age differences, analyses were repeated in a subset of age matched patients. Serum iron, TSAT and hemoglobin levels remained significantly lower at 14-30 DPSO in those reporting PS at Q1, reticulocyte counts were significantly higher at 30-90 DPSO, and IL-6 and CRP were significantly elevated at 90-180 DPSO (Supplementary Figure 19A-B).”

Thus, I would urge to run the following comparisons:

1) Run the PS versus NPS comparisons for severity groups C and D for which a decent number of both PS and NPS patients exists, and which are nicely age balanced.

We have run an age balanced analysis above by exclusion of individuals <30 years of age, which were disproportionately classified as having NPS. As shown above, and now presented in Supplementary Figure 19, our original observations are robust to this subsampling. This data shows evidence of low circulating iron and anaemia at 14-30 DPSO, followed by increased reticulocytosis at 30-90 DPSO and ongoing inflammation at 90-180 DPSO in individuals reporting PS at 3-5 months.

2) Run the PS versus NPS comparisons for different age groups.

We have performed the comparison of PASC symptom groups for the parameters above using individuals >50 years of age. We do not have sufficient sample numbers for a more granular age split than this due to varying degrees of missingness for some timepoints and parameters). Shown below is the age of the two comparison groups, coloured by initial disease severity, and example findings for serum iron and hsCRP (see Supplementary Methods Figure SM3 for all measures). Reported trends in CRP, IL-6, serum iron and TSAT are still observable in this age group.

3) Correct for severity.

Initial disease severity is confounded with, and a likely contributor to, PASC in this cohort. Identifying independent risk factors for the latter is difficult for this reason. We do not have the sample numbers to compare PASC groups within each disease severity group (though an attempt at this was presented in Supplementary Figure 20 and shows differences between serum iron and CRP between PS and NPS groups *within* group E, albeit with low power).

We have run two analyses as the reviewer suggests:

- 1) Correcting for severity (encoded as an ordinal factor B→E) in the linear model of PASC group effects:

$\text{lm}(\log_2(\text{measure}) \sim \text{symptom group} + \text{severity group})$

Age is not included here as it is strongly confounded with severity. It was expected that a substantial proportion of variability in clinical measures would be accounted for by initial (acute) disease severity, and this was the case. However, even with correction for severity, lower serum iron ($p=0.03$) and TSAT ($p=0.02$) at 14-30 DPSO, and higher CRP ($p=0.053$) and IL-6 ($p=0.052$) at 90-180 DPSO remain associated with PASC (figure right). This suggests an association of these parameters, most notably early hypoferremia, with long-term symptoms that is independent of acute disease severity as defined by requirement for hospitalisation and respiratory support.

- 2) Comparison of PS and NPS groups in hospitalised patients (C-E), for which there is a more even split between long-term symptom groups than in mild (group B) patients. These two groups are the closest to age, sex and severity matched we are able to achieve.

Again, differences in iron and TSAT are observable at 14-30 DPSO, and in hsCRP and IL6 at 90-180 DPSO in this subsample (figure to right and below), supporting the observations presently emphasised in the main text which references these trends observed *within* groups B and E (Supplementary Figure 20).

Page 13, Line 313:

“Together, these findings implicate ~~the control and recovery of~~ early systemic inflammation and associated disruptions to iron handling, rather than the need for oxygen therapy per se, in risk of developing PASC months following acute disease.”

We have now included the above sensitivity analyses in Supplementary Figure 19, and reference them in the text:

Page 13, Line 300:

“PASC was strongly associated with acute disease severity in this cohort (Supplementary Figure 15D) as seen in previous studies implicating more severe disease and hospitalization with COVID-19 in worse long-term outcome^{8,32}. Dissecting early differences between those with and without long-term symptoms independent of this strong confounding is difficult, and may indeed be impossible if acute and long-term symptom severity are causally linked. We repeated the above analyses both with correction for

acute disease severity, and upon excluding group B patients to compare previously hospitalized PS and NPS groups better matched for age, sex and severity. In both severity corrected and severity matched analyses, serum iron and TSAT was significantly lower at 14-30 DPO in the PS vs NPS group. In the latter comparison, CRP and IL-6 also remained significantly elevated at 90-180 DPO in those with PS (Supplementary Figure 19C-F)."

We have now made it clearer to readers that we cannot determine if all features that correlate with long-term symptoms do so independently or via their association with acute disease severity. This may be because we do not have sufficient statistical power, or of course because the features do indeed have a real biological correlation with both acute disease severity and, through that, PASC. The close link between correlates of severity and PASC could be important clinically, suggesting the measures that reduce disease severity may reduce the incidence of PASC. We can, however, be confident that iron dysregulation is an early correlate of PASC that survives correction for disease severity, and that age disparities between patient groups do not influence our conclusions. The strength of this study lies in its thorough characterisation of the early and persisting dysregulation of iron recirculation in a human infectious disease, and the association of this with long-term outcome. As stated in our previous rebuttal, this presents avenues for exploring early clinical intervention strategies to prevent long-COVID that have thus far lacked direction from previous studies ill-suited to address this question.

We have added the following text to the discussion:

Page 19, Line 471:

"Here we show that the association between early iron dysregulation and long-term outcome of COVID-19 is independent of acute disease severity, but much larger, targeted studies will be required to explicitly disentangle other early correlates of PASC from those of severe COVID-19. The tight correlation between COVID-19 severity and PASC could be important clinically, as it suggests measures that reduce disease severity may reduce the incidence of PASC."

Reviewer #3 (Remarks to the Author):

In my review I only address the point if co-founders (most importantly: age, severity) were adequately addressed statistically. The main issue in this analysis is that a large proportion of the non-PASC patients are much younger, and they are patients that had lower degrees of the disease (A,B vs. C-E), Moreover, the groups have different sex distributions (see figure 1).

It is correct that the patients who go on to report no persisting symptoms (NPS) of COVID-19 are younger than those who do (PS), due in part to the over-representation of younger, milder disease (group B) patients in the NPS group. As stated above, this is shown in Supplementary Figure 15G. PS and NPS groups are however well sex matched at both the timepoints of questionnaire responses, as Supplementary Figure 15E shows.

To dissect if e.g. the altered iron homeostasis contributes to PASC or is simply a function of age and severity of the patient cohort, the authors in their revision use linear modelling. The authors don't provide a clear methods description (e.g. how did they encode their variables? Is age a number, or is it a factor?).

Age is treated as a continuous numerical variable, we have now clarified this in the methods:

Page 40, Line 866:

"Clinical variables (serum cytokines, inflammatory markers, iron and reticulocyte parameters and absolute cell counts) were normalised by log2 transformation and COVID-19 severity group effects tested using ~~conducted using Wilcoxon test, or~~ multivariate linear regression with correction for age (treated as

continuous integer value) and sex as covariates. PASC symptom group effects were tested with age correction only as NPS and PS groups were sex matched at both questionnaire timepoints.”

We have now shown (as detailed in response to Reviewer 2 above) that altered iron homeostasis is an early correlate of PASC, as detected in both age matched and age and severity matched PS and NPS groups. Hypoferremia at 14-30 DPSO, as reflected by low serum iron and TSAT levels in the PS group, is the finding which is most robust to the sensitivity analyses we have conducted. We are confident that these observations, and the conclusions we have drawn from them, are not due to age disparities between symptom groups, and not entirely driven by acute disease severity.

If (what I assume) the age is encoded as a number, this “correction” would only be appropriate if the effect of age is indeed linear, which I doubt. If it is non-linear, then unevenness in the age distributions will result in significant associations, even when there is none.

As we have demonstrated above, and presented in detail in the additional Supplementary Methods document, age has no detectable association with the majority (72%) of measured clinical and cellular parameters when assessed in HC, or in COVID-19 severity groups with correction for severity (Supplementary Methods Figures SM1-2, Tables SM1-4). Where age associations are detected, these association can be suitably modelled linearly. A test for non-linearity in HCs, incorporating a quadratic term for age in the model of age effects, confirms this (the quadratic age term is not significantly associated with any parameter; Table SM1). For continuity, age correction has been included in all analyses. We are grateful to the two reviewers who raised the issue of correction for age, as it is an important one to emphasise and address. We have robustly demonstrated that it does not impact on our key observation, that early iron dysregulation in COVID-19 is a strong correlate of PASC.

The (most simple, yet not perfect) way to overcome this would be to group patients into small age bins and then perform linear modelling, where a parameter is estimated for each age group. A much more intuitive and more appropriate way to analyze this data is to test the associations in sub-groups that are homogenous in age, disease severity and sex. So for instance, take each severity group with sufficient number of PASC, make sure that age and sex are comparable, and then test differences between PASC and non-PASC this within each group.

We simply do not have the sample numbers for this to be conclusive. This is a real-world dataset - we received follow-up responses from ~50% of recruited patients, and there is variable missingness across certain timepoints and measured parameters. Shown in response to Reviewer 2 above is an attempt to do this with an age and sex matched collection of hospitalised patients (groups C-E amalgamated), which shows hypoferremia in the PS group at 14-30 DPSO (the time at which the symptom groups are most clinically distinguishable), and elevated IL-6 and CRP at 90-180 DPSO, indicative of more slowly resolving/persisting low-level inflammation. We have now included this analysis in Supplementary Figure 19.

It would also be very beneficial to plot the measured parameter (such as iron) against age as a scatter plot for each disease severity, and mark PASC and non-PASC patients.

This is shown in the Supplementary Methods Figures SM1 and SM2 (severity groups) and SM4 (PASC groups). As explained above, stronger age effects observed in early disease are almost entirely driven by the confounding of age and disease severity, and age associations lose significance upon correction for disease severity for most parameters. Also explained above, PASC symptom group effect assessed using linear modelling with age correction can be recapitulated upon direct comparison of age matched groups.

To conclude, the current analysis is not suited to dissect if the associations observed are mainly due to cofounders, or if they are associated with PASC.

We have shown that age is not a significant confounder of the differences observed between PS and NPS symptom groups across measured parameters. Not only do those parameters relevant to the conclusions of this study (iron, haemoglobin, CRP, reticulocyte counts and haemoglobin content) lack an association with age, but significant PASC symptom group associations hold up in repeated analyses of age matched groups. Disease severity *is* a confounder of long-term outcome: it is likely to be a biological driver and clinically relevant. Irrespective of this, upon correcting for acute disease severity, and in severity matched groups, our primary observation that early hypoferrremia is seen in those who subsequently develop PASC is replicated. This is a finding we believe warrants acknowledgment in the literature, so it can be more thoroughly interrogated in cohorts more precisely matched for the plethora of confounders associated with severity and outcome of SARS-CoV2 infection.

Further review of the statistical methods was hindered as the methods are not fully disclosed (see e.g. above), and the data was not available for analysis, as the link to zonodo provided required a name and email for excess, preventing anonymous peer review. Data analysis code was not available.

We have added the required detail to the methods as described above. The Zenodo link was established to allow reviewers anonymous data access, and we had checked several times that this process was indeed anonymised. The link did however expire after one month, and may have done so by the time this Reviewer attempted to access the data. We have included a new link below, which should not expire. The R code for re-analysis of the data as presented here is included for Reviewers in this re-submission, which uses the downloadable data directly.

Data:

https://zenodo.org/records/10005084?token=eyJhbGciOiJIUzUxMiJ9.eyJpZCI6IjA3NDU5M2ViLTExMzgtNDI1ZS1iMWJlTE0NDZjNTg0YmI5MSIsImRhdGEiOnt9LCJyYW5kb20iOiJmNjJlMzE1ZDkwOTQ0ZWQ5ZTk3NjhlNjMxMDhlOTlmZCJ9.fQV-33AtFoRObqcMSiCOm-ve_wee9PFK-RvE-R5bQBUumoE_wjPcRknHc0XLdTjY_CY8bKBtlk4lNx5OhtKG3g

Code:

<https://zenodo.org/records/10005190?token=eyJhbGciOiJIUzUxMiJ9.eyJpZCI6IjExODU5M2ViLTExMzgtNDI1ZS1hOWE0LTA4Y2M4M2Q0M2Y0NCIsImRhdGEiOnt9LCJyYW5kb20iOiI4ZjA0YmZkOGJlZGQ1OGY2YjQyODliZGFIMWM0MmRjOSJ9.jA-xI2Tq34PddHeVBeWCq7jxYkZlBN087iukqIgg4EOLIHUeA6Q85t4QKfN-5Oi2MkrCM3-QQcjRBZj88FkGEg>

We thank both reviewers for their thorough evaluation of the statistical methods used in this study. We agree that careful considerations of confounders is important in observational studies such as this, and acknowledge that the original iterations of the text had not made clear to readers how potential confounding by age and severity had been addressed. We believe the new changes to the text and incorporated sensitivity analyses now strengthen the manuscript. Our work provides a thorough interrogation of disrupted homeostatic iron handling in the context of SARS-CoV2 infection. From our observations, it is the delayed resolution of this disequilibrium, with flow on effects for erythropoiesis and oxygen transport, that is a potentially clinically relevant early correlate of poor long-term outcome. We have shown here that the parameters upon which we base these conclusions are not age or acute disease severity dependent. This work thus describes early correlates of long-term outcome following COVID-19 that could be targeted to reduce the incidence of PASC.

Decision letter, third revision:

9th Nov 2023

Dear Dr. Hanson,

Thank you for submitting your revised manuscript "Iron Dysregulation and Inflammatory Stress Erythropoiesis Associates with Long-Term Outcome of COVID-19" (NI-A35512C). It has now been seen by the original referees and their comments are below. We are happy to inform you that if you revise your manuscript appropriately according to our editorial requirements, your manuscript should be publishable in Nature Immunology.

I will now pre-edit the current version of your paper. We will also perform detailed checks on your paper and will send you a checklist detailing our editorial and formatting requirements in about two weeks. Please do not upload the final materials and make any revisions until you receive this additional information from us.

In the meantime however, please deposit all omic and code data into public repositories so that the accession codes are readily available to be added in the revised manuscript. We cannot accept the paper without them. In addition, please check that the ORCID of all corresponding authors is linked to their Nature account, as this frequently causes delays at acceptance. Should you have any query or comments about ORCID, please do not hesitate to contact our editorial assistant at immunology@us.nature.com.

If you had not uploaded a Word file for the current version of the manuscript, we will need one before beginning the editing process; please email that to immunology@us.nature.com at your earliest convenience.

Thank you again for your interest in Nature Immunology. Please do not hesitate to contact me if you have any questions.

Sincerely,

Ioana Visan, Ph.D.
Senior Editor
Nature Immunology

Tel: 212-726-9207
Fax: 212-696-9752
www.nature.com/ni

Reviewer #2 (Remarks to the Author):

The authors addressed all my comments adequately.

Reviewer #3 (Remarks to the Author):

Again, in my review I only address the point if co-founders (most importantly: age, severity) were adequately addressed statistically. The authors now convincingly show that linear models are suitable, and also updated their methods such that the necessary details are incorporated, and also provide code and datasets. Therefore, I feel that this point has been addressed adequately.

Final Decision Letter:

Dear Dr. Smith,

I am delighted to accept your manuscript entitled "Iron Dysregulation and Inflammatory Stress Erythropoiesis Associates with Long-Term Outcome of COVID-19" for publication in an upcoming issue of Nature Immunology.

Over the next few weeks, your paper will be copyedited to ensure that it conforms to Nature Immunology style. Once your paper is typeset, you will receive an email with a link to choose the appropriate publishing options for your paper and our Author Services team will be in touch regarding any additional information that may be required.

Please note that *Nature Immunology* is a Transformative Journal (TJ). Authors may publish their research with us through the traditional subscription access route or make their paper immediately open access through payment of an article-processing charge (APC). Authors will not be required to make a final decision about access to their article until it has been accepted. [Find out more about Transformative Journals](https://www.springernature.com/gp/open-research/transformative-journals).

Authors may need to take specific actions to achieve [a](https://www.springernature.com/gp/open-research/funding/policy-compliance-)

faqs"> compliance with funder and institutional open access mandates. If your research is supported by a funder that requires immediate open access (e.g. according to Plan S principles) then you should select the gold OA route, and we will direct you to the compliant route where possible. For authors selecting the subscription publication route, the journal's standard licensing terms will need to be accepted, including self-archiving policies. Those licensing terms will supersede any other terms that the author or any third party may assert apply to any version of the manuscript.

Your paper will be published online soon after we receive your corrections and will appear in print in the next available issue.

Also, if you have any spectacular or outstanding figures or graphics associated with your manuscript - though not necessarily included with your submission - we'd be delighted to consider them as candidates for our cover. Simply send an electronic version (accompanied by a hard copy) to us with a possible cover caption enclosed.

If you have not already done so, we strongly recommend that you upload the step-by-step protocols used in this manuscript to the Protocol Exchange. Protocol Exchange is an open online resource that allows researchers to share their detailed experimental know-how. All uploaded protocols are made freely available, assigned DOIs for ease of citation and fully searchable through nature.com. Protocols can be linked to any publications in which they are used and will be linked to from your article. You can also establish a dedicated page to collect all your lab Protocols. By uploading your Protocols to Protocol Exchange, you are enabling researchers to more readily reproduce or adapt the methodology you use, as well as increasing the visibility of your protocols and papers. Upload your Protocols at www.nature.com/protocolexchange/. Further information can be found at

www.nature.com/protocolexchange/about .

Please note that we encourage the authors to self-archive their manuscript (the accepted version before copy editing) in their institutional repository, and in their funders' archives, six months after publication. Nature Portfolio recognizes the efforts of funding bodies to increase access of the research they fund, and strongly encourages authors to participate in such efforts. For information about our editorial policy, including license agreement and author copyright, please visit www.nature.com/ni/about/ed_policies/index.html

Sincerely,

Ioana Staicu, Ph.D.
Senior Editor
Nature Immunology

Tel: 212-726-9207
Fax: 212-696-9752
www.nature.com/ni